# CONTINUOUS DIFFUSION FOR MIXED-TYPE TABULAR DATA

**Markus Mueller**[1]
mueller@ese.eur.nl

**Kathrin Gruber**[1]
gruber@ese.eur.nl

**Dennis Fok**[1]
dfok@ese.eur.nl

[1]Econometric Institute, Erasmus University Rotterdam

## ABSTRACT

Score-based generative models, commonly referred to as diffusion models, have proven to be successful at generating text and image data. However, their adaptation to mixed-type tabular data remains underexplored. In this work, we propose CDTD, a Continuous Diffusion model for mixed-type Tabular Data. CDTD is based on a novel combination of score matching and score interpolation to enforce a unified continuous noise distribution for *both* continuous and categorical features. We explicitly acknowledge the necessity of homogenizing distinct data types by relying on model-specific loss calibration and initialization schemes. To further address the high heterogeneity in mixed-type tabular data, we introduce adaptive feature- or type-specific noise schedules. These ensure balanced generative performance across features and optimize the allocation of model capacity across features and diffusion time. Our experimental results show that CDTD consistently outperforms state-of-the-art benchmark models, captures feature correlations exceptionally well, and that heterogeneity in the noise schedule design boosts sample quality. Replication code is available at https://github.com/muellermarkus/cdtd.

## 1 INTRODUCTION

Score-based generative models (Song et al., 2021), also termed diffusion models (Sohl-Dickstein et al., 2015; Ho et al., 2020), have shown remarkable potential for the generation of images (Dhariwal & Nichol, 2021; Rombach et al., 2022), videos (Ho et al., 2022), text (Li et al., 2022; Dieleman et al., 2022; Wu et al., 2023), molecules (Hoogeboom et al., 2022), and other highly complex data structures with continuous features. The framework has since been adapted to categorical data in various ways, including discrete diffusion processes (Austin et al., 2021; Hoogeboom et al., 2021), diffusion in continuous embedding space (Dieleman et al., 2022; Li et al., 2022; Regol & Coates, 2023; Strudel et al., 2022), and others (Campbell et al., 2022; Meng et al., 2022; Sun et al., 2023). However, their adaptation to mixed-type tabular data, which includes both continuous and categorical features, remains under explored. Existing models build directly on advances from the image domain (Kim et al., 2023; Kotelnikov et al., 2023; Lee et al., 2023; Jolicoeur-Martineau et al., 2024) and, therefore, are not designed to deal with challenges specific to mixed-type tabular data: The type-specific diffusion processes and their losses are neither aligned nor balanced. A naive combination of different losses can cause the generative model to favor the sample quality of some features or data types over others (Ma et al., 2020). Furthermore, tabular data often includes categorical features of high cardinality. However, existing diffusion models for tabular data typically rely on a discrete diffusion framework to model one-hot-encoded categorical features (e.g., Kotelnikov et al., 2023; Lee et al., 2023). As a consequence, these models do not scale well and fail to capture the full uncertainty during the denoising process, as a data sample can never be 'in-between' categories.

Noise schedules determine the amount of noise at each diffusion timestep and are typically defined manually (Nichol & Dhariwal, 2021; Karras et al., 2022), but can also be learned (Dieleman et al., 2022; Kingma et al., 2021). They are a crucial component in score-based generative models (Kingma

et al., 2021; Chen et al., 2023; Chen, 2023; Jabri et al., 2023; Wu et al., 2023) as they aim to focus the model capacity on the noise levels most important to sample quality. However, despite their importance, previous work on diffusion models for tabular data simply adopted noise schedules from the image domain, which is not optimal. First, diffusion models for mixed-type tabular data inherently rely on combining type-specific diffusion processes. This makes it important, but difficult, to balance noise schedules across feature types. Unbalanced noise schedules negatively affect the allocation of model capacity. For instance, both TabDDPM (Kotelnikov et al., 2023) and CoDi (Lee et al., 2023) use the discrete multinomial diffusion framework (Hoogeboom et al., 2021) to model categorical features. This induces different types of noise for continuous and categorical features, making alignment or comparison of noise schedules impossible. Second, the domain, nature, and marginal distribution can vary significantly across features (Xu et al., 2019). For instance, any two continuous features may be subject to different levels of discretization or skewness, even after applying common data preprocessing techniques; and any two categorical features may differ in the number of categories, or the degree of imbalance. The high heterogeneity and lack of balance warrant a rethinking of fundamental parts of the diffusion framework, including the noise schedule and the effective combination of diffusion processes for different data types, rather than simply copying what has been working for images.

In this paper, we introduce *Continuous Diffusion for mixed-type Tabular Data* (CDTD) to address the aforementioned shortcomings. We combine *score matching* (Hyvärinen, 2005) with *score interpolation* (Dieleman et al., 2022) to derive a score-based model that pushes the diffusion process for categorical data into embedding space, and thus enables a Gaussian diffusion process for *both* continuous and categorical features. This way, the different noise processes become directly comparable, easier to balance, and enable the application of, for instance, classifier-free guidance (Ho & Salimans, 2022), accelerated sampling (Lu et al., 2022), and other advances to mixed-type tabular data.

We counteract the high feature heterogeneity inherent to mixed-type tabular data with distinct feature- or type-specific adaptive noise schedules. The learnable noise schedules allow the model to directly take feature or type heterogeneity into account during both training and generation, and thus avoid the reliance on image-specific noise schedule designs. Likewise, Shi et al. (2024) also propose the use of feature-specific noise schedules for tabular data but do not account for the necessity of homogenizing different data types. In contrast, we propose a diffusion-specific loss calibration and initialization scheme for effective data type homogenization. These improvements ensure a better allocation of our model's capacity across features, feature types and timesteps, and yield high quality samples. CDTD outperforms state-of-the-art baseline models across a diverse set of sample quality metrics and datasets as well as computation time. Our experiments show that CDTD captures feature correlations exceptionally well, and that explicitly allowing for data-type heterogeneity in the noise schedules benefits sample quality.

In sum, we make several contributions specific to score-based modeling of tabular data:

- We propose a unified continuous diffusion model for *both* continuous and categorical features such that all noise distributions are Gaussian.

- We balance model capacity across continuous and categorical features with a novel loss calibration, an adjusted score model initialization and adaptive type- or feature-specific noise schedules.

- We suggest a novel functional form to efficiently learn adaptive noise schedules, and to allow for an exact evaluation and incorporation of prior information on the relative importance of noise levels.

- We drastically improve the scalability of tabular data diffusion models to high-cardinality features.

- We are the first to enable the use of advanced techniques, like classifier-free guidance, for mixed-type tabular data *directly in data space* as opposed to a latent space.

## 2 SCORE-BASED GENERATIVE FRAMEWORK

We start with outlining the score-based frameworks for continuous and categorical features. Next, we combine these into a single, unified model to learn the joint distribution of mixed-type tabular data.

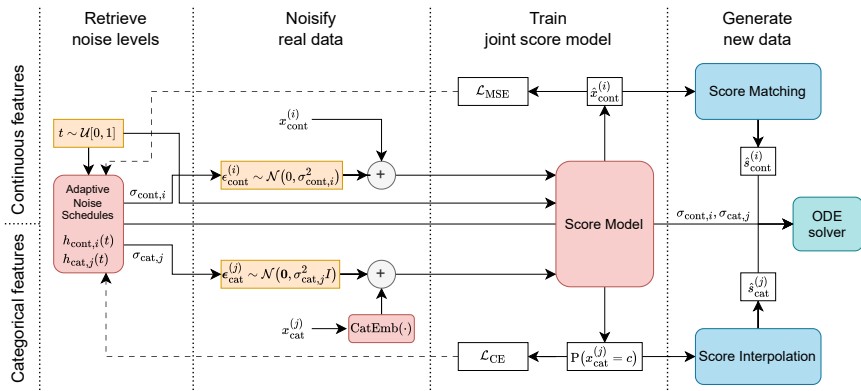

Figure 1: CDTD framework. Adaptive noise schedules are trained to fit the (possibly aggregated) MSE and CE losses and transform the uniform timestep $t$ to a potentially feature-specific noise level to diffuse ("noisify") the scalar values (for continuous features) or the embeddings (for categorical features). Associated sampling processes are highlighted in orange. The approximated score functions are concatenated and passed to an ODE solver for sample generation.

## 2.1 Continuous Features

We denote $x_{\text{cont}}^{(i)} \in \mathbb{R}$ as the $i$-th continuous feature and $\mathbf{x}_0 \equiv \mathbf{x}_{\text{cont}} \in \mathbb{R}^{K_{\text{cont}}}$ as the stacked feature vector. Further, let $\{\mathbf{x}_t\}_{t=0}^{t=1}$ be a diffusion process that gradually adds noise in continuous time $t \in [0, 1]$ to $\mathbf{x}_0$, and let $p_t(\mathbf{x})$ denote the density function of the data at time $t$. Then, this process transforms the real data distribution $p_0(\mathbf{x})$ into a terminal distribution of pure noise $p_1(\mathbf{x})$ from which we can sample. Our goal is to learn the reverse process that allows us to go from noise $\mathbf{x}_1 \sim p_1(\mathbf{x})$ to a new data sample $\mathbf{x}_0^* \sim p_0(\mathbf{x})$.

The forward-pass of this continuous-time diffusion process is formulated as the solution to a stochastic differential equation (SDE):

$$\mathrm{d}\mathbf{x} = \mathbf{f}(\mathbf{x}, t)\mathrm{d}t + g(t)\mathrm{d}\mathbf{w}, \tag{1}$$

where $\mathbf{f}(\cdot, t) : \mathbb{R}^{K_{\text{cont}}} \to \mathbb{R}^{K_{\text{cont}}}$ is the drift coefficient, $g(\cdot) : \mathbb{R} \to \mathbb{R}$ is the diffusion coefficient, and $\mathbf{w}$ is a Brownian motion (Song et al., 2021). The reversion yields the trajectory of $\mathbf{x}$ as $t$ goes backwards in time from 1 to 0, and is formulated as a probability flow ordinary differential equation (ODE):

$$\mathrm{d}\mathbf{x} = \left[\mathbf{f}(\mathbf{x}, t) - \frac{1}{2}g(t)^2 \nabla_{\mathbf{x}} \log p_t(\mathbf{x})\right]\mathrm{d}t. \tag{2}$$

We approximate the score function $\nabla_{\mathbf{x}} \log p_t(\mathbf{x})$, the only unknown in Equation (2), by training a time-dependent score-based model $s_{\boldsymbol{\theta}}(\mathbf{x}, t)$ via *score matching* (Hyvärinen, 2005). The parameters $\boldsymbol{\theta}$ are trained to minimize the *denoising score matching* objective:

$$\mathbb{E}_t\left[\lambda_t \mathbb{E}_{\mathbf{x}_0}\mathbb{E}_{\mathbf{x}_t|\mathbf{x}_0} \|s_{\boldsymbol{\theta}}(\mathbf{x}_t, t) - \nabla_{\mathbf{x}_t} \log p_{0t}(\mathbf{x}_t|\mathbf{x}_0)\|_2^2\right], \tag{3}$$

where $\lambda_t : [0, 1] \to \mathbb{R}_+$ is a positive weighting function for timesteps $t \sim \mathcal{U}_{[0,1]}$, and $p_{0t}(\mathbf{x}_t|\mathbf{x}_0)$ is the density of the noisy $\mathbf{x}_t$ given the ground-truth data $\mathbf{x}_0$ (Vincent, 2011).

In this paper, we use the EDM formulation (Karras et al., 2022), that is, $\mathbf{f}(\cdot, t) = \mathbf{0}$ and $g(t) = \sqrt{2[\frac{\mathrm{d}}{\mathrm{d}t}\sigma(t)]\sigma(t)}$ such that $p_{0t}(\mathbf{x}_t|\mathbf{x}_0) = \mathcal{N}(\mathbf{x}_t|\mathbf{x}_0, \sigma^2(t)\mathbf{I}_{K_{\text{cont}}})$. We standardize $\mathbf{x}_0$ to zero mean and unit variance. Then, for a sufficiently large $\sigma^2(1)$, we can start the reverse process with sampling $\mathbf{x}_1 \sim p_1(\mathbf{x}) = \mathcal{N}(\mathbf{0}, \sigma^2(1)\mathbf{I}_{K_{\text{cont}}})$. We then gradually guide $\mathbf{x}_1$ towards high density regions in the data space with $s_{\boldsymbol{\theta}}(\mathbf{x}, t)$ replacing the unknown, true score function in Equation (2).

## 2.2 Categorical Features

Let $x_{\text{cat}}^{(j)} \in \{1, \dots, C_j\}$ be the $j$-th categorical feature with $C_j$ distinct classes. We learn a feature-specific encoder to represent each category $c$ as a $d$-dimensional vector $\mathbf{e}_{x_{\text{cat}}}^{(j)} = \text{Enc}_j(x_{\text{cat}}^{(j)})$. Further,

let $\mathbf{x}_0^{(j)} \in \{\mathbf{e}_1^{(j)}, \ldots, \mathbf{e}_{C_j}^{(j)}\}$ be the noiseless embedding at $t = 0$. To unify the diffusion frameworks for categorical and continuous data as much as possible, we base *both* on the same Gaussian-type noise. Thus, instead of adding noise to the categorical variable directly, we add noise to the embedding $\mathbf{x}_t^{(j)} \sim p_{0t}(\mathbf{x}_t^{(j)}|\mathbf{x}_0^{(j)}) = \mathcal{N}(\mathbf{x}_t^{(j)}|\mathbf{x}_0^{(j)}, \sigma^2(t)I_d)$ such that $\mathbf{x}_1^{(j)} \sim p_1(\mathbf{x}^{(j)}) = \mathcal{N}(\mathbf{0}, \sigma^2(1)I_d)$, analogous to score matching.

For categorical data, denoising score matching (Equation (3)) is not directly applicable to learn $\nabla_{\mathbf{x}_t^{(j)}} \log p_{0t}(\mathbf{x}_t^{(j)}|\mathbf{x}_0^{(j)})$, since the score can only take on $C_j$ distinct values. To proceed, we transform the score matching approach into a discrete choice problem. Note that for a given $t$ and $\mathbf{x}_t^{(j)}$ it is sufficient to find $\mathbb{E}_{\mathrm{P}(\mathbf{x}_0^{(j)}|\mathbf{x}_t^{(j)},t)}[\nabla_{\mathbf{x}_t^{(j)}} \log p_{0t}(\mathbf{x}_t^{(j)}|\mathbf{x}_0^{(j)})]$ as it minimizes Equation (3). Presuming Gaussian noise, we have

$$\mathbb{E}_{\mathrm{P}(\mathbf{x}_0^{(j)}|\mathbf{x}_t^{(j)},t)}\left[\nabla_{\mathbf{x}_t^{(j)}} \log p_{0t}(\mathbf{x}_t^{(j)}|\mathbf{x}_0^{(j)})\right] = \frac{1}{\sigma^2(t)}\left[\mathbb{E}_{\mathrm{P}(\mathbf{x}_0^{(j)}|\mathbf{x}_t^{(j)},t)}[\mathbf{x}_0^{(j)}] - \mathbf{x}_t^{(j)}\right]. \tag{4}$$

We can thus approximate the score by computing the probability-weighted average of the $C_j$ possible embedding vectors, that is, $\hat{\mathbf{x}}_0^{(j)} = \mathbb{E}_{\mathrm{P}(\mathbf{x}_0^{(j)}|\mathbf{x}_t^{(j)},t)}[\mathbf{x}_0^{(j)}]$. Since $\mathrm{P}(\mathbf{x}_0^{(j)} = \mathbf{e}_{x_{\mathrm{cat}}}^{(j)}|\mathbf{x}_t^{(j)}, t) = \mathrm{P}(x_{\mathrm{cat}}^{(j)} = c|\mathbf{x}_t^{(j)}, t)$, we can estimate $\mathrm{P}(\mathbf{x}_0^{(j)}|\mathbf{x}_t^{(j)}, t)$ via a classifier that predicts the $C_j$ class probabilities and is trained to minimize the cross-entropy (CE). This procedure effectively interpolates between the $C_j$ ground-truth embeddings $\mathbf{x}_0^{(j)}$ and is therefore known as *score interpolation* (Dieleman et al., 2022).

This framework easily extends to multiple features. Most importantly, $\mathrm{Enc}_j$ is trained alongside the model such that $\mathbf{x}_0^{(j)}$ is directly optimized for denoising the data. During sampling, the model only has to commit to a category at the final step of generation, i.e., we allow for a smooth, continuous transition between states. This is unlike multinomial diffusion (Hoogeboom et al., 2021), which imposes *discrete* transitioning steps and is the framework used by several existing diffusion models for tabular data (Kotelnikov et al., 2023; Lee et al., 2023). By defining diffusion for categorical data in embedding space, we allow our model to take uncertainty at intermediate timesteps fully into account, which improves the consistency of the generated samples (Dieleman et al., 2022). Thus, CDTD can more accurately capture subtle dependencies both *within* and *across* data types.

## 3 METHOD

To model the joint distribution of mixed-type data, we combine *score matching* (Equation (3)) with *score interpolation* (Equation (4)). Next, we discuss the important components of our method. In particular, the combination of the different losses for score matching and score interpolation, initialization and loss weighting concerns, and the adaptive type- or feature-specific noise schedule designs.

### 3.1 GENERAL FRAMEWORK

Figure 1 gives an overview of our *Continuous Diffusion for mixed-type Tabular Data* (CDTD) framework. The score model is conditioned on (1) the noisy continuous features, (2) the noisy embeddings of categorical features, and (3) the timestep $t$. It predicts the ground-truth value $x_{\mathrm{cont}}^{(i)}$ for continuous features and the class-specific probabilities $\mathrm{P}(x_{\mathrm{cat}}^{(j)} = c)$ for categorical features. Additional conditioning information is straightforward to add. Note that while the Gaussian noise process acts directly on continuous features, it acts on the *embedded* categorical features $\mathbf{x}_0^{(j)}$. This way, we ensure a unified continuous noise process for both data types. Further details on the implementation are provided in Appendix K. During generation, we concatenate the score estimates, $\hat{s}_{\mathrm{cont}}^{(i)}$ and $\hat{s}_{\mathrm{cat}}^{(j)}$, for all features $i$ and $j$ before passing them to an ODE solver. Our model is the first to utilize feature- or type-specific noise schedules also during the generation of tabular data, which allows the sampler to take *feature- or type-specific steps*. Details on the sampling process and the algorithm are given in Appendix L.

## 3.2 HOMOGENIZATION OF DATA TYPES

Let $\mathcal{L}_{\text{MSE}}(x_{\text{cont}}^{(i)}, t)$ be the time-weighted MSE (i.e., score matching) loss of the $i$-th continuous feature at a single timestep $t$, and $\mathcal{L}_{\text{CE}}(x_{\text{cat}}^{(j)}, t)$ the CE (i.e., score interpolation) loss of the $j$-th categorical feature. Naturally, the two losses are defined on different scales. This leads to an unintended importance weighting of features in the generative process (Ma et al., 2020). To solve this, we observe that an unconditional generative model should a priori, i.e., without having any information, be indifferent between all features. For diffusion models this reflects the state of the model at the terminal timestep $t = 1$.

**Definition 1** (Calibrated losses). *Scaled losses $\mathcal{L}_{MSE}^*$ and $\mathcal{L}_{CE}^*$ are called calibrated if*

$$\mathbb{E}[\mathcal{L}_{MSE}^*(x_{cont}^{(i)}, 1)] = \mathbb{E}[\mathcal{L}_{CE}^*(x_{cat}^{(j)}, 1)] = 1,$$

*for all continuous features $i$ and categorical features $j$.*

**Assumption 1** (No information at $t = 1$). *The noise level $\sigma(1)$ is high enough such that the input to the score model cannot be distinguished from noise, i.e., the model has no information.*

**Proposition 1** (Homogenization of feature-specific losses). *Under Assumption 1, if each $x_{cont}^{(i)}$ has unit variance and $Z_j$ is the feature-specific entropy of categorical feature $j$, then the losses $\mathcal{L}_{MSE}^* = \mathcal{L}_{MSE}$ and $\mathcal{L}_{CE}^* = \mathcal{L}_{CE}(x_{cat}^{(j)}, 1)/Z_j$ are calibrated for all continuous features $i$ and categorical features $j$. See Appendix B for a detailed proof.*

Given Assumption 1 and Proposition 1, we can derive a joint loss function without unintended importance weighting by averaging the $K = K_{\text{cont}} + K_{\text{cat}}$ calibrated losses at a given $t$:

$$\mathcal{L}(t) = \frac{1}{K}\Big[\sum_{i=1}^{K_{\text{cont}}} \mathcal{L}_{\text{MSE}}^*(x_{\text{cont}}^{(i)}, t) + \sum_{j=1}^{K_{\text{cat}}} \mathcal{L}_{\text{CE}}^*(x_{\text{cat}}^{(j)}, t)\Big]. \tag{5}$$

**Implications for the score model initialization.** The loss calibration and the multi-modality of the data have implications for the optimal initialization of the score model. To match the loss calibration, we aim to initialize the model such that all *feature-specific* losses are one. We therefore initialize the output layer weights and biases for continuous features to zero and rely on the timestep weights of the EDM parameterization (Karras et al., 2022) to achieve a unit loss for all $t$. For the categorical features, we initialize the biases to match each category's empirical log probability in the training set (see Appendix C for details).

**Weighting across time.** The initial equal importance across all $t$ will change over the course of training. To allow for changes in the relative importance among features but ensure equal importance of all timesteps throughout training, we employ a normalization scheme for the average diffusion loss (Karras et al., 2024; Kingma & Gao, 2023). Specifically, we learn the time-dependent normalization $Z(t)$ such that $\mathcal{L}(t)/Z(t) \approx 1$. This ensures a consistent gradient signal and can be implemented by training a neural network to predict $\mathcal{L}(t)$ alongside our diffusion model (see Appendix D for details).

## 3.3 NOISE SCHEDULES

Evidently, the noise schedule of one feature impacts the optimality of noise schedules for other features, and different data types have different sensitivities to additive noise. For instance, given the same embedding dimension, more noise may be needed to remove the same amount of signal from embeddings of features with fewer classes (see Appendix A). Likewise, a delayed noise schedule for one feature might improve sample quality as the model can rely on other correlated features that have been (partially) generated first. Therefore, we introduce *feature-specific* or *type-specific* noise schedules. We make the noise schedules learnable, and therewith *adaptive* to avoid the reliance on designs for other data modalities.

We investigate the following noise schedule variants: (1) a single adaptive noise schedule, (2) adaptive noise schedules differentiated per data type and (3) feature-specific adaptive noise schedules. We only introduce the feature-specific noise schedules explicitly. The other noise schedule types are easily derived from our argument by appropriately aggregating terms across features.

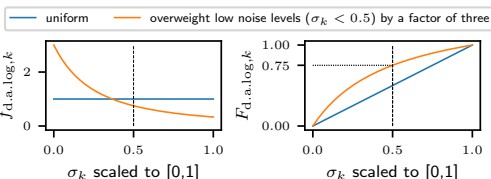

Figure 2: (Left) pdf ($f_{\mathrm{d.a.log},k}$) and cdf ($F_{\mathrm{d.a.log},k}$) of the domain-adapted Logistic distribution for five different values of the location parameter $\mu_k$ and for a given curve steepness $\nu_k = 3$. (Right) impact of uniform vs. adjusted timewarping initialization on the pdf ($f_{\mathrm{d.a.log},k}$) and the cdf ($F_{\mathrm{d.a.log},k}$).

**Feature-specific noise schedules.** Following Equation (1), we define the diffusion process of $x_{\mathrm{cont}}^{(i)}$ as

$$\mathrm{d}x_{\mathrm{cont}}^{(i)} = \sqrt{2\left[\frac{\mathrm{d}}{\mathrm{d}t}h_{\mathrm{cont},i}(t)\right]h_{\mathrm{cont},i}(t)}\,\mathrm{d}w_t^{(i)}, \tag{6}$$

and likewise the trajectory of $\mathbf{x}_{\mathrm{cat}}^{(j)}$ as

$$\mathrm{d}\mathbf{x}_{\mathrm{cat}}^{(j)} = \sqrt{2\left[\frac{\mathrm{d}}{\mathrm{d}t}h_{\mathrm{cat},j}(t)\right]h_{\mathrm{cat},j}(t)}\,\mathrm{d}\mathbf{w}_t^{(j)}, \tag{7}$$

where $\mathbf{x}_{\mathrm{cat}}^{(j)} \in \mathbb{R}^d$ is the embedding of $x_{\mathrm{cat}}^{(j)}$ in Euclidean space. The noise schedules $h_{\mathrm{cont},i}(t)$ and $h_{\mathrm{cat},j}(t)$ represent the *feature-specific* standard deviations $\sigma_{\mathrm{cont},i}(t)$ and $\sigma_{\mathrm{cat},j}(t)$ of the added Gaussian noise, respectively. Thus, each feature is affected by a distinct adaptive noise schedule. On the other hand, *type-specific* noise schedules involve only two functions, $h_{\mathrm{cat}}(t)$ and $h_{\mathrm{cont}}(t)$, such that all features of the same type are affected by the same noise schedule.

**Adaptive noise schedules.** We aim to learn a noise schedule $h_k : t \mapsto \sigma_k$ for each feature $k$. Inspired by Dieleman et al. (2022), we learn a function $F_k$ that predicts the feature-specific (not explicitly weighted) loss $\ell_k$ given the noise level $\sigma_k$. By normalizing and inverting $F_k$, we achieve the mapping of interest $h_k = \tilde{F}_k^{-1}$. This encourages the relation between $t$ and $\ell_k$ to be linear.

Higher noise levels imply a lower signal-to-noise ratio and therefore a larger incurred loss for the score model. Accordingly, $F_k$ must be a monotonically increasing and *S-shaped* function. We let $F_k = \gamma_k F_{\mathrm{d.a.log},k}(\sigma_k)$ where $\gamma_k > 0$ is a scaling factor that enables fitting a loss $\ell_k > 1$ at $t = 1$ near the start of training, and a loss $\ell_k < 1$ in case conditioning information is included. Further, we use the cdf of the domain-adapted Logistic distribution $F_{\mathrm{d.a.log},k}(\sigma_k)$, where the input is pre-processed via a Logit function, with parameters $0 < \mu_k < 1$ (the location of the inflection point) and $\nu_k \geq 1$ (the steepness of the curve). Note that with pre-specified minimum and maximum noise levels, we can always scale $\sigma_k$ to lie in $[0, 1]$. Figure 2 illustrates the effect of the location parameter. The implicit importance of the noise levels is conveniently represented by the pdf $f_{\mathrm{d.a.log},k}$. To normalize and invert $F_k$, we set $\gamma_k = 1$ and directly use the closed-form quantile function $F_{\mathrm{d.a.log},k}^{-1}$. The detailed derivation of all relevant functions is given in Appendix E. To avoid biasing the noise schedule to frequently sampled timesteps during training, we derive importance weights from $f_{\mathrm{d.a.log},k}$ when fitting $h_k$. We use the adaptive noise schedules during both training and generation, and give examples of learned noise schedules in Appendix P.

Our functional choice has several advantages. First, each noise schedule can be evaluated exactly without the need for approximations and only requires three parameters. Second, these parameters are well interpretable in the diffusion context and provide information on the inner workings of the model. For instance, for $\mu_1 < \mu_2$, the model starts generating feature 2 before feature 1 in the reverse process. Third, the proposed functional form is less flexible than the original piece-wise linear function (Dieleman et al., 2022) such that an EMA on the parameters is not necessary, and the fit is more robust to "outliers" encountered during training. This is crucial when using a feature-specific specification as opposed to a single noise schedule.

### 3.4 ADDITIONAL CUSTOMIZATION TO TABULAR DATA

In the diffusion process, we add noise directly to the continuous features but to the embeddings of categorical features. We generally need more noise to remove all the signal from the categorical representations. We therefore define *type-specific* minimum and maximum noise levels: For categorical

features, we let $\sigma_{\text{cat,min}} = 0$ and $\sigma_{\text{cat,max}} = 100$; for continuous features, we set $\sigma_{\text{cont,min}} = 0$ and $\sigma_{\text{cont,max}} = 80$.

Lastly, an uninformative initialization of the adaptive noise schedules requires $\mu_k = 0.5, \nu_k \approx 1$ and $\gamma_k = 1$ such that $F_{\text{d.a.log},k}$ corresponds approximately to the cdf of a uniform distribution. We can improve upon this with a more informative prior: As opposed to images, in tabular data the location of features in the data matrix, and therefore the high-level structure, is fixed. Instead, we are interested in generating details as accurately as possible. Note that the inflection point, $\mu_k$, corresponds to the proportion of high noise levels (i.e., $\sigma_k \geq 0.5 \cdot (\sigma_{\max} + \sigma_{\min})$) in the distribution. Therefore, we empirically choose $\mu_k = 1/4$ such that low noise levels (i.e., $\sigma_k < 0.5 \cdot (\sigma_{\max} + \sigma_{\min})$) are initially presumed to be three times more important than high noise levels (see Figure 2). We let $\nu_k \approx 1$ for a dispersed initial probability mass and initialize the scaling factor to $\gamma_k = 1$.

## 4 EXPERIMENTS

We benchmark our model against several generative models across multiple datasets. Additionally, we investigate three different noise schedule specifications: (1) a single adaptive noise schedule for both data types (*single*), (2) continuous and categorical data type-specific adaptive noise schedules (*per type*), and (3) feature-specific adaptive noise schedules (*per feature*).

**Baseline models.**   We use a diverse benchmark set of state-of-the-art generative models for mixed-type tabular data. This includes SMOTE (Chawla et al., 2002), ARF (Watson et al., 2023), CTGAN (Xu et al., 2019), TVAE (Xu et al., 2019), TabDDPM (Kotelnikov et al., 2023), CoDi (Lee et al., 2023) and TabSyn (Zhang et al., 2024). Each model follows a different design and/or modeling philosophy. Note that CoDi is an extension of STaSy (Kim et al., 2023, the same group of authors) that has shown to be superior in performance. For scaling reasons, ForestDiffusion (Jolicoeur-Martineau et al., 2024) is not an applicable benchmark.[1] Further details on the respective benchmark models and their implementations are provided in Appendix G and Appendix H, respectively. We provide an in-depth comparison of CDTD to the diffusion-based baselines in Appendix O. To keep the comparison fair, we use the same internal architecture for CDTD as TabDDPM (which was also adopted by TabSyn), with minor changes to accommodate the different inputs (see Appendix K).

**Datasets.**   We systematically investigate our model on 10 publicly available datasets (see Appendix F for details). The datasets vary in size, prediction task (regression vs. binary classification[2]), number of continuous *and* categorical features and their distributions. The number of categories for categorical features varies significantly across datasets. We remove observations with missings in the target or any of the continuous features and encode missings in the categorical features as a separate category. All datasets are split in train (60%), validation (20%) and test (20%) partitions, hereinafter denoted $\mathcal{D}_{\text{train}}, \mathcal{D}_{\text{valid}}$ and $\mathcal{D}_{\text{test}}$, respectively. For classification tasks, we use stratification with respect to the outcome. We round the integer-valued continuous features after generation.

### 4.1 EVALUATION METRICS

In our experiments, we follow conventions from previous papers and use four sample quality criteria, which we assess using a comprehensive set of measures. All metrics are averaged over five random seeds that affect the sampling process of synthetic data $\mathcal{D}_{\text{gen}}$ of size $\min(|\mathcal{D}_{\text{train}}|, 50\,000)$.

**Machine learning efficiency.**   We follow the conventional train-synthetic-test-real strategy (see, Borisov et al., 2023; Liu et al., 2023; Kotelnikov et al., 2023; Kim et al., 2023; Xu et al., 2019; Watson et al., 2023). We train a logistic/ridge regression, a random forest and a catboost model, on the data-specific prediction task (see Appendix J for details) and then compare the model-averaged

---

[1] Jolicoeur-Martineau et al. (2024) report that they used 10-20 CPUs with 64-256 GB of memory for datasets with a median number of 540 observations. With the suggested hyperparameters (for improved efficiency) and 64 CPUs, the model took approx. 500 min of training on the small `nmes` data. Note that the model estimates $KT$ separate models, with $K$ being the number of features and $T$ the noise levels. Therefore, we consider ForestDiffusion to be prohibitively expensive for higher-dimensional data generation.

[2] For ease of presentation, we only analyze binary targets. However, CDTD trivially extends to targets with multiple classes.

real test performance, $\text{Perf}(\mathcal{D}_{\text{train}}, \mathcal{D}_{\text{test}})$, to the performance when trained on the synthetic data, $\text{Perf}(\mathcal{D}_{\text{gen}}, \mathcal{D}_{\text{test}})$. The results are averaged over ten different model seeds (in addition to the five seeds for the sampling). For regression tasks, we consider the RMSE and for classification tasks, the macro-averaged F1 and AUC scores. We report $|\text{Perf}(\mathcal{D}_{\text{gen}}, \mathcal{D}_{\text{test}}) - \text{Perf}(\mathcal{D}_{\text{train}}, \mathcal{D}_{\text{test}})|$. An absolute difference close to zero is preferable since then synthetic and real data induce the same performance.

**Detection score.** For each generative model, we report the accuracy of a catboost model that is trained to distinguish between real and generated (fake) samples (Borisov et al., 2023; Liu et al., 2023; Zhang et al., 2024). First, we subsample the real data subsets, $\mathcal{D}_{\text{train}}$, $\mathcal{D}_{\text{valid}}$ and $\mathcal{D}_{\text{test}}$, to a maximum of 25 000 data samples to limit evaluation time. Then, we construct $\mathcal{D}_{\text{train}}^{\text{detect}}$, $\mathcal{D}_{\text{valid}}^{\text{detect}}$ and $\mathcal{D}_{\text{test}}^{\text{detect}}$ with equal proportions of real and fake samples. We tune each catboost model on $\mathcal{D}_{\text{valid}}^{\text{detect}}$ and report the accuracy of the best-fitting model on $\mathcal{D}_{\text{test}}^{\text{detect}}$ (see Appendix I for details). A (perfect) detection score of 0.5 indicates that the model is unable to distinguish fake from real samples.

**Statistical similarity.** Similar to Zhang et al. (2024), we assess the statistical similarity between real and generated data at both the feature and sample levels. We largely follow Zhao et al. (2021) and compare: (1) the Jensen-Shannon divergence (JSD; Lin, 1991) to quantify the difference in categorical distributions, (2) the Wasserstein distance (WD; Ramdas et al., 2017) to quantify the difference in continuous distributions, and (3) the $L_2$ distance between pairwise correlation matrices. We use the Pearson correlation coefficient for two continuous features, the Theil uncertainty coefficient for two categorical features, and the correlation ratio for mixed types.

**Privacy.** We compute the distance to closest record (DCR) as the minimum Euclidean distance of a generated data point to any observation in $\mathcal{D}_{\text{train}}$ (Borisov et al., 2023; Zhao et al., 2021). We one-hot encode categorical features and standardize all features to zero mean and unit variance to ensure each feature contributes equally to the distance. We compute the average DCR as a robust estimate. For brevity, we report the absolute difference of the DCR of the synthetic data and the DCR of the real test set. A good DCR value, indicating both realistic and sufficiently private data, should be close to zero.

## 4.2 RESULTS

Table 1: Average performance rank of each generative model across eleven datasets. Per metric, **bold** indicates the best, underline the second best result. We assigned the rank 10 for CoDi on `lending` and `diabetes`, TabDDPM on `acsincome` and `diabetes`, SMOTE on `acsincome` and `covertype`. RMSE, F1, AUC and DCR are measured in abs. differences to the real test set.

| | SMOTE | ARF | CTGAN | TVAE | TabDDPM | CoDi | TabSyn | CDTD (single) | CDTD (per type) | CDTD (per feature) |
|---|---|---|---|---|---|---|---|---|---|---|
| RMSE | $3.6_{\pm 3.3}$ | $3.4_{\pm 2.1}$ | $8.0_{\pm 1.6}$ | $7.8_{\pm 2.2}$ | $8.4_{\pm 1.0}$ | $7.0_{\pm 1.8}$ | $7.2_{\pm 1.5}$ | $4.0_{\pm 1.7}$ | $\mathbf{2.6}_{\pm 1.0}$ | $\underline{3.2}_{\pm 1.5}$ |
| F1 | $4.2_{\pm 2.9}$ | $6.2_{\pm 2.3}$ | $8.0_{\pm 1.4}$ | $8.0_{\pm 1.0}$ | $4.3_{\pm 3.1}$ | $6.5_{\pm 3.0}$ | $8.2_{\pm 1.7}$ | $4.0_{\pm 1.1}$ | $\mathbf{2.3}_{\pm 1.4}$ | $\underline{3.5}_{\pm 1.4}$ |
| AUC | $4.3_{\pm 3.0}$ | $5.7_{\pm 1.9}$ | $8.3_{\pm 1.2}$ | $7.8_{\pm 1.1}$ | $4.3_{\pm 3.1}$ | $6.8_{\pm 3.1}$ | $8.2_{\pm 1.6}$ | $\underline{3.3}_{\pm 1.1}$ | $\mathbf{2.7}_{\pm 1.5}$ | $3.7_{\pm 1.2}$ |
| $L_2$ dist. of corr. | $5.0_{\pm 2.8}$ | $5.7_{\pm 2.2}$ | $8.3_{\pm 1.9}$ | $7.9_{\pm 1.4}$ | $6.0_{\pm 3.3}$ | $7.0_{\pm 2.5}$ | $7.1_{\pm 1.2}$ | $3.4_{\pm 1.1}$ | $\mathbf{2.0}_{\pm 0.8}$ | $\underline{2.8}_{\pm 1.5}$ |
| Detection score | $5.7_{\pm 2.5}$ | $6.1_{\pm 1.7}$ | $8.8_{\pm 1.5}$ | $7.6_{\pm 1.1}$ | $4.8_{\pm 3.2}$ | $7.9_{\pm 2.4}$ | $6.2_{\pm 1.6}$ | $\underline{3.1}_{\pm 1.5}$ | $\mathbf{1.6}_{\pm 1.0}$ | $3.3_{\pm 1.1}$ |
| JSD | $7.2_{\pm 2.2}$ | $\mathbf{1.2}_{\pm 0.4}$ | $7.6_{\pm 2.4}$ | $8.8_{\pm 1.0}$ | $6.9_{\pm 2.1}$ | $7.2_{\pm 1.5}$ | $6.8_{\pm 1.3}$ | $\underline{2.5}_{\pm 0.7}$ | $2.8_{\pm 1.1}$ | $4.1_{\pm 0.9}$ |
| WD | $\underline{3.2}_{\pm 3.3}$ | $5.8_{\pm 1.8}$ | $7.4_{\pm 2.4}$ | $7.9_{\pm 1.6}$ | $5.6_{\pm 3.5}$ | $8.4_{\pm 1.8}$ | $5.8_{\pm 2.0}$ | $4.5_{\pm 1.6}$ | $\mathbf{3.2}_{\pm 1.5}$ | $3.4_{\pm 1.9}$ |
| DCR | $5.8_{\pm 2.7}$ | $6.5_{\pm 2.1}$ | $8.4_{\pm 1.7}$ | $6.1_{\pm 2.9}$ | $4.5_{\pm 3.4}$ | $6.6_{\pm 2.4}$ | $6.3_{\pm 2.0}$ | $4.1_{\pm 2.5}$ | $\underline{3.9}_{\pm 2.3}$ | $\mathbf{3.0}_{\pm 2.0}$ |

Table 1 shows the average rank of each generative model across all datasets for the considered metrics. Detailed results (including standard errors) for each model and dataset are reported in Appendix S. The ranks in terms of the F1 and AUC scores are averaged over the classification task datasets. Likewise, the RMSE rank averages include the regression task datasets. We assign the maximum possible rank when a model could not be trained on a given dataset or could not be evaluated in reasonable time. This includes TabDDPM, which outputs NaNs for `acsincome` and `diabetes` and CoDi, which we consider to be prohibitively expensive to train on `diabetes` (estimated 14.5 hours) and `lending` (estimated 60 hours). Similarly, SMOTE is very inefficient in sampling for large datasets (78 min for 1000 samples on `acsincome` and 182 min on `covertype`) and does not finish the evaluation within 12 hours. We provide visualizations of the captured correlations in the synthetic sample compared to the real training set in Appendix R and distribution plots for a qualitative comparison in Appendix Q.

Table 2: Ablation study for five CDTD configurations. We report the median performance. The grey column depicts the impact of our data type homogenization.

| Config. | A | B | C | D | CDTD (per type) |
|---|---|---|---|---|---|
| RMSE (abs. diff.; ↓) | 0.090 | 0.069 | 0.060 | 0.058 | 0.055 |
| F1 (abs. diff.; ↓) | 0.015 | 0.011 | 0.010 | 0.006 | 0.009 |
| AUC (abs. diff.; ↓) | 0.008 | 0.005 | 0.005 | 0.005 | 0.006 |
| $L_2$ distance of corr. (↓) | 0.483 | 0.421 | 0.457 | 0.450 | 0.444 |
| Detection score (↓) | 0.762 | 0.739 | 0.774 | 0.662 | 0.701 |
| JSD (↓) | 0.010 | 0.013 | 0.011 | 0.011 | 0.011 |
| WD (↓) | 0.008 | 0.006 | 0.007 | 0.005 | 0.006 |
| DCR (abs. diff. to test; ↓) | 0.639 | 0.552 | 0.574 | 0.350 | 0.544 |

**Sample quality.** CDTD consistently outperforms the considered benchmark models in most sample quality metrics. Specifically, we see a major performance edge in terms of the detection score, the $L_2$ distance of the correlation matrices and the ML efficiency metrics. Using score interpolation, CDTD is able to model the intricate correlation structure more accurately than other frameworks. Due to their inductive biases, SMOTE and ARF perform very well in terms of univariate distribution fit for continuous features (as measured by WD) and categorical (as measured by JSD), respectively. However, in both of those cases, CDTD performs competitively, in particular when compared to the diffusion-based models. Interestingly, TabSyn, a latent space diffusion model, performs considerably worse than CDTD and often TabDDPM, which define diffusion in data space. In Appendix N, we further compare CDTD and TabSyn and investigate the benefits of defining a diffusion model in data space.

Most importantly, type-specific noise schedules mostly outperform the feature-specific and single noise schedule variants. This illustrates the necessity to account for the high heterogeneity in tabular data on the feature-type level. Having distinct noise schedules per feature instead appears to force too many constraints on the model and, thus, decreases sample quality. Per-feature noise schedules would also require more training steps to fully converge, as can be seen in Appendix P. We investigate the sensitivity of CDTD to important hyperparameters in Appendix M.

**Training and sampling time.** Figure 3 shows the average training and sampling wall-clock times (for all feasible models) over all datasets (see Appendix U for details on sample quality as a function of sampling time). We exclude SMOTE due to its considerably longer sampling times with an average of 1377 seconds for 1000 samples. CDTD's use of embeddings (instead of one-hot encoding) for categorical features improves its scaling to increasing number of categories and thus, drastically reduces training times. The sampling speed of CDTD is

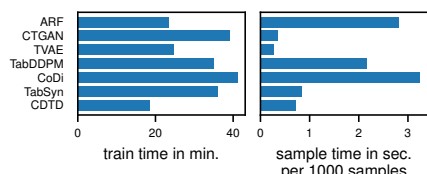

Figure 3: Average training and sampling wall-clock times (excl. SMOTE, `acsincome`, `diabetes`, `lending`).

competitive, in particular compared to the diffusion-based benchmarks CoDi, TabDDPM and TabSyn.

**Ablation study.** We investigate the separate components of our CDTD framework. The summarized results are given in Table 2 and the detailed results in Appendix T. The baseline model *Config. A* includes a single noise schedule with the original piece-wise linear functional form (Dieleman et al., 2022) and the CE and MSE losses are naively averaged. Note that this configuration is still a novel contribution. *Config. B* adds our data type homogenization (i.e., loss calibration, improved initialization and time-dependent normalization schemes), *Config. C* adds our proposed functional form for a single noise schedule with uniform initialization, and *Config. D* imposes per-type noise schedules. Lastly, we increase the importance of low noise levels at initialization to arrive at the full CDTD (*per type*) model.

The results show that data type homogenization benefits sample quality significantly. Metrics associated with continuous features, i.e., RMSE and WD, as well as those relying on *all* features being generated well, i.e., the detection score, $L_2$ distance of corr. matrices and DCR, improve dramatically. Switching from the piecewise linear noise schedule to our more robust functional form slightly harms sample quality. However, the per-type variant and the improved initialization more than compensate for this. The latter appears to trade-off some sample quality for faster convergence during training.

## 5    CONCLUSION AND DISCUSSION

In this paper, we introduce a Continuous Diffusion model for mixed-type Tabular Data (CDTD) that combines score matching and score interpolation and imposes Gaussian diffusion processes on both continuous and embedded categorical features. Our results indicate that addressing the high feature heterogeneity in tabular data on the feature type level by aligning type-specific diffusion elements, such as the noise schedules or losses, substantially benefits sample quality. Moreover, CDTD shows vastly improved scalability and can accommodate an arbitrary number of categories.

Our paper serves as an important step to customizing score-based models to tabular data. The common type of noise schedules allows for an easy to extend framework. Crucially, CDTD allows the direct application of diffusion-related advances from the image domain, like classifier-free guidance, to tabular data without the need for a latent encoding. We leave further extensions to the tabular data domain, e.g., the exploration of accelerated sampling, efficient score model architectures, or the adaptation to the data imputation task for future work.

Finally, we want to warn against the potential misuse of synthetic data to support unwarranted claims. Any generated data should not be blindly trusted, and synthetic data based inferences should always be compared to results from the real data. However, the correct use of generative models for tabular data enables better privacy preservation and facilitates data sharing and open science practices.

### LIMITATIONS

The main limitation of CDTD is the addition of hyperparameters, and tuning hyperparameters of a generative model can be a costly endeavor. However, our results also show that (1) a per type schedule is most often optimal and (2) our default hyperparameters perform well across a diverse set of datasets. Dieleman et al. (2022) show that the results of score interpolation for text data can be sensitive to the initialization of the embeddings. We have not encountered similar problems on tabular datasets (see Table 7). While the DCR indicates no privacy issues for the benchmark datasets used, additional caution must be taken when generating synthetic data from privacy sensitive sources. Lastly, for specific types of tabular data, such as time-series, our model may be outperformed by other generative models specialized for that type. While CDTD could be directly used for imputation using RePaint (Lugmayr et al., 2022), a separate training process is required to achieve the best results (Liu et al., 2024). Therefore, we leave the adaptation of CDTD to the imputation task for future work.

### ACKNOWLEDGEMENTS

This work used the Dutch national e-infrastructure with the support of the SURF Cooperative using grant no. EINF-7437. We also thank Sander Dieleman for helpful discussions.

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

## A  NOISE IMPLICATIONS OF HETEROGENEOUS CARDINALITIES

One distinct characteristic of mixed-type tabular data is that each categorical feature may have a different cardinality, i.e., a different number of categories. Below, we briefly empirically investigate how this data characteristic warrants a rethinking of diffusion noise schedules. We show that, under some assumptions, categorical features of higher cardinality require more added noise than lower cardinality features to remove the same amount of signal.

Let $x_{\text{cat}}^{(j)}$ be a single categorical feature with $C_j$ categories. We assume that each category $c$ has equal probability $\text{P}(c) = 1/C_j$ and train a CDTD model with a fixed, linear noise schedule to learn the distribution of a single $x_{\text{cat}}^{(j)}$. For each class $c \in \{1, \ldots, C_j\}$, we extract the associated learned embedding $\mathbf{e}_c^{(j)}$. For 500 timesteps $t$, linearly spaced on $[0, 1]$, we derive the associated noise levels $\sigma(t)$ from the learned noise schedules and for each $\sigma(t)$, we sample 1000 noisy embeddings $\mathbf{e}_c^{(j)}(t) = \mathbf{e}_c^{(j)} + \sigma(t)\epsilon$ such that $\epsilon \sim \mathcal{N}(\mathbf{0}, \mathbf{I})$. We input all $\mathbf{e}_c^{(j)}(t)$ into the score model to compute the average (calibrated) cross-entropy (CE) loss. We define the remaining signal of the feature at a given $t$ as $S_c = 1 - \text{CE}_c(t)$, where $\text{CE}_c$ reflects the CE loss when using the noisy embedding for class $c$ to predict the true class. Lastly, we compute the overall remaining signal over all classes as $\max_c S_c$. Figure 4 shows the result of repeating this procedure with varying $C_j$ and four-dimensional embeddings. As we can see, lower cardinality features, i.e., those with a low $C_j$, attain the same signal only at a higher noise levels $\sigma(t)$. This example of the effect of feature heterogeneity in tabular data encourages us to investigate *feature-specific* noise schedules.

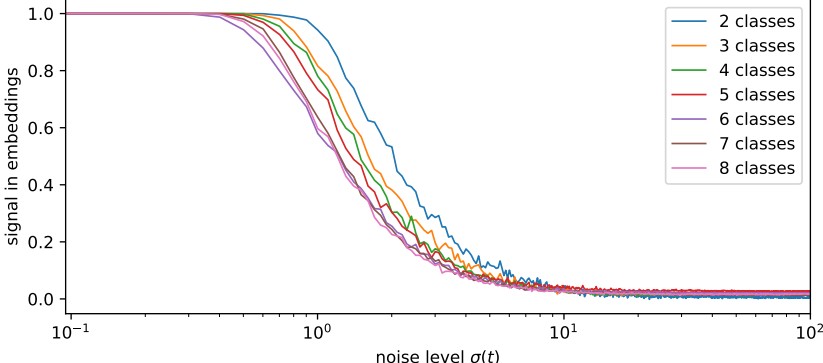

Figure 4: Comparison of signal remaining in embedded categorical features given by $\max_c S_c$, where $S_c = 1 - \text{CE}_c(t)$ and $\text{CE}_c$ reflects the CE loss when using the noisy embedding for class $c \in \{1, \ldots, C_j\}$ to predict the true class. Lower cardinality features tend to require a higher noise level $\sigma(t)$ to achieve the same amount of remaining signal as higher cardinality features.

## B  LOSS CALIBRATION

Under Assumption 1, the signal-to-noise ratio at the terminal timestep is sufficiently low to approximate a situation in which the model has no information about the data. We want to let the model be indifferent between features, that is, we scale the loss of each feature such that at the terminal timestep the same expected loss is attained. Therefore, we are looking for scaled losses $\mathcal{L}_{\text{MSE}}^*(x_{\text{cont}}^{(i)}, 1)$ and $\mathcal{L}_{\text{CE}}^*(x_{\text{cat}}^{(j)}, 1)$ which at $t = 1$ achieve unit loss in expectation.

For a single scalar feature and a given timestep $t$, we can write the empirical denoising score matching loss (Equation (3)) when using the EDM parameterization (Karras et al., 2022) as:

$$\mathcal{L}_{\text{MSE}}(x_{\text{cont}}^{(i)}, t) = \lambda(t)\Big(\underbrace{c_{\text{skip}}(t)x_t + c_{\text{out}}(t)F_{\boldsymbol{\theta}}^{(i)}}_{\hat{x}_{\boldsymbol{\theta}}(x_t, t)} - x_{\text{cont}}^{(i)}\Big)^2,$$

where $F_{\boldsymbol{\theta}}^{(i)}$ denotes the neural network output for feature $i$ that parameterizes the denoiser $\hat{x}_{\boldsymbol{\theta}}$. The score model is then given by $s_{\boldsymbol{\theta}}(x_t, t) = \sigma(t)^{-2}(\hat{x}_{\boldsymbol{\theta}}(x_t, t) - x_t)$.

The parameters $c_{\text{skip}}(t) = \sigma_{\text{data}}^2/(\sigma^2(t) + \sigma_{\text{data}}^2)$ and $c_{\text{out}}(t) = \sigma(t) \cdot \sigma_{\text{data}}/(\sqrt{\sigma^2(t) + \sigma_{\text{data}}^2})$ depend on $\sigma(t)$ (and $\sigma_{\text{data}}$) and therefore on timestep $t$. For $t \to 1$, $\sigma(t)$ approaches the maximum noise level $\sigma_{\text{cont,max}}$ and $c_{\text{skip}}(t) \to 0$ and $c_{\text{out}}(t) \to 1$ such that the score model directly predicts the data at high noise levels. For $t \to 0$, the model shifts increasingly towards predicting the error that has been added to the true data. In the EDM parameterization, the explicit timestep weight (used to achieve a unit loss across timesteps at initialization, see Appendix C) is $\lambda(t) = 1/c_{\text{out}}(t)^2 \approx 1$ for $t = 1$.

At the terminal timestep $t = 1$, we then have:

$$\mathbb{E}_{p(x_{\text{cont}}^{(i)})}[\mathcal{L}_{\text{MSE}}(x_{\text{cont}}^{(i)}, 1)] = \lambda(1) \, \mathbb{E}_{p(x_{\text{cont}}^{(i)})}\left(c_{\text{skip}}(1)x_1 + c_{\text{out}}(1)F_{\boldsymbol{\theta}}^{(i)} - x_{\text{cont}}^{(i)}\right)^2,$$

$$\approx \mathbb{E}_{p(x_{\text{cont}}^{(i)})}\left(0 \cdot x_1 + 1 \cdot F_{\boldsymbol{\theta}}^{(i)} - x_{\text{cont}}^{(i)}\right)^2,$$

$$= \mathbb{E}_{p(x_{\text{cont}}^{(i)})}\left(F_{\boldsymbol{\theta}}^{(i)} - x_{\text{cont}}^{(i)}\right)^2.$$

Without information, it is optimal to always predict the average value $\mathbb{E}_{p(x_{\text{cont}}^{(i)})}[x_{\text{cont}}^{(i)}]$ and thus, the minimum expected loss becomes:

$$\mathbb{E}_{p(x_{\text{cont}}^{(i)})}[\mathcal{L}_{\text{MSE}}(x_{\text{cont}}^{(i)}, 1)] = \mathbb{E}_{p(x_{\text{cont}}^{(i)})}\left(\mathbb{E}_{p(x_{\text{cont}}^{(i)})}[x_{\text{cont}}^{(i)}] - x_{\text{cont}}^{(i)}\right)^2 = \text{Var}[x_{\text{cont}}^{(i)}] \,.$$

Therefore, we have $\mathcal{L}_{\text{MSE}}^*(x_{\text{cont}}^{(i)}, 1) = \mathcal{L}_{\text{MSE}}(x_{\text{cont}}^{(i)}, 1)$ as long as we standardize $x_{\text{cont}}^{(i)}$ to unit variance.

For a single categorical feature, $x_{\text{cat}}^{(j)}$ is distributed according to the proportions $p_c$ (for categories $c = 1, \ldots, C_j$). The denoising model for score interpolation is trained with the CE loss:

$$\mathcal{L}_{\text{CE}}(x_{\text{cat}}^{(j)}, t) = - \sum_{c=1}^{C_j} I(x_{\text{cat}}^{(j)} = c) \log F_{\boldsymbol{\theta}, c}^{(j)} \,,$$

where $F_{\boldsymbol{\theta}, c}^{(j)}$ denotes the score model's prediction of the class probability at timestep $t$. Without information, it is optimal to assign the $c$-th category the same proportion as in the training set. At $t = 1$, we thus let $F_{\boldsymbol{\theta}, c}^{(j)} = p_c$ such that the loss equals:

$$\mathbb{E}_{p(x_{\text{cat}}^{(j)})}[\mathcal{L}_{\text{CE}}(x_{\text{cat}}^{(j)}, 1)] = -\mathbb{E}_{p(x_{\text{cat}}^{(j)})} \sum_{c=1}^{C_j} I(x_{\text{cat}}^{(j)} = c) \log F_{\boldsymbol{\theta}, c}^{(j)} \,, \tag{8}$$

$$= -\sum_{c=1}^{C_j} \mathbb{E}_{p(x_{\text{cat}}^{(j)})}[I(x_{\text{cat}}^{(j)} = c) \log p_c] \,, \tag{9}$$

$$= -\sum_{c=1}^{C_j} p_c \log p_c. \tag{10}$$

Note that this is the feature-specific entropy and we can use the training set proportions to compute this normalization constant $Z_j = -\sum_{c=1}^{C_j} p_c \log p_c$ to scale the loss for categorical features. Then,

$$\mathbb{E}_{p(x_{\text{cat}}^{(j)})}[\mathcal{L}_{\text{CE}}^*(x_{\text{cat}}^{(j)}, 1)] = \mathbb{E}_{p(x_{\text{cat}}^{(j)})}[\mathcal{L}_{\text{CE}}(x_{\text{cat}}^{(j)}, 1)/Z_j] = 1 \,.$$

We have thus achieved calibrated losses with respect to the terminal timestep $t = 1$, that is, $\mathbb{E}_{p(x_{\text{cont}}^{(i)})}[\mathcal{L}_{\text{MSE}}^*(x_{\text{cont}}^{(i)}, 1)] = \mathbb{E}_{p(x_{\text{cat}}^{(j)})}[\mathcal{L}_{\text{CE}}^*(x_{\text{cat}}^{(j)}, 1)] = 1$ for all continuous features $i$ and categorical features $j$.

## C  OUTPUT LAYER INITIALIZATION

At initialization, we want the neural network to reflect the state of no information (see Appendix B). Likewise, our goal is a loss of one across all features and timesteps.

For continuous features $i$, we initialize the output layer weights (and biases) to zero such that the output of the score model for a single continuous feature, $F_{\boldsymbol{\theta}}^{(i)}$, is also zero. Since we use the EDM parameterization (Karras et al., 2022), we apply the associated explicit timestep weight $\lambda(t) = \frac{\sigma^2(t) + \sigma_{\text{data}}^2}{(\sigma(t) \cdot \sigma_{\text{data}})^2}$. This is explicitly designed to achieve a unit loss across timesteps at initialization and we show this analytically below. We denote the variances of the data $x_{\text{cont}}^{(i)}$ and of the Gaussian noise $\epsilon$ at time $t$ as $\sigma_{\text{data}}^2$ and $\sigma^2(t)$, respectively. At initialization we have:

$$
\begin{aligned}
\mathbb{E}_{p(x_{\text{cont}}^{(i)}), p(\epsilon)}[\mathcal{L}_{\text{MSE}}^*(x_{\text{cont}}^{(i)}, t)] &= \lambda(t) \, \mathbb{E}_{p(x_{\text{cont}}^{(i)}), p(\epsilon)} \Big( c_{\text{skip}}(t)(x_{\text{cont}}^{(i)} + \epsilon) + c_{\text{out}}(t) F_{\boldsymbol{\theta}}^{(i)} - x_{\text{cont}}^{(i)} \Big)^2 , \\
&= \lambda(t) \, \mathbb{E}_{p(x_{\text{cont}}^{(i)}), p(\epsilon)} \Big( c_{\text{skip}}(t)(x_{\text{cont}}^{(i)} + \epsilon) - x_{\text{cont}}^{(i)} \Big)^2 , \\
&= \frac{\sigma^2(t) + \sigma_{\text{data}}^2}{(\sigma(t) \cdot \sigma_{\text{data}})^2} \mathbb{E}_{p(x_{\text{cont}}^{(i)}), p(\epsilon)} \Big( \frac{\sigma_{\text{data}}^2}{\sigma^2(t) + \sigma_{\text{data}}^2}(x_{\text{cont}}^{(i)} + \epsilon) - x_{\text{cont}}^{(i)} \Big)^2 , \\
&= \frac{\sigma^2(t) + \sigma_{\text{data}}^2}{(\sigma(t) \cdot \sigma_{\text{data}})^2} \mathbb{E}_{p(x_{\text{cont}}^{(i)}), p(\epsilon)} \Big( \frac{\sigma_{\text{data}}^2 \epsilon - \sigma^2(t) x_{\text{cont}}^{(i)}}{\sigma^2(t) + \sigma_{\text{data}}^2} \Big)^2 , \\
&= \frac{1}{\sigma^2(t) + \sigma_{\text{data}}^2} \mathbb{E}_{p(x_{\text{cont}}^{(i)}), p(\epsilon)} \Big( \frac{\sigma_{\text{data}}}{\sigma(t)} \epsilon - \frac{\sigma(t)}{\sigma_{\text{data}}} x_{\text{cont}}^{(i)} \Big)^2 , \\
&= \frac{1}{\sigma^2(t) + \sigma_{\text{data}}^2} \mathbb{E}_{p(x_{\text{cont}}^{(i)}), p(\epsilon)} \Big( \frac{\sigma_{\text{data}}^2}{\sigma^2(t)} \epsilon^2 + \frac{\sigma^2(t)}{\sigma_{\text{data}}^2}(x_{\text{cont}}^{(i)})^2 - 2\epsilon x_{\text{cont}}^{(i)} \Big) , \\
&= \frac{1}{\sigma^2(t) + \sigma_{\text{data}}^2} \Big( \frac{\sigma_{\text{data}}^2}{\sigma^2(t)} \underbrace{\text{Var}(\epsilon)}_{\sigma^2(t)} + \frac{\sigma^2(t)}{\sigma_{\text{data}}^2} \underbrace{\text{Var}(x_{\text{cont}}^{(i)})}_{\sigma_{\text{data}}^2} - 2 \underbrace{\text{Cov}(\epsilon, x_{\text{cont}}^{(i)})}_{0} \Big) , \\
&= \frac{1}{\sigma^2(t) + \sigma_{\text{data}}^2} \Big( \sigma_{\text{data}}^2 + \sigma^2(t) \Big) = 1.
\end{aligned}
$$

For categorical features $j$, we initialize the output layer such that the model achieves the respective losses under no information. Using the loss normalization constant $Z_j$ (see Appendix B) and dropping the expectation over $p(\epsilon)$, we have

$$
\mathbb{E}_{p(x_{\text{cat}}^{(j)})}[\mathcal{L}_{\text{CE}}^*(x_{\text{cat}}^{(j)}, t)] = \mathbb{E}_{p(x_{\text{cat}}^{(j)})}[\mathcal{L}_{\text{CE}}(x_{\text{cat}}^{(j)}, t)/Z_j] = \frac{1}{Z_j} \mathbb{E}_{p(x_{\text{cat}}^{(j)})}[\mathcal{L}_{\text{CE}}(x_{\text{cat}}^{(j)}, t)].
$$

Hence, for $E_{p(x_{\text{cat}}^{(j)})}[\mathcal{L}_{\text{CE}}(x_{\text{cat}}^{(j)}, t)] = Z_j$, we obtain an expected loss of one irrespective of $t$. The neural network outputs a vector of logits $F_{\boldsymbol{\theta}}^{(j)}$ that are transformed into probabilities with a softmax function for each categorical feature. We denote the $c$-th element of that vector $\text{softmax}(\cdot)_c$. Since $Z_j$ is derived in Equation (10) by imposing probabilities equal to the training set proportions for that category, $p_c$, we have

$$
\log p_c = \log \text{softmax}(F_{\boldsymbol{\theta}}^{(j)})_c = \log \frac{\exp(F_{\boldsymbol{\theta},c}^{(j)})}{\sum_{k=1}^{C} \exp(F_{\boldsymbol{\theta},k}^{(j)})} = F_{\boldsymbol{\theta},c}^{(j)} - \log \sum_{k=1}^{C} \exp(F_{\boldsymbol{\theta},k}^{(j)}).
$$

We initialize the neural network such that $F_{\boldsymbol{\theta},c}^{(j)} = \log p_c$ for all $c$. This is achieved by initializing the output layer weights to zero and the output layer biases to the relevant training set log-proportions of the corresponding class. Hence, this initialization gives us

$$
F_{\boldsymbol{\theta},c}^{(j)} - \log \sum_{k=1}^{C} \exp(F_{\boldsymbol{\theta},k}^{(j)}) = \log p_c - \log \sum_{k=1}^{C} p_k = \log p_c,
$$

which in turn leads to an initial loss of $Z_j$ for all $t$ and therefore achieves a uniform, calibrated loss of one at initialization similar to the continuous feature case.

# D  ADAPTIVE NORMALIZATION OF THE AVERAGE DIFFUSION LOSS

Both the loss calibration (see Appendix B) and output layer initialization (see Appendix C) ensure that the losses across timesteps (and features) are equal *at* initialization. During training, the adaptive noise schedules allow the model to focus automatically on the noise levels that matter most, i.e., where the loss increase is steepest. However, the better the model becomes at a given timestep $t$, the lower the loss at the respective timestep, and the lower the gradient signal relative to the signal for timesteps $\tilde{t} > t$. We counteract this with adaptive normalization of the average diffusion loss (averaged over the features) across timesteps. Specifically, we want to weight the average diffusion loss at timestep $t$, $\mathcal{L}(t)$ given in Equation (5), such that the normalized loss is the same (equal to one) for all $t$. Similar methods have been used by Karras et al. (2024) and Kingma & Gao (2023), we follow the latter in the setup of the corresponding network.

We train a neural network alongside our diffusion model to predict $\mathcal{L}(t)$ based on $t$ and use the MSE loss to learn this weighting. First, we compute $c_{\text{noise}}(t) = \log(t)/4$ following the EDM parameterization (Karras et al., 2022). Then, we embed $c_{\text{noise}}$ in frequency space (1024-dimensional) using Fourier features. The result is passed through a single linear layer to output a scalar value and through an exponential function to ensure that the prediction $\hat{\mathcal{L}}(t) \geq 0$. We initialize the weights and biases to zero, to ensure that at model initialization we have a unit normalization.

# E  DERIVATION OF THE FUNCTIONAL TIMEWARPING FORM

Since higher noise levels, $\sigma$, imply a lower signal-to-noise ratio, and in turn a larger loss, $\ell$, we know that the loss must be a monotonically increasing and S-shaped function of the noise level. Additionally, the function has to be easy to invert and differentiate. We incorporate this prior information in the functional timewarping form of $F : \sigma \mapsto \ell$. A convenient choice is the cdf of the logistic distribution:

$$F_{\log}(y) = \left[1 + \exp\left(-\nu(y - \mu^*)\right)\right]^{-1}, \tag{11}$$

where $\mu^*$ describes the location of the inflection point of the S-shaped function and $\nu \geq 1$ indicates the steepness of the curve.

We let $y = \text{logit}(\sigma) = \log(\sigma/(1 - \sigma))$ to change the domain of $F_{\log}$ from $(-\infty, \infty)$ to $(0, 1)$. The latter covers all possible values of the noise level $\sigma$ scaled to $[0, 1]$ with the pre-specified minimum and maximum noise levels $\sigma_{\min}$ and $\sigma_{\max}$. To define the parameter $\mu$ in the same space and ensure that $0 < \mu < 1$, we also let $\mu^* = \text{logit}(\mu)$. Accordingly, we derive the cdf of the *domain-adapted* Logistic distribution:

$$F_{\text{d.a.log}}(\sigma) = \left[1 + \left(\frac{\sigma}{1 - \sigma}\frac{1 - \mu}{\mu}\right)^{-\nu}\right]^{-1}. \tag{12}$$

Since $\ell$ is not bounded, we introduce a multiplicative scale parameter, $\gamma > 0$, such that for timewarping we predict the potentially feature-specific loss as $\hat{\ell} = F(\sigma) = \gamma F_{\text{d.a.log}}(\sigma)$. $F_{\text{d.a.log}}$ can also be initialized to the cdf of the uniform distribution with $\mu = 0.5$, $\nu \approx 1$ and $\gamma = 1$ such that all noise levels are initially equally weighted. However, an initial overweighting of lower noise levels is beneficial for tabular data (see also Section 3.4).

Likewise, we can derive the inverse cdf $F_{\text{d.a.log}}^{-1}(t)$, that is our mapping of interest from timestep $t$ to noise level $\sigma$, in closed form:

$$\sigma = F_{\text{d.a.log}}^{-1}(t) = \text{sigmoid}(c), \text{ with } c = \ln\left(\frac{\mu}{1 - \mu}\right) + \frac{1}{\nu}\ln\left(\frac{t}{1 - t}\right). \tag{13}$$

When training the diffusion model, we learn the parameters of $F_{\text{d.a.log}}$ as well as $\gamma$ by predicting the diffusion loss using $F(\sigma)$ and the noise levels scaled to $[0, 1]$. At the beginning of each training step, we then use the current state of the parameters and $F_{\text{d.a.log}}^{-1}$, with a sampled timestep $t \sim \mathcal{U}_{[0,1]}$ as input, to derive $\sigma$. To allow for *feature-specific*, adaptive noise schedules, we separately introduce $F_k(\sigma_k)$ for each feature $k$, to predict the feature-specific loss $\ell_k$ based on the feature-specific scaled noise level $\sigma_k$.

Note that with timewarping we create a feedback loop in which we generate more and more $\sigma$s from the region of interest, decreasing the number of observations available to learn the parameters in

different noise level regions. We thus weight the timewarping loss, $||\ell - \hat{\ell}||_2^2$, when fitting $F(\sigma)$ to the data by the reciprocal of the pdf $f_{d.a.log}(\sigma)$ to mitigate this adverse effect (see Dieleman et al., 2022). Again, this function is available to us in closed form. With $F_{\log}$ and $f_{\log}$ denoting the respective cdf and pdf of the Logistic distribution, we have

$$
\begin{aligned}
f_{d.a.log}(\sigma) &= \left. \frac{\partial}{\partial y} F_{\log}(y) \right|_{y=\text{logit}(\sigma)} \frac{\partial}{\partial \sigma} \ln \frac{\sigma}{1-\sigma} \\
&= f_{\log}(\text{logit}(\sigma)) \frac{1}{\sigma(1-\sigma)} \\
&= \frac{\nu}{\sigma(1-\sigma)} \cdot \frac{Z(\sigma, \mu, \nu)}{\left(1 + Z(\sigma, \mu, \nu)\right)^2},
\end{aligned}
$$

where we defined $Z(\sigma, \mu, \nu) = \left( \frac{\sigma}{1-\sigma} \frac{1-\mu}{\mu} \right)^{-\nu}$ and used the definitions of $f_{\log}$ and the parameter $\mu^*$.

## F    BENCHMARK DATASETS

Our selected benchmark datasets are highly diverse, particularly in the number of categories for categorical features (see Table 3). For the `diabetes` and `covertype` datasets, we transform the original multi-class classification problem into a binary classification task for ease of presentation. For the `diabetes` data, we convert the task to predicting whether a patient was readmitted to a hospital. For the `covertype` data, the task is converted into predicting whether a forest of type 2 is present in a given $30 \times 30$ meter area. All datasets are publicly accessible and (except `nmes`) licensed under creative commons.

Table 3: Overview of the selected experimental datasets. We count the target towards the respective features that remain after removing continuous features with an excessive number of missings. The minimum and maximum number of categories are taken over all categorical features.

| Dataset | License | Prediction task | Total no. observations | No. of features categorical | continuous | No. of categories min | max |
|---|---|---|---|---|---|---|---|
| acsincome (Ding et al., 2021) | CC0 | regression | 1 664 500 | 8 | 3 | 2 | 529 |
| adult (Becker & Kohavi, 1996) | CC BY 4.0 | binary class. | 48 842 | 9 | 6 | 2 | 42 |
| bank (Moro et al., 2014) | CC BY 4.0 | binary class. | 41 188 | 11 | 10 | 2 | 12 |
| beijing (Chen, 2015) | CC BY 4.0 | regression | 41 757 | 1 | 10 | 4 | 4 |
| churn (Keramati et al., 2014) | CC BY 4.0 | binary class. | 3 150 | 5 | 9 | 2 | 5 |
| covertype (Blackard, 1998) | CC BY 4.0 | binary class. | 581 012 | 44 | 10 | 2 | 2 |
| default (Yeh, 2009) | CC BY 4.0 | binary class. | 30 000 | 10 | 14 | 2 | 11 |
| diabetes (Clore et al., 2014) | CC BY 4.0 | binary class. | 101 766 | 28 | 9 | 2 | 716 |
| lending (Lending Club, 2015) | DbCL 1.0 | regression | 9 182 | 10 | 34 | 2 | 3151 |
| news (Fernandes et al., 2015) | CC BY 4.0 | regression | 39 644 | 14 | 46 | 2 | 2 |
| nmes (Deb & Trivedi, 1997) | unknown | regression | 4 406 | 8 | 11 | 2 | 4 |

## G    BASELINE MODELS

Below, we give a brief description of our selected generative baseline models (including code sources).

**SMOTE** (Chawla et al., 2002) – a technique (not a generative model) typically used to oversample minority classes based on interpolation between ground-truth observations. We use SMOTENC for mixed-type data from the scikit-learn package and mostly adapt the code from the TabDDPM repository (Kotelnikov et al., 2023). For sampling, we utilize 16 CPU cores.

**ARF** (Watson et al., 2023) – a recent generative approach that is based on a random forest for density estimation. The implementation is available at `https://github.com/bips-hb/arfpy` and licensed under the MIT license. We use package version 0.1.1. For training, we utilize 16 CPU cores.

**CTGAN** (Xu et al., 2019) – one of the most popular Generative-Adversarial-Network-based models for tabular data. The implementation is available as part of the Synthetic Data Vault (Patki et al., 2016) at `https://github.com/sdv-dev/CTGAN` and licensed under the Business Source License 1.1. We use package version 0.9.0.

**TVAE** (Xu et al., 2019) – a Variational-Autoencoder-based model for tabular data. Similar to CTGAN. The implementation is available as part of the Synthetic Data Vault (Patki et al., 2016) at `https://github.com/sdv-dev/CTGAN` and licensed under the Business Source License 1.1. We use package version 0.9.0. Note that since we only use TVAE (and CTGAN) as benchmark, and do not provide a synthetic data creation service, the license permits the free usage.

**TabDDPM** (Kotelnikov et al., 2023) – a diffusion-based generative model for tabular data that combines multinomial diffusion (Hoogeboom et al., 2021) and diffusion in continuous space. An implementation is available as part of the `synthcity` package (Qian et al., 2023) at `https://github.com/vanderschaarlab/synthcity/` and licensed under the Apache 2.0 license. We use package version 0.2.7 with slightly adjusted code to allow for the manual specification of categorical features.

**CoDi** (Lee et al., 2023) – a diffusion model trained with an additional contrastive loss, and which factorizes the joint distribution of mixed-type tabular data into a distribution for continuous data conditional on categorical features and a distribution for categorical data conditional on continuous features. Similarly, the authors utilize the multinomial diffusion framework (Hoogeboom et al., 2021) to model categorical data. An implementation is available at `https://github.com/ChaejeongLee/CoDi` under an unknown license.

**TabSyn** (Zhang et al., 2024) – a diffusion-based model that first learns a transformer-based VAE to map mixed-type data to a continuous latent space. Then, the diffusion model is trained on that latent space. Note that despite TabSyn utilizing a separately trained encoder, this does *not* result in a lower-dimensional latent space and therefore, does not speed up sampling. We use the official code available at `https://github.com/amazon-science/tabsyn` under the Apache 2.0 license.

## H  IMPLEMENTATION DETAILS

Each of the selected benchmark models requires a rather different, more specialized neural network architecture. Imposing the same architecture across models is therefore not possible. The same inability holds for the comparison of CDTD to other diffusion-based models: Our model is the first to use a continuous noise distribution on both continuous and categorical features, and therefore the alignment of important design choices, like the noise schedule, across models is not possible. In particular, the forward process of the multinomial diffusion framework (Hoogeboom et al., 2021) used in TabDDPM and CoDi, which is based on Markov transition matrices, does not translate to our setting.

To ensure a fair comparison in terms of sampling steps, we set the steps for CDTD, TabDDPM, CoDi and TabSyn to $\max(200, \text{default})$. We therefore increase the default number of sampling steps for CoDi and TabSyn (from 50 steps) and TabDDPM (from 100 steps for classification datasets). For TabDDPM and regression datasets, we use the suggested default of 1000 sampling steps.

We adjust each architecture to a total of $\sim 3$ million trainable parameters on the `adult` dataset to improve the comparability further (see Table 4) and use the same architectures for all considered datasets. Note that the total number of parameters may vary slightly across datasets due to different number of features and categories affecting the one-hot encoding but is still comparable across models. We also align the embedding/bottleneck dimensions for CTGAN, TVAE, TabDDPM, TabSyn and CDTD to 256. To align TabDDPM, TabSyn and CDTD further, we use the TabDDPM architecture for all models, with appropriate adjustments for different input types and dimensions. If applicable, all models are trained for 30k steps on a single RTX 4090 instance, using PyTorch version 2.2.2.

Below, we briefly discuss our model-specific hyperparameter choices.

Table 4: Total number of trainable parameters per model on the `adult` dataset.

| Model | Trainable parameters |
|---|---|
| CTGAN | 3 000 397 |
| TVAE | 2 996 408 |
| TabDDPM | 3 001 786 |
| CoDi | 2 998 043 |
| TabSyn | 2 997 765 |
| CDTD (per type) | 3 002 969 |

**SMOTE** (Chawla et al., 2002): We use the default hyperparameters suggested for the SMOTENC scikit-learn implementation.

**ARF** (Watson et al., 2023): We use the authors's suggested default hyperparameters. In particular, we use 20 trees, $\delta = 0$ and a minimum node size of 5. We follow the official package implementation and set the maximum number of iterations to 10 (see `https://github.com/bips-hb/arfpy`).

**CTGAN** (Xu et al., 2019): We follow the popular implementation in the Synthetic Data Vault package (see `https://github.com/sdv-dev/CTGAN`). For this model to work, the batch size must be divisible by 10. Therefore, we adjust the batch size if necessary. We use a 256-dimensional embedding (instead of the default embedding dimension of 128) to better align the CTGAN architecture with TVAE, TabDDPM, TabSyn and CDTD.

**TVAE** (Xu et al., 2019): We again follow the implementation in the Synthetic Data Vault. We use a 256-dimensional embedding to better align the architecture with CTGAN, TabDDPM, TabSyn and CDTD.

**TabDDPM** (Kotelnikov et al., 2023): There are no general default hyperparameters provided. Hence, we mostly adapt the papers' tuned hyperparameters for the `adult` dataset (one of the few used datasets that includes both continuous and categorical features). However, we decrease the learning rate from 0.002 to 0.001, since most of the tuned models in the paper used learning rates around 0.001. For regression task datasets, we use 1000 sampling steps in accordance with the author's settings. For classification task datasets, we use 200 sampling steps (instead of the default 100 steps), to better align the model with CoDi and CDCD. Note also that for classification task datasets, TabDDPM models the conditional distribution $p(x|y)$, instead of the unconditional distribution $p(x)$ which is modeled for regression tasks. We adjust the dimension of the bottleneck to 256 (instead of the default 128) to also accommodate also larger datasets and align the model with CTGAN, TVAE, and CDTD.

**CoDi** (Lee et al., 2023): We use the default hyperparameters from the official code (see `https://github.com/ChaejeongLee/CoDi`).

**TabSyn** (Zhang et al., 2024): We use the default hyperparameters as suggested by the authors. The training steps that go towards training the VAE and the denoising network follow the proportions given in the official code (see `https://github.com/amazon-science/tabsyn`). To improve comparability to TabDDPM, CoDi and CDTD, we use the same neural network architecture as TabDDPM, which only differs slightly from the original architecture. We leave the VAE untouched.

**CDTD** (ours): To ensure comparability in particular to TabDDPM, CoDi and TabSyn, we use the same neural network architecture as TabDDPM. We only change the input layers to accommodate our embedding-based framework. In the input layer, we vectorize all embedded categorical features and concatenate them with the scalar valued continuous features. The adjusted output layer ensures that we predict a single value for each continuous features and set of class-specific probabilities for each categorical feature. Since our use of embeddings introduces additional parameters, we scale the hidden layers slightly down relative to the TabDDPM to ensure approximately 3 million trainable parameters (instead of 798 neurons per layer we use 796) on the `adult` dataset. More details on the CDTD implementation are given in Appendix K.

## I  TUNING OF THE DETECTION MODEL

We use a catboost model (Prokhorenkova et al., 2018) to test whether real and generated samples can be distinguished. We generate the same number of fake observations for each of the real train, validation and test sets. We cap the maximum size of the real data subsets to 25 000, and subsample them if necessary, to limit the computational load. Per set, we combine real and fake observations to $\mathcal{D}_{\text{train}}^{\text{detect}}, \mathcal{D}_{\text{valid}}^{\text{detect}}$, and $\mathcal{D}_{\text{test}}^{\text{detect}}$, respectively. The catboost model is trained on $\mathcal{D}_{\text{train}}^{\text{detect}}$ with the task of predicting whether an observation is real or fake. We tune the catboost model with optuna and for 50 trials to maximize the accuracy on $\mathcal{D}_{\text{valid}}^{\text{detect}}$. The catboost hyperparameter search space is given in Table 5. Afterwards, we repeat the sampling process and the creation of $\mathcal{D}_{\text{train}}^{\text{detect}}, \mathcal{D}_{\text{valid}}^{\text{detect}}$ and $\mathcal{D}_{\text{test}}^{\text{detect}}$ for five different seeds. Each time, the model is trained on $\mathcal{D}_{\text{train}}^{\text{detect}}$ with the previously tuned hyperparameters, and evaluated on $\mathcal{D}_{\text{test}}^{\text{detect}}$. The average test set accuracy over the five seeds yields the estimated detection score.

Table 5: Catboost hyperparameter space settings. The model is tuned for 50 trials.

| Parameter | Distribution |
|---|---|
| no. iterations | = 1000 |
| learning rate | Log Uniform [0.001, 1.0] |
| depth | Cat([3,4,5,6,7,8]) |
| $L_2$ regularization | Uniform [0.1, 10] |
| bagging temperature | Uniform [0, 1] |
| leaf estimation iters | Integer Uniform [1, 10] |

## J   MACHINE LEARNING EFFICIENCY MODELS

For the group of machine learning efficiency models, we use the scikit-learn and catboost package implementations including the default parameter settings, if not specified otherwise below:

**Logistic or Ridge Regression:** max. iterations = 1000

**Random Forest:** max. depth = 12, no. estimators = 100

**Catboost:** no. iterations = 2000, early stopping rounds = 50, overfitting detector pval = 0.001

We subsample $\mathcal{D}_{\text{train}}$ in case of more than 50 000 observations to upper-bound the computational load.

## K   CDTD IMPLEMENTATION DETAILS

To enable a fair comparison to the other methods, and to TabDDPM and TabSyn in particular, the CDTD score model utilizes the exact same architecture as Kotelnikov et al. (2023), which was also adapted by TabSyn (Zhang et al., 2024). We use the QuantileTransformer to pre-process continuous features, followed by a standardization to zero mean and unit variance. An overview of the score model is provided in Figure 5: First, the noisy data, i.e., the noisy scalars for continuous features and the noisy embeddings for categorical features, and the timestep $t$, are projected onto a 256-dimensional space. Then, all 256-dimensional vectors are added and the result is processed by a

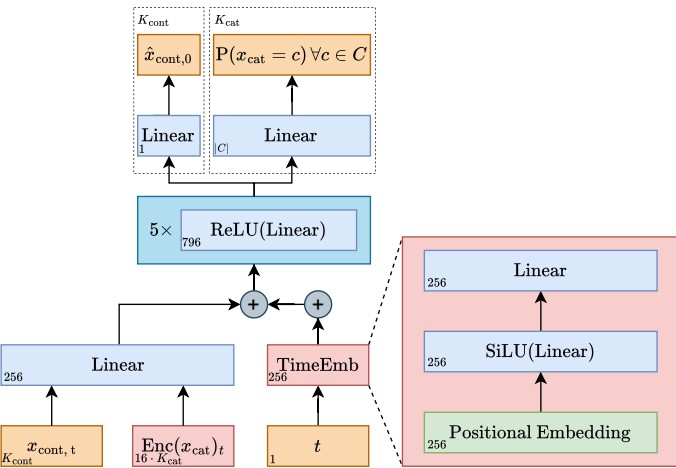

Figure 5: Overview of the CDTD architecture adapted from TabDDPM. The dimensions of the inputs and layer outputs are stated in the lower-left hand corner for a continuous features $x_{\text{cont}}$ and a categorical features $x_{\text{cat}}$. Note that each categorical features can have a different number of categories $|C|$, impacting the output dimension of the final layer. Scalars are colored orange, embeddings red and linear layers blue. The positional embedding highlighted in green refers to the positional sinusoidal embedding. CDTD only conditions on $y$, i.e., the target feature, for classification task datasets.

set of five fully-connected linear layers with ReLU activation functions. Lastly, a linear projection maps the output of the fully-connected layers to the required output dimensions, which depend on the number of features and number of categories per feature.

The only major difference to the TabDDPM setup are the inputs, as we need to embed the categorical features in Euclidean space. The output dimensions are the same, as we need to predict a single scalar for each $x_{\text{cont}}^{(i)}$, and $C_j$ values for each $x_{\text{cat}}^{(j)}$, with $C_j$ being the number of categories of feature $j$. We change the initialization of the output layer as described in Appendix C. To handle our inputs, we embed the categorical features in 16-dimensional space and add a feature-specific bias of the same dimension, which captures feature-specific information common to all categories and is initialized to zero. We $L_2$-normalize each embedding to prevent a degenerate embedding space in which embeddings are pushed further and further apart (see Dieleman et al., 2022). Also, Dieleman et al. (2022) argue that the standard deviation of the Normal distribution used to initialize the embeddings, denoted by $\sigma_{\text{init}}$, is an important hyperparameter. In this paper, we set $\sigma_{\text{init}} = 0.001$ for all datasets and have not seen detrimental effects. Table 7 indicates that CDTD is not sensitive to the choice of $\sigma_{\text{init}}$.

Since we utilize embeddings, we have to scale the neurons per layer slightly down in the stack of the five fully-connected layers (from 798 for TabDDPM to 796). Also, since TabDDPM samples discrete steps from $[0, T]$, with $T \gg 1$, we scale our timesteps $t \in [0, 1]$ up by 1000. We use the same optimizer (Adam), learning rate (0.001), learning rate decay (linear), EMA decay (0.999), and training steps (30000). However, since we work with embeddings we add a linear warmup schedule over the first 1000 steps. Lastly, instead of using uniform (time)step sampling as TabDDPM, the CDTD model uses antithetic sampling (Dieleman et al., 2022; Kingma et al., 2021). The timesteps are still uniformly distributed but spread out more evenly over the domain, which benefits the training of the adaptive noise schedules. For generation, we use a deterministic (Euler) sampler with 200 steps to minimize the discretization error (see Appendix L for details).

## L   CDTD SAMPLING

Algorithm 1 shows our deterministic sampling approach, where $D_{\boldsymbol{\theta}}^{\text{cont},i}$ and $D_{\boldsymbol{\theta}}^{\text{cat},j}$ represent the score model output for the $i$-th continuous and $j$-th categorical feature, respectively. We found results similar to Karras et al. (2022), i.e., that adding stochasticity to the sampling process does not benefit sample quality. We also experimented with second-order samplers (e.g., Heun) but found them to add little benefit over our first-order method while requiring double the NFEs. As a default, we use 200 sampling steps. Table 6 shows that the gains in sample quality are marginal to non-existent after more than 500 sampling steps.

There is another subtlety in our sampler: Unlike in EDM (Karras et al., 2022), in the final step the sampler steps into $t = 0$, which is not associated with $\sigma_0 = 0$ but $\sigma_0 = \sigma_{\text{min}}$. This is a consequence of utilizing more than one noise schedule. We condition the score model not on $\sigma$ but on $t$, which indicates the "global" time common to all noise schedules. Moving from $\sigma_0 = \sigma_{\text{min}}$ to $\sigma = 0$ in the final step, would imply $t < 0$. Therefore, we let $\sigma_{\text{min}} = 0$ for all feature types.

To sample from the learned distribution, we need to run the reverse process of the probability flow ODE (Equation (2)). For example, for two different features $x_1$ and $x_2$, we deconstruct the ODE as:

$$
\begin{aligned}
\mathrm{d}\mathbf{x} &= -\frac{1}{2}\mathbf{G}(t)\mathbf{G}(t)^{\mathsf{T}}\nabla_{\mathbf{x}}\log p_t(\mathbf{x})\mathrm{d}t \\
&= -\begin{bmatrix} \dot{\sigma}_1(t)\sigma_1(t) & \\ & \dot{\sigma}_2(t)\sigma_2(t) \end{bmatrix}\begin{bmatrix} \frac{\hat{x_1}-x_1}{\sigma_1(t)^2} \\ \frac{\hat{x_2}-x_2}{\sigma_2(t)^2} \end{bmatrix}\mathrm{d}t \\
&= -\begin{bmatrix} \dot{\sigma}_1(t) & \\ & \dot{\sigma}_2(t) \end{bmatrix}\begin{bmatrix} \frac{\hat{x_1}-x_1}{\sigma_1(t)} \\ \frac{\hat{x_2}-x_2}{\sigma_2(t)} \end{bmatrix}\mathrm{d}t
\end{aligned}
$$

In practice, we use an Euler sampler with 200 discrete timesteps $\Delta t = t_{i+1} - t_i < 0$. The timesteps are generated as a linearly spaced grid on $[0, 1]$ and transformed afterwards into noise levels $\sigma_k(t)$ via the described timewarping procedure. For the discretized and simplified ODE at step $i$, this yields

$$
\mathbf{x}_{i+1} = \mathbf{x}_i - \begin{bmatrix} \frac{\Delta\sigma_1(t)}{\Delta t} & \\ & \frac{\Delta\sigma_2(t)}{\Delta t} \end{bmatrix}\begin{bmatrix} \frac{\hat{x_1}-x_1}{\sigma_1(t_i)} \\ \frac{\hat{x_2}-x_2}{\sigma_2(t_i)} \end{bmatrix}\Delta t = \mathbf{x}_i + \begin{bmatrix} \frac{x_1-\hat{x_1}}{\sigma_1(t_i)} \\ \frac{x_2-\hat{x_2}}{\sigma_2(t_i)} \end{bmatrix} \odot \begin{bmatrix} \Delta\sigma_1(t) \\ \Delta\sigma_2(t) \end{bmatrix},
$$

---

**Algorithm 1** Deterministic Sampling

---

**Input:** $N$ sampling steps

   Sample $\mathbf{x}_{\text{cont},t_0}^{(i)} \sim \mathcal{N}(\mathbf{0}, \sigma_{\text{cont,max}}^2 \mathbf{I}_{K_{\text{cont}}}) \,\forall i \in \{1, \ldots, K_{\text{cont}}\}$

   Sample $\mathbf{x}_{\text{cat},t_0}^{(j)} \sim \mathcal{N}(\mathbf{0}, \sigma_{\text{cat,max}}^2 \mathbf{I}_{K_{\text{cat}}}) \,\forall j \in \{1, \ldots, K_{\text{cat}}\}$

1: **for** $s \in \{0, \ldots, N-1\}$ **do**
2:    $t_s = 1 - s/N$
3:    $t_{s+1} = 1 - (s+1)/N$
4:    $\mathbf{x}_s \leftarrow (\mathbf{x}_{\text{cat},t_s}^{(1)}, \ldots, \mathbf{x}_{\text{cat},t_s}^{(K_{\text{cat}})}, x_{\text{cont},t_s}^{(1)}, \ldots, x_{\text{cont},t_s}^{(K_{\text{cont}})})$
5:    **for all** $i \in \{1, \ldots, K_{\text{cont}}\}$ **do**
6:       $\mathrm{d}x_{\text{cont}}^{(i)} = (x_{\text{cont},t_s}^{(i)} - D_{\boldsymbol{\theta}}^{\text{cont},i}(\mathbf{x}_s, t_s))/\sigma_{\text{cont},i}(t_s)$
7:       $x_{\text{cont},t_{s+1}}^{(i)} \leftarrow x_{\text{cont},t_s}^{(i)} + [\sigma_{\text{cont},i}(t_{s+1}) - \sigma_{\text{cont},i}(t_s)]\,\mathrm{d}x_{\text{cont}}^{(i)}$
8:    **end for**
9:    **for all** $j \in \{1, \ldots, K_{\text{cat}}\}$ **do**
10:       $\hat{\mathrm{P}}(\mathbf{x}_{\text{cat},0}^{(j)}|\mathbf{x}_{\text{cat},t_s}^{(j)}) = D_{\boldsymbol{\theta}}^{\text{cat},j}(\mathbf{x}_s, t_s)$
11:       $\mathrm{d}\mathbf{x}_{\text{cat}}^{(j)} = \left(\mathbf{x}_{\text{cat},t_s}^{(j)} - \mathbb{E}_{\hat{\mathrm{P}}(\mathbf{x}_{\text{cat},0}^{(j)}|\mathbf{x}_{\text{cat},t_s}^{(j)})}[\mathbf{x}_{\text{cat},0}^{(j)}]\right)/\sigma_{\text{cat},j}(t_s)$ from Equation (4)
12:       $\mathbf{x}_{\text{cat},t_{s+1}}^{(j)} \leftarrow \mathbf{x}_{\text{cat},t_s}^{(j)} + [\sigma_{\text{cat},j}(t_{s+1}) - \sigma_{\text{cat},j}(t_s)]\,\mathrm{d}\mathbf{x}_{\text{cat}}^{(j)}$
13:    **end for**
14: **end for**
    Recover classes from embeddings with additional pass to score model
15: $\mathbf{x}_N \leftarrow (\mathbf{x}_{\text{cat},t_N}^{(1)}, \ldots, \mathbf{x}_{\text{cat},t_N}^{(K_{\text{cat}})}, x_{\text{cont},t_N}^{(1)}, \ldots, x_{\text{cont},t_N}^{(K_{\text{cont}})})$
16: **for all** $j \in \{1, \ldots, K_{\text{cat}}\}$ **do**
17:    $\hat{\mathrm{P}}(\mathbf{x}_{\text{cat},0}^{(j)}|\mathbf{x}_{\text{cat},t_N}^{(j)}) = D_{\boldsymbol{\theta}}^{\text{cat},j}(\mathbf{x}_N, t_{N-1})$
18:    $x_{\text{cat}}^{(j)} \leftarrow \arg\max_{c \in C_j} \hat{\mathrm{P}}(\mathbf{x}_{\text{cat},0}^{(j)} = \mathbf{e}_c^{(j)}|\mathbf{x}_{\text{cat},t_N}^{(j)})$ (pick most likely category)
19: **end for**
20: **return** $(x_{\text{cat}}^{(1)}, \ldots, x_{\text{cat}}^{(K_{\text{cat}})}, x_{\text{cont},t_N}^{(1)}, \ldots, x_{\text{cont},t_N}^{(K_{\text{cont}})})$

---

where $\odot$ denotes the element-wise product. Hence, we are effectively taking *feature-specific* steps of length $\Delta\sigma_k(t)$. The adaptive noise schedules (timewarping) therefore not only affect the training process, but also focus most work in the reverse process on the noise levels that matter most for sample quality (i.e., where $\Delta\sigma_k(t)$ is small).

We use finite differences to approximate $\dot{\sigma}_i$, instead of the available, analytical variant, since $\frac{d\sigma_k(t)}{dt} \to \infty$ as $t \to 1$. The step $\Delta t$ would therefore be required to decrease as $t \to 1$ to ensure $\Delta t \approx \mathrm{d}t$ holds. For a large number of steps, this assumption does not hold in practice, and for $\frac{d\sigma_k(t)}{dt}$ the update of $\mathbf{x}$ overshoots the target drastically. Intuitively, $\sigma_k(t)$ becomes too steep near the terminal timestep $t = 1$ such that the step size can not sufficiently compensate for the slope increase to turn $\frac{d\sigma_k(t)}{dt}$ into a good approximation of the actual change in $\sigma_k(t)$. Moreover, the analytical solution would approximate $\mathrm{d}\sigma_k(t) = \dot{\sigma}_k(t)\mathrm{d}t$, i.e., the change in the noise level caused by a change in $t$. Since we know *exactly* where $\sigma_k(t)$ will end up when changing $t$, we are better off using that exact value and let $\mathrm{d}\sigma_k(t) = \Delta\sigma_k(t)$.

Table 6: Performance sensitivity of CDTD (per type) to increasing number of sampling steps. Each metric is averaged over five seeds. As a robust measure, we report the median over the ablation study datasets `acsincome`, `adult`, `beijing` and `churn`.

| Steps | RMSE | F1 | AUC | $L_2$ distance of corr. | Detection score | JSD | WD | DCR |
|---|---|---|---|---|---|---|---|---|
| 100 | 0.038 | 0.011 | 0.007 | 0.131 | 0.574 | 0.012 | 0.003 | 0.315 |
| 200 (default) | 0.030 | 0.009 | 0.006 | 0.130 | 0.565 | 0.012 | 0.003 | 0.316 |
| 500 | 0.027 | 0.009 | 0.006 | 0.130 | 0.557 | 0.013 | 0.002 | 0.313 |
| 1000 | 0.028 | 0.008 | 0.006 | 0.130 | 0.562 | 0.013 | 0.002 | 0.311 |

# M    SENSITIVITY TO IMPORTANT HYPERPARAMETERS

The training and sampling processes of CDTD are affected by various novel hyperparameters. Generally, a per-type noise schedule works best, as we show in our main results in Table 1 for a diverse set of benchmark datasets. Here, we examine the sensitivity of CDTD to two additional important hyperparameters: (1) the standard deviation of the noise used to initialize the embeddings (and therefore specific to score interpolation), $\sigma_{\text{init}}$, and (2) the weight of the low noise levels used to initialize the $\mu_k$ in the adaptive noise schedule parameterization.

The experiments in Dieleman et al. (2022) show that $\sigma_{\text{init}}$ is a crucial hyperparameter for score interpolation on text data. Table 7 shows that this sensitivity does not translate to the tabular data domain. This may be explained by the much smaller embedding dimension (16 vs. 256) or by our usage of *feature-specific* embeddings. Compared to a vocabulary size of 32000 for text data (Dieleman et al., 2022), we only face a maximum of 3151 categories in the `lending` dataset (see Table 3). Thus, unlike other generative (diffusion) models for tabular data, CDTD scales to a practically arbitrary number of categories.

Our proposed functional form for the adaptive noise schedules (see Appendix E) is the first to allow for the incorporation of prior information about the importance of low vs. high (normalized) noise levels. For this, we adjust the weight of low noise levels which directly determines the location of the inflection point $\mu_k$ (see Section 3.3). The results in Table 8 indicate low sensitivity of sample quality to weight changes for a per-type noise schedule. The initialization only impacts the time to convergence but not (much) the location of the optimum. In our experiments, the number of training steps (30000) appears to be high enough for all model variants to converge to similar states.

Table 7: Performance sensitivity of CDTD (per type) to changes in the standard deviation $\sigma_{\text{init}}$ in the initialization of the embeddings of categorical features. Each metric is averaged over five seeds. As a robust measure, we report the median over the ablation study datasets `acsincome`, `adult`, `beijing` and `churn`.

| $\sigma_{\text{init}}$ | RMSE | F1 | AUC | L$_2$ distance of corr. | Detection score | JSD | WD | DCR |
|---|---|---|---|---|---|---|---|---|
| 1 | 0.031 | 0.012 | 0.004 | 0.123 | 0.555 | 0.013 | 0.003 | 0.294 |
| 0.1 | 0.027 | 0.012 | 0.004 | 0.125 | 0.556 | 0.013 | 0.003 | 0.323 |
| 0.01 | 0.031 | 0.007 | 0.005 | 0.129 | 0.575 | 0.013 | 0.003 | 0.320 |
| 0.001 (default) | 0.030 | 0.009 | 0.006 | 0.130 | 0.565 | 0.012 | 0.003 | 0.316 |

Table 8: Performance sensitivity of CDTD (per type) to changes in the prior weight of low noise levels in the initialization of the adaptive noise schedules. Each metric is averaged over five seeds. As a robust measure, we report the median over the ablation study datasets `acsincome`, `adult`, `beijing` and `churn`.

| Weight | RMSE | F1 | AUC | L$_2$ distance of corr. | Detection score | JSD | WD | DCR |
|---|---|---|---|---|---|---|---|---|
| 1 | 0.031 | 0.010 | 0.005 | 0.126 | 0.570 | 0.012 | 0.002 | 0.238 |
| 2 | 0.030 | 0.009 | 0.004 | 0.125 | 0.570 | 0.013 | 0.002 | 0.289 |
| 3 (default) | 0.030 | 0.009 | 0.006 | 0.130 | 0.565 | 0.012 | 0.003 | 0.316 |
| 4 | 0.028 | 0.011 | 0.005 | 0.134 | 0.574 | 0.011 | 0.003 | 0.350 |

# N    ADVANTAGES OF DIFFUSION IN DATA SPACE

These days, inspired from diffusion models in the image and video domains, much work relies on the idea of latent diffusion. Here, we want to briefly discuss and emphasize that for tabular data, diffusion in latent space (represented by TabSyn) has important drawbacks and how CDTD, a diffusion model defined in data space alleviates those.

Latent diffusion models first encode the data in a latent space. The diffusion model itself is then trained in that latent space instead of directly on the features. Hence, the performance of the diffusion model directly depends on a second model, with a separate training procedure. TabSyn uses a VAE model to encode mixed-type data into a common continuous space that is *not* lower-dimensional, so as to minimize reconstruction errors. Any reconstruction errors caused by the VAE reduce the

sample quality of the eventually generated samples, no matter the capacity of the diffusion model. This suggests that we would want to train a highly capable encoder/decoder, which adds additional training costs. Figure 3 shows that latent diffusion is not necessarily more efficient in the tabular data domain. In particular, if the latent space is not lower-dimensional or the encoder/decoder is very complex, then sampling speed is not improved.

We further hypothesize that much tabular data, due to the lack of redundancy and spatial or sequential correlation, is difficult to summarize efficiently in a joint latent space. Hence, compared to other domains, larger VAEs and higher-dimensional latent spaces are required, increasing the training time. Also, there is the risk of the VAE not picking up on subtle correlations within the data or distorting existing correlations by mapping into the latent space. Any correlations not properly encoded cannot be learned by the diffusion model. Since we optimize the VAE on an *average* loss, its reconstruction and encoding performance of, for instance, minority classes or extreme values in long-tailed distributions is likely lacking. This makes the job of the diffusion model more difficult, if not sometimes impossible.

Lastly, we take great care in homogenizing categorical and continuous features throughout the training process (see Appendix B and C). This is a crucial part of modeling *mixed*-type data. Using a VAE to define a diffusion process in latent space only shifts the necessity for homogenization to the VAE training process. Not balancing different feature- or data-types and their losses induces implicit importance weights. Thus, the VAE may sacrifice the reconstruction quality of some features in favor of others (Kendall et al., 2018; Ma et al., 2020).

To empirically investigate the difference of diffusion in data space (CDTD) and latent diffusion (TabSyn), we examine *feature-specific* sample quality metrics and those that directly benefit from *all* features being generated well. Our results in Table 9 show that latent diffusion comes with a considerable decrease in sample quality (while imposing a similar architecture and number of parameters as well as sampling steps, see Appendix H). In particular, the attained maximum univariate metrics as well as the detection score and the $L_2$ distance of the correlation matrices indicate that TabSyn has issues modeling *all* features and their correlations sufficiently well. This supports our argument that a homogenization of data types is important and that an encoding in latent space may complicate the learning of joint data characteristics.

Table 9: A comparison of the CDTD model to latent diffusion (TabSyn). We average each metric over five sampling seeds and as a robust measure report the median over the 11 datasets. Abs. diff. in corr. matrices refers to the absolute differences in the correlation matrices between ground truth and synthetic data. The max, min and mean are taken across features.

| | Detection score | $L_2$ dist. of corr. | JSD | | | WD | | | Abs. diff. in corr. matrices | |
|---|---|---|---|---|---|---|---|---|---|---|
| | | | min | mean | max | min | mean | max | min | max |
| TabSyn | 0.859 | 0.919 | 0.004 | 0.044 | 0.141 | 0.002 | 0.012 | 0.025 | 0.000 | 0.261 |
| CDTD (per type) | 0.701 | 0.444 | 0.002 | 0.011 | 0.025 | 0.001 | 0.006 | 0.010 | 0.000 | 0.088 |
| Improv. over TabSyn | 18.4% | 51.7% | 50.0% | 75.0% | 82.3% | 50.0% | 50.0% | 60.0% | - | 66.3% |

## O    COMPARISON TO RELATED WORK

Table 10 summarizes our comparison of CDTD to the diffusion-based benchmark models, that is, TabSyn, TabDDPM and CoDi. Of those models, only TabSyn applies diffusion in latent space, which comes with both advantages and costs (as discussed in Appendix N). TabSyn is the only other model besides CDTD that avoids one-hot encoding categorical features by using embeddings. This improves the scalability to a higher number of categories without blowing up the input dimensions. Although both models utilize embeddings, TabSyn's generative capabilities are more constrained by jointly encoding all features in a latent space. It should also be noted that TabSyn makes use of a Transformer architecture in its VAE, which means that it scales quadratically in the number of features and therefore may not be easily scaled to high-dimensional data.

CDTD is the first model to utilize adaptive and type- or feature-specific noise schedules to model tabular data. Further, we take great care in homogenizing categorical and continuous features

throughout the training process (see Appendix B and C). No other model attempts balancing the different features types. This is problematic as it suggests that other models may suffer from feature-specific implicit importance weights that impact both training and generation. Hence, the sample quality of some features may be unintentionally sacrificed in favor of increasing the sample quality of other features (Kendall et al., 2018; Ma et al., 2020). Note that this also applies to TabSyn: Even though their diffusion model avoids this issue by relying on a single type of loss due to the continuous latent space, the VAE training process does not account for any balancing issues between the two data types. Hence, the balancing issue is not eliminated but got only shifted to the VAE.

Lastly, CDTD and TabSyn are the only models that define the diffusion process in *continuous* space. As such, other advanced techniques, like classifier-free guidance or ODE/SDE samplers, can be directly applied. To accommodate categorical data, CoDi and TabDDPM make use of multinomial diffusion (Hoogeboom et al., 2021), which is an inherently *discrete* process and therefore prohibits such applications.

Table 10: Comparison of CDTD to the diffusion-based generative models CoDi, TabDDPM and TabSyn. (∗) Note that the VAE trained as part of the TabSyn model does not balance type-specific losses, which induces an implicit weighting among features. This can worsen the sample quality of some features in favor of others.

| | defined in feature space | avoids one-hot encoding | balances feature types | adaptive noise schedule | type- or feature-specific noise schedules | diffusion in continuous space |
|---|---|---|---|---|---|---|
| CoDi | ✓ | | | | | |
| TabDDPM | ✓ | | | | | |
| TabSyn | | ✓ | ∗ | | | ✓ |
| **CDTD (ours)** | ✓ | ✓ | ✓ | ✓ | ✓ | ✓ |

# P    EXAMPLES OF LEARNED NOISE SCHEDULES

Next, we show the learned noise schedules for the largest (`acsincome`) and the smallest (`churn`) datasets. Additionally, we illustrate the fit of *single*, *per type* and *per feature* schedules to the respective diffusion losses they were trained to fit.

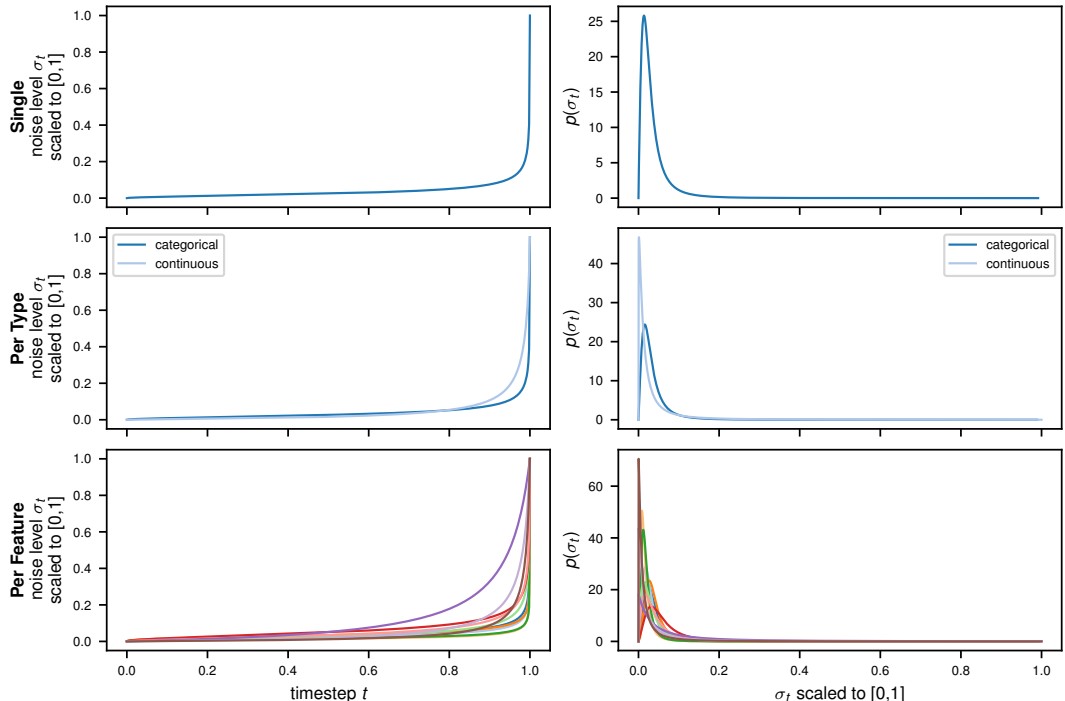

Figure 6: (Left): Learned noise schedules for `acsincome`. This reflects $F_{\mathrm{d.a.log},k}^{-1}$. (Right): Implicit weighting of noise levels / timesteps. This visualizes $f_{\mathrm{d.a.log},k}$.

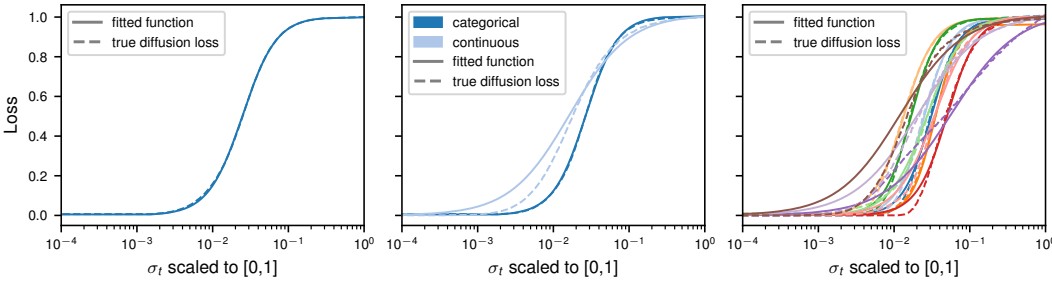

Figure 7: Illustration of the goodness of fit of the timewarping function $F_k$ for single (left), per type (middle) and per feature noise schedules (right) on the `acsincome` data.

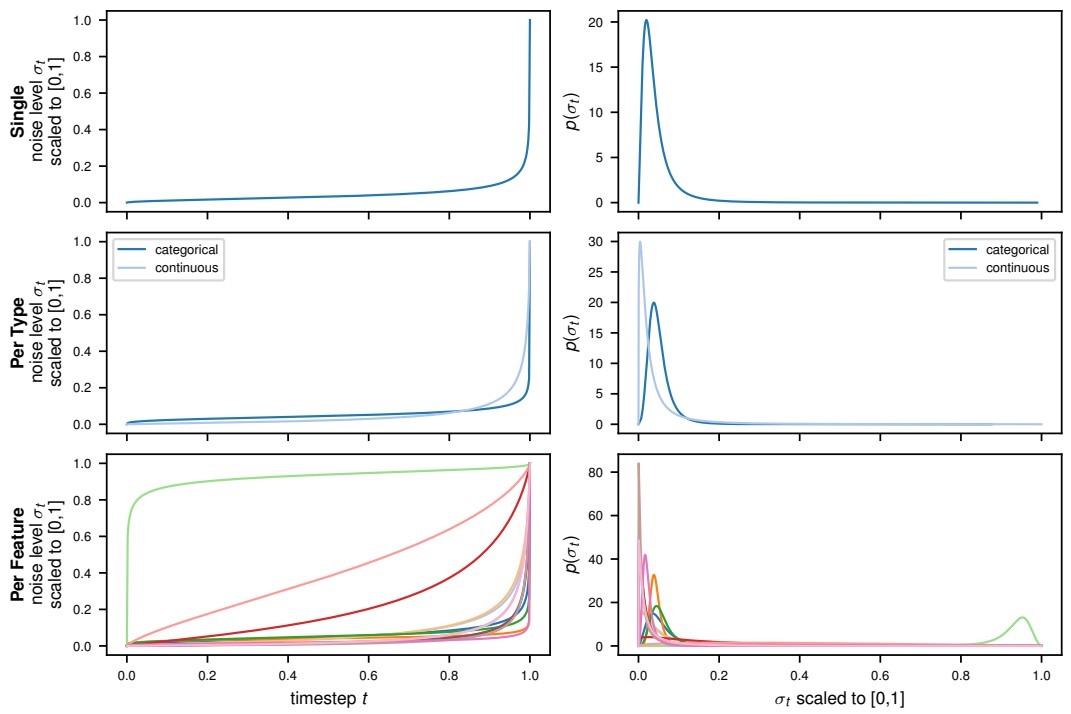

Figure 8: (Left): Learned noise schedules for churn. This reflects $F_{\mathrm{d.a.log},k}^{-1}$. (Right): Implicit weighting of noise levels / timesteps. This visualizes $f_{\mathrm{d.a.log},k}$.

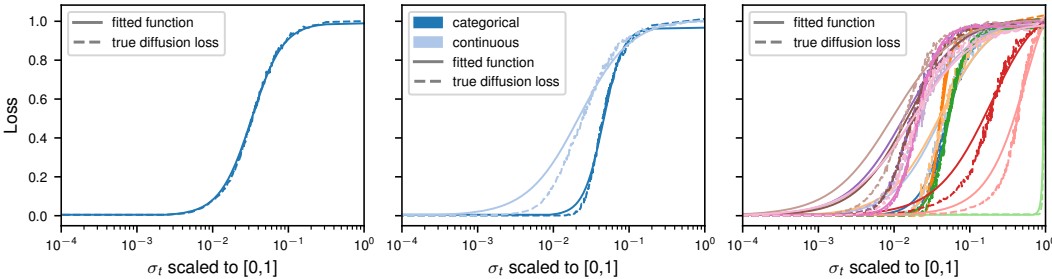

Figure 9: Illustration of the goodness of fit of the timewarping function $F_k$ for single (left), per type (middle) and per feature noise schedules (right) on the churn data.

# Q QUALITATIVE COMPARISONS

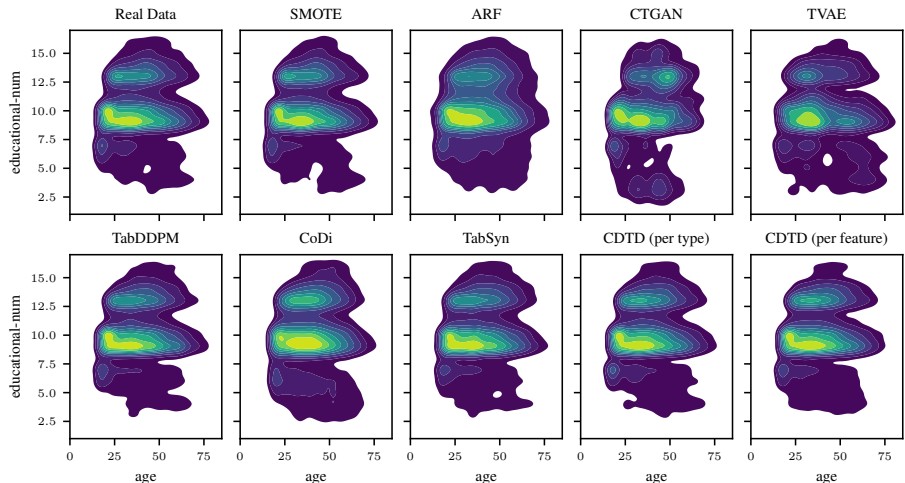

Figure 10: Bivariate density for age and educational-num from the `adult` data.

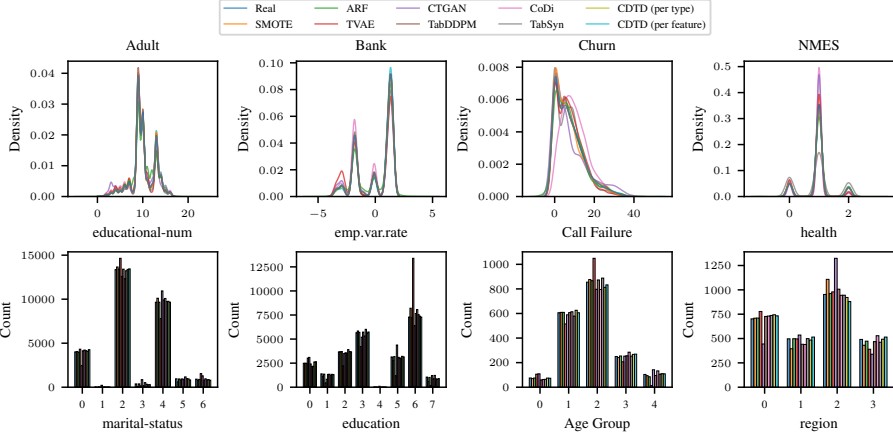

Figure 11: Comparison of some univariate distributions for `adult`, `bank`, `churn`, `nmes`.

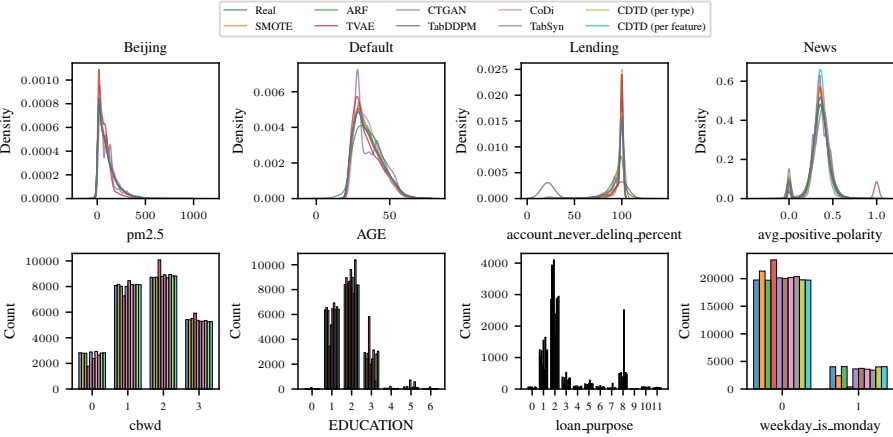

Figure 12: Comparison of some univariate distributions for `beijing`, `default`, `lending`, `news`. (Note that CoDi is prohibitively expensive to train on `lending` and therefore excluded.)

# R  VISUALIZATIONS OF CAPTURED CORRELATIONS

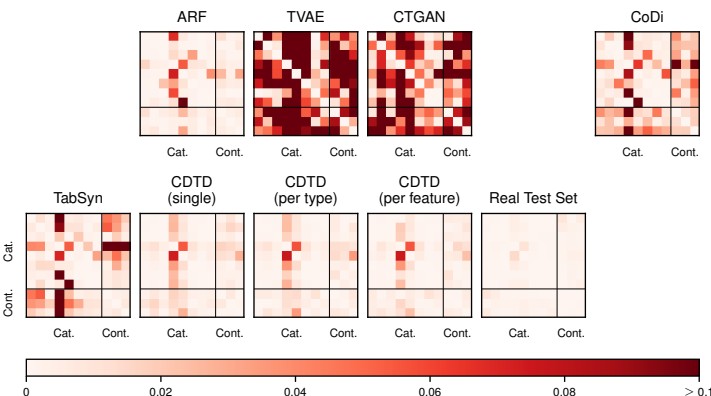

Figure 13: Element-wise absolute differences of the correlation matrices between the real training set and the synthetic data for the `acsincome` dataset. TabDDPM generates NaNs for this dataset and is therefore excluded. SMOTE takes too long for sampling. Continuous (cont.) and categorical (cat.) features are indicated on the axes.

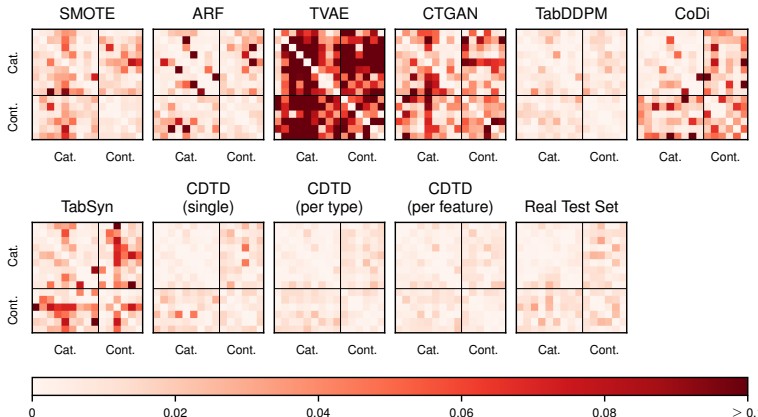

Figure 14: Element-wise absolute differences of the correlation matrices between the real training set and the synthetic data for the `adult` dataset. Continuous (cont.) and categorical (cat.) features are indicated on the axes.

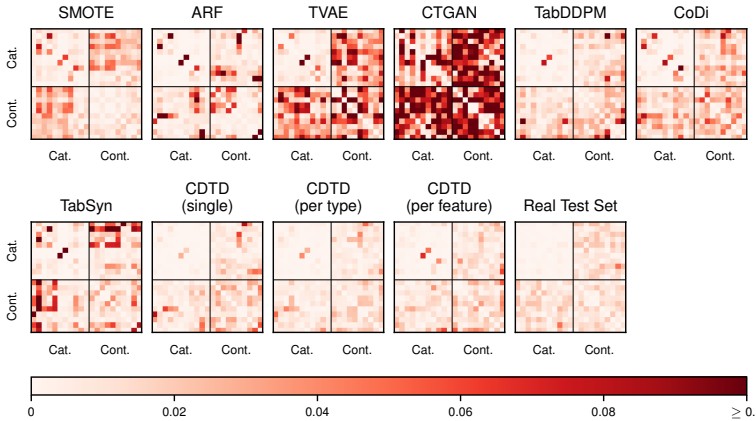

Figure 15: Element-wise absolute differences of the correlation matrices between the real training set and the synthetic data for the `bank` dataset. Continuous (cont.) and categorical (cat.) features are indicated on the axes.

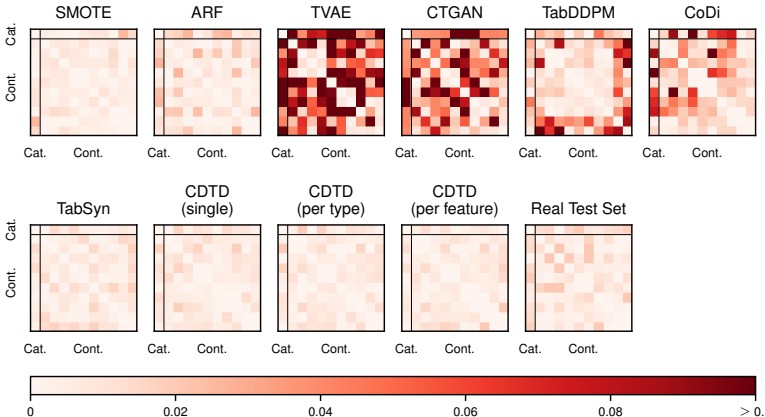

Figure 16: Element-wise absolute differences of the correlation matrices between the real training set and the synthetic data for the `beijing` dataset. Continuous (cont.) and categorical (cat.) features are indicated on the axes.

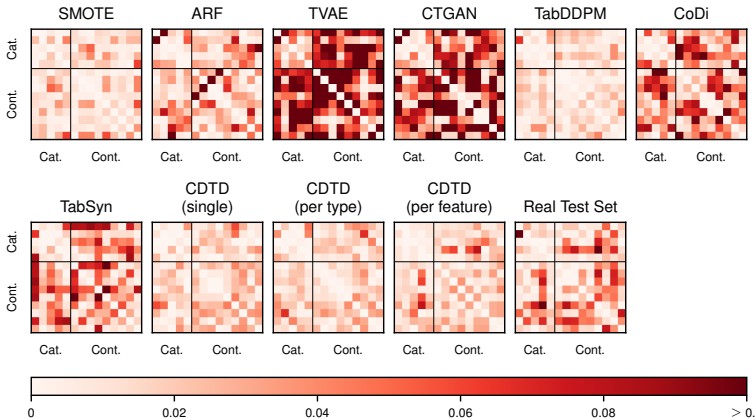

Figure 17: Element-wise absolute differences of the correlation matrices between the real training set and the synthetic data for the `churn` dataset. Continuous (cont.) and categorical (cat.) features are indicated on the axes.

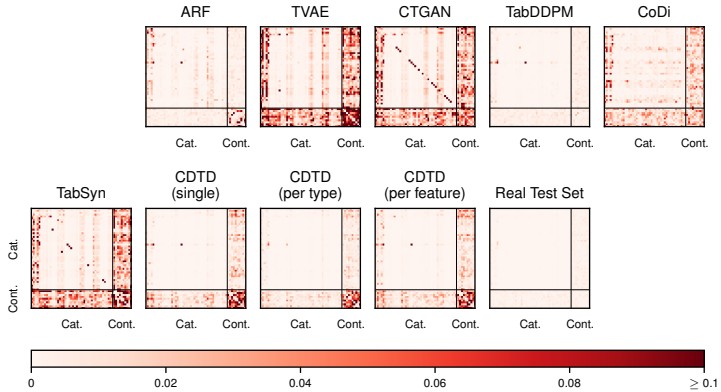

Figure 18: Element-wise absolute differences of the correlation matrices between the real training set and the synthetic data for the `covertype` dataset. SMOTE takes too long for sampling. Continuous (cont.) and categorical (cat.) features are indicated on the axes.

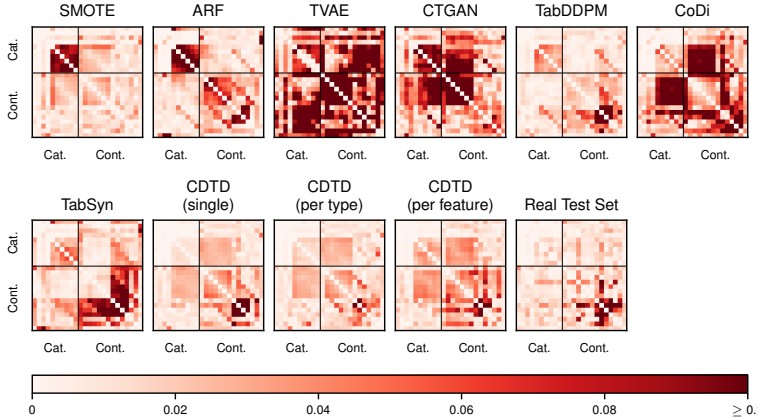

Figure 19: Element-wise absolute differences of the correlation matrices between the real training set and the synthetic data for the `default` dataset. Continuous (cont.) and categorical (cat.) features are indicated on the axes.

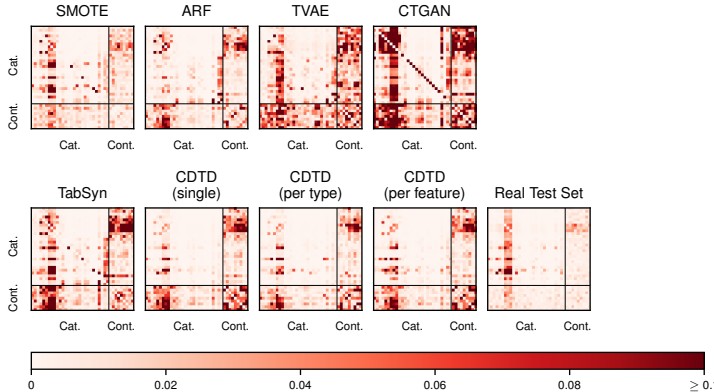

Figure 20: Element-wise absolute differences of the correlation matrices between the real training set and the synthetic data for the `diabetes` dataset. TabDDPM generates NaNs for this dataset and is therefore excluded. CoDi is prohibitively expensive to train and therefore excluded. Continuous (cont.) and categorical (cat.) features are indicated on the axes.

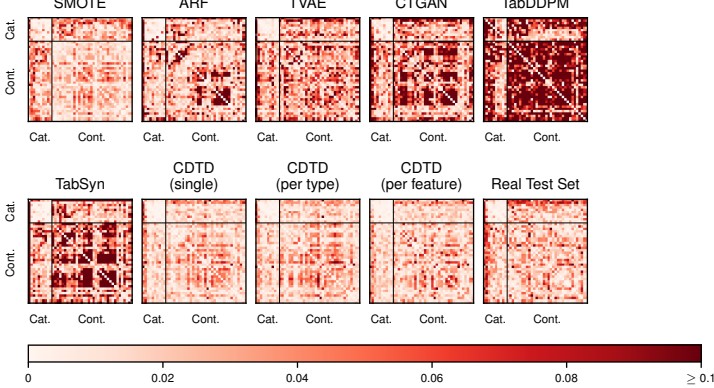

Figure 21: Element-wise absolute differences of the correlation matrices between the real training set and the synthetic data for the `lending` dataset. CoDi is prohibitively expensive to train and therefore excluded. Continuous (cont.) and categorical (cat.) features are indicated on the axes.

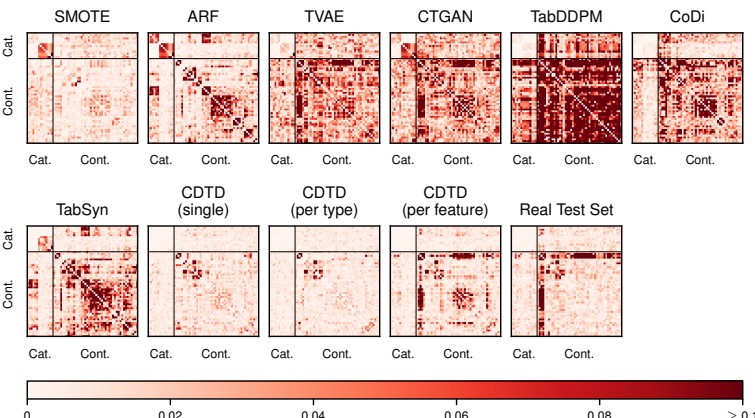

Figure 22: Element-wise absolute differences of the correlation matrices between the real training set and the synthetic data for the `news` dataset. Continuous (cont.) and categorical (cat.) features are indicated on the axes.

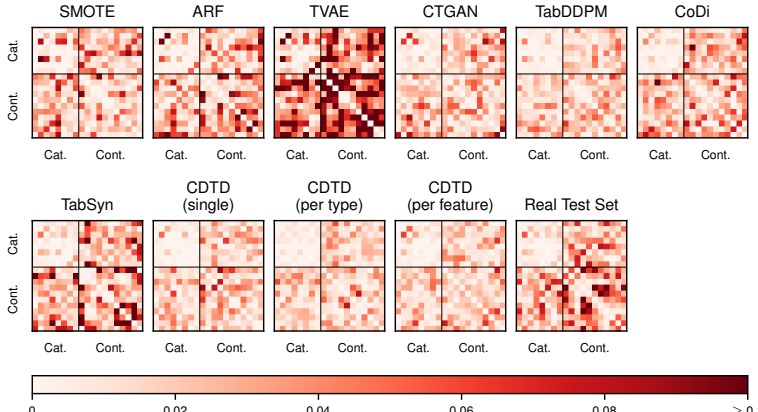

Figure 23: Element-wise absolute differences of the correlation matrices between the real training set and the synthetic data for the `nmes` dataset. Continuous (cont.) and categorical (cat.) features are indicated on the axes.

## S  DETAILED RESULTS

CoDi is prohibitively expensive to train on `lending` and `diabetes` and TabDDPM produces NaNs for `acsincome` and `diabetes`. SMOTE takes too long to sample datasets of a sufficient size for `acsincome` and `covertype` (see Table 29). For those models, the performance metrics on these datasets are therefore not reported. They are assigned a rank of 10 in Table 1 and the worst metric-specific performance across all models before computing the average metrics reported in Table 11.

Table 11: Model evaluation results averaged over 11 datasets (if a model was not trainable on a given dataset, we assign the maximum, i.e., worst, value over all models for that dataset to this model) for seven benchmark models and for CDTD with three different noise schedules. Per performance metric, **bold** indicates the best, underline the second best result.

| | SMOTE | ARF | CTGAN | TVAE | TabDDPM | CoDi | TabSyn | CDTD (single) | CDTD (per type) | CDTD (per feature) |
|---|---|---|---|---|---|---|---|---|---|---|
| RMSE (abs. diff.; $\downarrow$) | 0.366 | 0.091 | 0.635 | 0.868 | 0.674 | 0.278 | 0.309 | **0.086** | 0.091 | 0.102 |
| F1 (abs. diff.; $\downarrow$) | 0.068 | 0.053 | 0.130 | 0.074 | 0.020 | 0.048 | 0.111 | 0.020 | 0.015 | **0.014** |
| AUC (abs. diff.; $\downarrow$) | 0.043 | 0.020 | 0.080 | 0.025 | 0.039 | 0.066 | 0.066 | 0.015 | **0.014** | 0.015 |
| L$_2$ distance of corr. ($\downarrow$) | 1.287 | 1.321 | 2.190 | 2.707 | 3.001 | 2.383 | 2.097 | 0.792 | **0.684** | 0.920 |
| Detection score ($\downarrow$) | 0.869 | 0.933 | 0.986 | 0.977 | 0.790 | 0.947 | 0.858 | 0.761 | **0.739** | 0.770 |
| JSD ($\downarrow$) | 0.077 | **0.011** | 0.101 | 0.135 | 0.086 | 0.073 | 0.052 | 0.015 | 0.016 | 0.018 |
| WD ($\downarrow$) | 0.011 | 0.011 | 0.024 | 0.023 | 0.050 | 0.059 | 0.017 | 0.010 | **0.008** | 0.009 |
| DCR (abs. diff. to test; $\downarrow$) | 1.813 | 1.588 | 3.317 | 1.602 | 2.061 | 2.838 | 2.648 | 0.927 | 0.867 | **0.760** |

Table 12: L$_2$ norm (incl. standard errors in subscripts) of the correlation matrix differences of real and synthetic train sets for seven benchmark models and for CDTD with three different noise schedules.

| | SMOTE | ARF | CTGAN | TVAE | TabDDPM | CoDi | TabSyn | CDTD (single) | CDTD (per type) | CDTD (per feature) |
|---|---|---|---|---|---|---|---|---|---|---|
| acsincome | - | $0.242_{\pm0.002}$ | $1.696_{\pm0.008}$ | $1.136_{\pm0.004}$ | - | $0.517_{\pm0.006}$ | $0.560_{\pm0.005}$ | $0.134_{\pm0.002}$ | $0.135_{\pm0.002}$ | $0.123_{\pm0.002}$ |
| adult | $0.414_{\pm0.016}$ | $0.576_{\pm0.006}$ | $1.858_{\pm0.010}$ | $0.735_{\pm0.012}$ | $0.160_{\pm0.012}$ | $0.493_{\pm0.009}$ | $0.514_{\pm0.013}$ | $0.175_{\pm0.007}$ | $0.125_{\pm0.005}$ | $0.123_{\pm0.011}$ |
| bank | $0.404_{\pm0.015}$ | $0.819_{\pm0.024}$ | $0.947_{\pm0.019}$ | $2.758_{\pm0.049}$ | $0.529_{\pm0.050}$ | $0.499_{\pm0.021}$ | $0.759_{\pm0.013}$ | $0.333_{\pm0.018}$ | $0.231_{\pm0.009}$ | $0.288_{\pm0.013}$ |
| beijing | $0.081_{\pm0.007}$ | $0.128_{\pm0.009}$ | $1.470_{\pm0.007}$ | $1.226_{\pm0.008}$ | $0.368_{\pm0.085}$ | $0.373_{\pm0.008}$ | $0.086_{\pm0.008}$ | $0.073_{\pm0.009}$ | $0.073_{\pm0.007}$ | $0.071_{\pm0.009}$ |
| churn | $0.264_{\pm0.036}$ | $0.635_{\pm0.026}$ | $1.355_{\pm0.043}$ | $1.301_{\pm0.041}$ | $0.273_{\pm0.069}$ | $0.746_{\pm0.062}$ | $0.613_{\pm0.022}$ | $0.269_{\pm0.028}$ | $0.255_{\pm0.021}$ | $0.271_{\pm0.040}$ |
| covertype | - | $1.192_{\pm0.017}$ | $3.685_{\pm0.005}$ | $4.668_{\pm0.003}$ | $1.124_{\pm0.183}$ | $1.029_{\pm0.032}$ | $3.749_{\pm0.181}$ | $1.970_{\pm0.010}$ | $1.357_{\pm0.155}$ | $1.972_{\pm0.011}$ |
| default | $0.709_{\pm0.048}$ | $1.228_{\pm0.021}$ | $2.697_{\pm0.021}$ | $1.564_{\pm0.029}$ | $0.685_{\pm0.131}$ | $1.672_{\pm0.061}$ | | $0.724_{\pm0.116}$ | $0.641_{\pm0.130}$ | $0.681_{\pm0.037}$ |
| diabetes | $2.355_{\pm0.026}$ | $1.189_{\pm0.004}$ | $1.654_{\pm0.008}$ | $5.351_{\pm0.095}$ | - | - | $2.796_{\pm0.066}$ | $1.381_{\pm0.016}$ | $1.213_{\pm0.029}$ | $1.351_{\pm0.044}$ |
| lending | $1.321_{\pm0.063}$ | $3.473_{\pm0.057}$ | $2.420_{\pm0.016}$ | $5.895_{\pm0.026}$ | $10.046_{\pm0.007}$ | - | $6.792_{\pm0.034}$ | $1.148_{\pm0.087}$ | $1.239_{\pm0.090}$ | $1.351_{\pm0.050}$ |
| news | $1.684_{\pm1.466}$ | $4.333_{\pm0.128}$ | $4.641_{\pm0.028}$ | $4.612_{\pm0.016}$ | $12.356_{\pm0.097}$ | $4.874_{\pm0.148}$ | $5.153_{\pm0.014}$ | $2.050_{\pm0.594}$ | $1.811_{\pm0.295}$ | $3.446_{\pm1.111}$ |
| nmes | $0.565_{\pm0.047}$ | $0.717_{\pm0.054}$ | $1.663_{\pm0.035}$ | $0.532_{\pm0.030}$ | $0.426_{\pm0.041}$ | $0.609_{\pm0.032}$ | $0.919_{\pm0.067}$ | $0.454_{\pm0.043}$ | $0.444_{\pm0.075}$ | $0.445_{\pm0.071}$ |

Table 13: Jensen-Shannon divergence (incl. standard errors in subscripts) for seven benchmark models and for CDTD with three different noise schedules.

| | SMOTE | ARF | CTGAN | TVAE | TabDDPM | CoDi | TabSyn | CDTD (single) | CDTD (per type) | CDTD (per feature) |
|---|---|---|---|---|---|---|---|---|---|---|
| acsincome | - | $0.013_{\pm0.001}$ | $0.256_{\pm0.000}$ | $0.309_{\pm0.000}$ | - | $0.076_{\pm0.001}$ | $0.052_{\pm0.001}$ | $0.021_{\pm0.001}$ | $0.022_{\pm0.001}$ | $0.019_{\pm0.000}$ |
| adult | $0.064_{\pm0.001}$ | $0.007_{\pm0.001}$ | $0.112_{\pm0.001}$ | $0.113_{\pm0.001}$ | $0.035_{\pm0.001}$ | $0.045_{\pm0.001}$ | $0.022_{\pm0.001}$ | $0.011_{\pm0.001}$ | $0.015_{\pm0.001}$ | $0.015_{\pm0.001}$ |
| bank | $0.039_{\pm0.001}$ | $0.004_{\pm0.000}$ | $0.086_{\pm0.001}$ | $0.191_{\pm0.001}$ | $0.021_{\pm0.001}$ | $0.038_{\pm0.001}$ | $0.063_{\pm0.001}$ | $0.011_{\pm0.000}$ | $0.011_{\pm0.001}$ | $0.015_{\pm0.001}$ |
| beijing | $0.006_{\pm0.002}$ | $0.004_{\pm0.000}$ | $0.005_{\pm0.002}$ | $0.074_{\pm0.002}$ | $0.024_{\pm0.002}$ | $0.011_{\pm0.003}$ | $0.011_{\pm0.003}$ | $0.006_{\pm0.002}$ | $0.006_{\pm0.001}$ | $0.007_{\pm0.002}$ |
| churn | $0.012_{\pm0.004}$ | $0.011_{\pm0.004}$ | $0.095_{\pm0.003}$ | $0.048_{\pm0.004}$ | $0.015_{\pm0.006}$ | $0.043_{\pm0.001}$ | $0.031_{\pm0.002}$ | $0.011_{\pm0.002}$ | $0.010_{\pm0.002}$ | $0.012_{\pm0.003}$ |
| covertype | - | $0.002_{\pm0.000}$ | $0.044_{\pm0.000}$ | $0.043_{\pm0.000}$ | $0.004_{\pm0.000}$ | $0.008_{\pm0.000}$ | $0.044_{\pm0.000}$ | $0.004_{\pm0.000}$ | $0.003_{\pm0.000}$ | $0.007_{\pm0.000}$ |
| default | $0.042_{\pm0.001}$ | $0.008_{\pm0.001}$ | $0.194_{\pm0.001}$ | $0.177_{\pm0.001}$ | $0.028_{\pm0.001}$ | $0.073_{\pm0.002}$ | $0.093_{\pm0.001}$ | $0.012_{\pm0.002}$ | $0.016_{\pm0.001}$ | $0.017_{\pm0.001}$ |
| diabetes | $0.067_{\pm0.000}$ | $0.009_{\pm0.000}$ | $0.093_{\pm0.000}$ | $0.187_{\pm0.000}$ | - | - | $0.098_{\pm0.000}$ | $0.024_{\pm0.000}$ | $0.025_{\pm0.000}$ | $0.031_{\pm0.000}$ |
| lending | $0.143_{\pm0.001}$ | $0.049_{\pm0.002}$ | $0.092_{\pm0.001}$ | $0.188_{\pm0.001}$ | $0.287_{\pm0.002}$ | - | $0.119_{\pm0.001}$ | $0.056_{\pm0.001}$ | $0.057_{\pm0.001}$ | $0.065_{\pm0.002}$ |
| news | $0.063_{\pm0.001}$ | $0.002_{\pm0.000}$ | $0.022_{\pm0.001}$ | $0.128_{\pm0.001}$ | $0.017_{\pm0.001}$ | $0.012_{\pm0.001}$ | $0.017_{\pm0.001}$ | $0.003_{\pm0.001}$ | $0.003_{\pm0.001}$ | $0.003_{\pm0.001}$ |
| nmes | $0.060_{\pm0.001}$ | $0.008_{\pm0.002}$ | $0.117_{\pm0.002}$ | $0.029_{\pm0.003}$ | $0.025_{\pm0.004}$ | $0.027_{\pm0.003}$ | $0.019_{\pm0.001}$ | $0.009_{\pm0.001}$ | $0.007_{\pm0.002}$ | $0.012_{\pm0.004}$ |

Table 14: Wasserstein distance (incl. standard errors in subscripts) for seven benchmark models and for CDTD with three different noise schedules.

| | SMOTE | ARF | CTGAN | TVAE | TabDDPM | CoDi | TabSyn | CDTD (single) | CDTD (per type) | CDTD (per feature) |
|---|---|---|---|---|---|---|---|---|---|---|
| acsincome | - | $0.007_{\pm0.000}$ | $0.037_{\pm0.000}$ | $0.021_{\pm0.000}$ | - | $0.017_{\pm0.000}$ | $0.005_{\pm0.000}$ | $0.002_{\pm0.000}$ | $0.001_{\pm0.000}$ | $0.001_{\pm0.000}$ |
| adult | $0.003_{\pm0.000}$ | $0.012_{\pm0.000}$ | $0.016_{\pm0.000}$ | $0.021_{\pm0.000}$ | $0.002_{\pm0.000}$ | $0.013_{\pm0.000}$ | $0.007_{\pm0.000}$ | $0.007_{\pm0.000}$ | $0.003_{\pm0.000}$ | $0.002_{\pm0.000}$ |
| bank | $0.002_{\pm0.001}$ | $0.012_{\pm0.000}$ | $0.021_{\pm0.000}$ | $0.040_{\pm0.000}$ | $0.004_{\pm0.000}$ | $0.030_{\pm0.001}$ | $0.006_{\pm0.000}$ | $0.006_{\pm0.000}$ | $0.003_{\pm0.001}$ | $0.007_{\pm0.001}$ |
| beijing | $0.002_{\pm0.000}$ | $0.008_{\pm0.000}$ | $0.030_{\pm0.000}$ | $0.036_{\pm0.000}$ | $0.007_{\pm0.000}$ | $0.019_{\pm0.000}$ | $0.004_{\pm0.000}$ | $0.003_{\pm0.000}$ | $0.002_{\pm0.000}$ | $0.002_{\pm0.000}$ |
| churn | $0.006_{\pm0.001}$ | $0.013_{\pm0.001}$ | $0.027_{\pm0.001}$ | $0.032_{\pm0.001}$ | $0.007_{\pm0.002}$ | $0.048_{\pm0.002}$ | $0.013_{\pm0.002}$ | $0.006_{\pm0.001}$ | $0.006_{\pm0.001}$ | $0.006_{\pm0.001}$ |
| covertype | - | $0.006_{\pm0.000}$ | $0.041_{\pm0.000}$ | $0.022_{\pm0.000}$ | $0.002_{\pm0.000}$ | $0.012_{\pm0.000}$ | $0.019_{\pm0.000}$ | $0.018_{\pm0.000}$ | $0.009_{\pm0.000}$ | $0.013_{\pm0.000}$ |
| default | $0.002_{\pm0.000}$ | $0.005_{\pm0.000}$ | $0.011_{\pm0.000}$ | $0.005_{\pm0.000}$ | $0.002_{\pm0.000}$ | $0.013_{\pm0.000}$ | $0.003_{\pm0.000}$ | $0.004_{\pm0.000}$ | $0.003_{\pm0.000}$ | $0.003_{\pm0.000}$ |
| diabetes | $0.004_{\pm0.000}$ | $0.012_{\pm0.000}$ | $0.020_{\pm0.000}$ | $0.038_{\pm0.000}$ | - | - | $0.012_{\pm0.000}$ | $0.042_{\pm0.000}$ | $0.034_{\pm0.000}$ | $0.041_{\pm0.000}$ |
| lending | $0.006_{\pm0.000}$ | $0.013_{\pm0.001}$ | $0.011_{\pm0.000}$ | $0.016_{\pm0.000}$ | $0.410_{\pm0.001}$ | - | $0.053_{\pm0.000}$ | $0.012_{\pm0.000}$ | $0.013_{\pm0.000}$ | $0.011_{\pm0.000}$ |
| news | $0.007_{\pm0.000}$ | $0.024_{\pm0.000}$ | $0.009_{\pm0.000}$ | $0.018_{\pm0.000}$ | $0.033_{\pm0.001}$ | $0.030_{\pm0.000}$ | $0.029_{\pm0.000}$ | $0.008_{\pm0.000}$ | $0.006_{\pm0.000}$ | $0.008_{\pm0.000}$ |
| nmes | $0.005_{\pm0.001}$ | $0.012_{\pm0.000}$ | $0.036_{\pm0.000}$ | $0.008_{\pm0.000}$ | $0.007_{\pm0.001}$ | $0.016_{\pm0.001}$ | $0.032_{\pm0.001}$ | $0.006_{\pm0.001}$ | $0.006_{\pm0.000}$ | $0.006_{\pm0.000}$ |

Table 15: Detection score (incl. standard errors in subscripts) for seven benchmark models and for CDTD with three different noise schedules.

| | SMOTE | ARF | CTGAN | TVAE | TabDDPM | CoDi | TabSyn | CDTD (single) | CDTD (per type) | CDTD (per feature) |
|---|---|---|---|---|---|---|---|---|---|---|
| acsincome | - | $0.808_{\pm0.001}$ | $0.989_{\pm0.001}$ | $0.985_{\pm0.000}$ | - | $0.825_{\pm0.002}$ | $0.688_{\pm0.003}$ | $0.529_{\pm0.005}$ | $0.527_{\pm0.002}$ | $0.528_{\pm0.002}$ |
| adult | $0.687_{\pm0.003}$ | $0.889_{\pm0.002}$ | $0.997_{\pm0.000}$ | $0.967_{\pm0.001}$ | $0.594_{\pm0.003}$ | $0.992_{\pm0.001}$ | $0.641_{\pm0.003}$ | $0.621_{\pm0.002}$ | $0.590_{\pm0.004}$ | $0.605_{\pm0.004}$ |
| bank | $0.839_{\pm0.003}$ | $0.955_{\pm0.002}$ | $1.000_{\pm0.000}$ | $0.988_{\pm0.001}$ | $0.781_{\pm0.002}$ | $0.853_{\pm0.003}$ | $0.808_{\pm0.003}$ | $0.701_{\pm0.005}$ | $0.835_{\pm0.001}$ |
| beijing | $0.938_{\pm0.002}$ | $0.989_{\pm0.002}$ | $0.996_{\pm0.001}$ | $0.995_{\pm0.001}$ | $0.738_{\pm0.004}$ | $0.989_{\pm0.001}$ | $0.723_{\pm0.003}$ | $0.574_{\pm0.003}$ | $0.620_{\pm0.005}$ | $0.614_{\pm0.003}$ |
| churn | $0.567_{\pm0.015}$ | $0.853_{\pm0.002}$ | $0.945_{\pm0.006}$ | $0.843_{\pm0.011}$ | $0.556_{\pm0.013}$ | $0.730_{\pm0.012}$ | $0.859_{\pm0.005}$ | $0.614_{\pm0.016}$ | $0.541_{\pm0.008}$ | $0.639_{\pm0.011}$ |
| covertype | - | $0.945_{\pm0.002}$ | $0.997_{\pm0.000}$ | $0.989_{\pm0.001}$ | $0.584_{\pm0.002}$ | $0.900_{\pm0.002}$ | $0.991_{\pm0.000}$ | $0.989_{\pm0.000}$ | $0.981_{\pm0.000}$ | $0.989_{\pm0.001}$ |
| default | $0.928_{\pm0.004}$ | $0.991_{\pm0.001}$ | $0.998_{\pm0.001}$ | $0.997_{\pm0.001}$ | $0.827_{\pm0.005}$ | $0.995_{\pm0.000}$ | $0.914_{\pm0.002}$ | $0.823_{\pm0.001}$ | $0.793_{\pm0.005}$ | $0.827_{\pm0.003}$ |
| diabetes | $0.726_{\pm0.001}$ | $0.854_{\pm0.002}$ | $0.935_{\pm0.002}$ | $0.997_{\pm0.001}$ | - | - | $0.946_{\pm0.001}$ | $0.864_{\pm0.002}$ | $0.837_{\pm0.001}$ | $0.862_{\pm0.001}$ |
| lending | $0.966_{\pm0.003}$ | $0.997_{\pm0.001}$ | $0.995_{\pm0.002}$ | $0.995_{\pm0.001}$ | $1.000_{\pm0.000}$ | - | $0.998_{\pm0.002}$ | $0.959_{\pm0.009}$ | $0.955_{\pm0.003}$ | $0.960_{\pm0.005}$ |
| news | $0.998_{\pm0.000}$ | $0.998_{\pm0.000}$ | $1.000_{\pm0.000}$ | $1.000_{\pm0.000}$ | $0.974_{\pm0.001}$ | $1.000_{\pm0.000}$ | $0.999_{\pm0.000}$ | $0.947_{\pm0.002}$ | $0.950_{\pm0.002}$ | $0.972_{\pm0.002}$ |
| nmes | $0.926_{\pm0.007}$ | $0.987_{\pm0.002}$ | $0.992_{\pm0.003}$ | $0.988_{\pm0.002}$ | $0.652_{\pm0.011}$ | $0.988_{\pm0.000}$ | $0.829_{\pm0.010}$ | $0.646_{\pm0.005}$ | $0.633_{\pm0.010}$ | $0.636_{\pm0.010}$ |

Table 16: Distance to closest record of the generated data (incl. standard errors in subscripts) for seven benchmark models and for CDTD with three different noise schedules.

| | Test Set | SMOTE | ARF | CTGAN | TVAE | TabDDPM | CoDi | TabSyn | CDTD (single) | CDTD (per type) | CDTD (per feature) |
|---|---|---|---|---|---|---|---|---|---|---|---|
| acsincome | $7.673_{\pm0.017}$ | - | $8.637_{\pm0.027}$ | $10.758_{\pm0.054}$ | $6.652_{\pm0.032}$ | - | $10.877_{\pm0.092}$ | $10.797_{\pm0.101}$ | $8.337_{\pm0.053}$ | $8.397_{\pm0.062}$ | $8.344_{\pm0.027}$ |
| adult | $1.870_{\pm0.000}$ | $1.371_{\pm0.013}$ | $2.523_{\pm0.012}$ | $5.012_{\pm0.028}$ | $2.227_{\pm0.013}$ | $1.656_{\pm0.008}$ | $2.735_{\pm0.028}$ | $2.408_{\pm0.031}$ | $1.138_{\pm0.015}$ | $1.327_{\pm0.015}$ | $1.334_{\pm0.012}$ |
| bank | $2.369_{\pm0.000}$ | $1.369_{\pm0.011}$ | $3.025_{\pm0.017}$ | $3.840_{\pm0.014}$ | $3.136_{\pm0.007}$ | $2.211_{\pm0.011}$ | $3.062_{\pm0.012}$ | $3.022_{\pm0.009}$ | $1.749_{\pm0.008}$ | $1.913_{\pm0.006}$ | $1.997_{\pm0.008}$ |
| beijing | $0.385_{\pm0.000}$ | $0.139_{\pm0.003}$ | $0.731_{\pm0.003}$ | $0.800_{\pm0.002}$ | $0.724_{\pm0.005}$ | $0.639_{\pm0.002}$ | $0.588_{\pm0.003}$ | $0.633_{\pm0.002}$ | $0.474_{\pm0.002}$ | $0.473_{\pm0.002}$ | $0.469_{\pm0.001}$ |
| churn | $0.347_{\pm0.000}$ | $0.232_{\pm0.028}$ | $1.136_{\pm0.015}$ | $1.804_{\pm0.036}$ | $1.146_{\pm0.039}$ | $0.368_{\pm0.044}$ | $0.852_{\pm0.016}$ | $1.209_{\pm0.010}$ | $0.329_{\pm0.010}$ | $0.278_{\pm0.012}$ | $0.350_{\pm0.023}$ |
| covertype | $0.529_{\pm0.001}$ | - | $5.773_{\pm0.017}$ | $3.173_{\pm0.013}$ | $1.508_{\pm0.020}$ | $2.593_{\pm0.020}$ | $3.033_{\pm0.012}$ | $1.741_{\pm0.011}$ | $1.825_{\pm0.016}$ | $1.594_{\pm0.009}$ | $1.805_{\pm0.009}$ |
| default | $1.812_{\pm0.000}$ | $1.032_{\pm0.010}$ | $3.095_{\pm0.026}$ | $5.880_{\pm0.020}$ | $3.216_{\pm0.013}$ | $1.437_{\pm0.020}$ | | $2.801_{\pm0.032}$ | $1.192_{\pm0.022}$ | $1.355_{\pm0.017}$ | $1.352_{\pm0.016}$ |
| diabetes | $15.608_{\pm0.055}$ | $13.909_{\pm0.050}$ | $17.736_{\pm0.107}$ | $21.935_{\pm0.046}$ | $8.214_{\pm0.022}$ | - | - | $28.794_{\pm0.054}$ | $14.356_{\pm0.050}$ | $14.468_{\pm0.028}$ | $14.866_{\pm0.033}$ |
| lending | $11.184_{\pm0.000}$ | $17.752_{\pm0.143}$ | $17.776_{\pm0.132}$ | $20.239_{\pm0.222}$ | $10.688_{\pm0.025}$ | $14.310_{\pm0.093}$ | - | $16.239_{\pm0.052}$ | $14.958_{\pm0.292}$ | $14.962_{\pm0.090}$ | $14.146_{\pm0.240}$ |
| news | $3.615_{\pm0.000}$ | $3.553_{\pm0.134}$ | $6.147_{\pm0.010}$ | $4.789_{\pm0.005}$ | $5.821_{\pm0.003}$ | $4.358_{\pm0.013}$ | $4.661_{\pm0.023}$ | $5.410_{\pm0.005}$ | $3.615_{\pm0.009}$ | $3.676_{\pm0.008}$ | $3.736_{\pm0.094}$ |
| nmes | $1.931_{\pm0.000}$ | $1.394_{\pm0.019}$ | $2.203_{\pm0.028}$ | $2.971_{\pm0.008}$ | $1.710_{\pm0.019}$ | $0.890_{\pm0.027}$ | $1.231_{\pm0.024}$ | $2.105_{\pm0.022}$ | $0.801_{\pm0.017}$ | $0.780_{\pm0.031}$ | $0.803_{\pm0.017}$ |

Table 17: Machine learning efficiency F1 score for seven benchmark models, the real training data and for CDTD with three different noise schedules. The standard deviation takes into account five different sampling seeds and uses the average results of the four machine learning efficiency models computed across ten model seeds.

| | Real Data | SMOTE | ARF | CTGAN | TVAE | TabDDPM | CoDi | TabSyn | CDTD (single) | CDTD (per type) | CDTD (per feature) |
|---|---|---|---|---|---|---|---|---|---|---|---|
| adult | $0.797_{\pm0.000}$ | $0.784_{\pm0.001}$ | $0.769_{\pm0.002}$ | $0.647_{\pm0.015}$ | $0.756_{\pm0.002}$ | $0.788_{\pm0.001}$ | $0.745_{\pm0.004}$ | $0.782_{\pm0.002}$ | $0.788_{\pm0.001}$ | $0.789_{\pm0.002}$ | $0.789_{\pm0.002}$ |
| bank | $0.745_{\pm0.002}$ | $0.740_{\pm0.004}$ | $0.682_{\pm0.006}$ | $0.680_{\pm0.006}$ | $0.629_{\pm0.006}$ | $0.744_{\pm0.005}$ | $0.673_{\pm0.006}$ | $0.661_{\pm0.008}$ | $0.782_{\pm0.003}$ | $0.766_{\pm0.005}$ | $0.747_{\pm0.004}$ |
| churn | $0.873_{\pm0.003}$ | $0.865_{\pm0.008}$ | $0.780_{\pm0.015}$ | $0.761_{\pm0.009}$ | $0.802_{\pm0.017}$ | $0.855_{\pm0.012}$ | $0.865_{\pm0.008}$ | $0.748_{\pm0.015}$ | $0.859_{\pm0.007}$ | $0.863_{\pm0.007}$ | $0.860_{\pm0.007}$ |
| covertype | $0.817_{\pm0.001}$ | - | $0.783_{\pm0.004}$ | $0.442_{\pm0.008}$ | $0.711_{\pm0.002}$ | $0.799_{\pm0.001}$ | $0.767_{\pm0.001}$ | $0.620_{\pm0.014}$ | $0.766_{\pm0.001}$ | $0.767_{\pm0.001}$ | $0.765_{\pm0.001}$ |
| default | $0.674_{\pm0.001}$ | $0.677_{\pm0.001}$ | $0.627_{\pm0.003}$ | $0.686_{\pm0.002}$ | $0.632_{\pm0.007}$ | $0.680_{\pm0.002}$ | $0.638_{\pm0.008}$ | $0.485_{\pm0.016}$ | $0.673_{\pm0.003}$ | $0.675_{\pm0.002}$ | $0.677_{\pm0.002}$ |
| diabetes | $0.621_{\pm0.002}$ | $0.615_{\pm0.002}$ | $0.572_{\pm0.005}$ | $0.557_{\pm0.004}$ | $0.553_{\pm0.003}$ | - | - | $0.566_{\pm0.006}$ | $0.614_{\pm0.003}$ | $0.619_{\pm0.002}$ | $0.614_{\pm0.003}$ |

Table 18: Machine learning efficiency AUC score for seven benchmark models, the real training data and for CDTD with three different noise schedules. The standard deviation takes into account five different sampling seeds and uses the average results of the four machine learning efficiency models computed across ten model seeds.

| | Real Data | SMOTE | ARF | CTGAN | TVAE | TabDDPM | CoDi | TabSyn | CDTD (single) | CDTD (per type) | CDTD (per feature) |
|---|---|---|---|---|---|---|---|---|---|---|---|
| adult | $0.915_{\pm0.000}$ | $0.906_{\pm0.001}$ | $0.901_{\pm0.000}$ | $0.836_{\pm0.006}$ | $0.889_{\pm0.002}$ | $0.909_{\pm0.000}$ | $0.880_{\pm0.005}$ | $0.906_{\pm0.001}$ | $0.909_{\pm0.000}$ | $0.909_{\pm0.001}$ | $0.908_{\pm0.001}$ |
| bank | $0.947_{\pm0.000}$ | $0.943_{\pm0.001}$ | $0.938_{\pm0.001}$ | $0.934_{\pm0.003}$ | $0.830_{\pm0.020}$ | $0.942_{\pm0.004}$ | $0.929_{\pm0.005}$ | $0.922_{\pm0.008}$ | $0.945_{\pm0.000}$ | $0.946_{\pm0.000}$ | $0.944_{\pm0.003}$ |
| churn | $0.964_{\pm0.001}$ | $0.961_{\pm0.002}$ | $0.939_{\pm0.007}$ | $0.882_{\pm0.006}$ | $0.948_{\pm0.004}$ | $0.957_{\pm0.006}$ | $0.961_{\pm0.001}$ | $0.911_{\pm0.013}$ | $0.960_{\pm0.001}$ | $0.957_{\pm0.008}$ | $0.960_{\pm0.003}$ |
| covertype | $0.892_{\pm0.000}$ | - | $0.860_{\pm0.001}$ | $0.677_{\pm0.007}$ | $0.777_{\pm0.001}$ | $0.876_{\pm0.001}$ | $0.845_{\pm0.001}$ | $0.675_{\pm0.013}$ | $0.840_{\pm0.001}$ | $0.844_{\pm0.000}$ | $0.840_{\pm0.001}$ |
| default | $0.768_{\pm0.000}$ | $0.759_{\pm0.003}$ | $0.754_{\pm0.002}$ | $0.744_{\pm0.002}$ | $0.751_{\pm0.004}$ | $0.765_{\pm0.002}$ | $0.739_{\pm0.008}$ | $0.732_{\pm0.021}$ | $0.763_{\pm0.000}$ | $0.764_{\pm0.001}$ | $0.765_{\pm0.002}$ |
| diabetes | $0.693_{\pm0.001}$ | $0.679_{\pm0.001}$ | $0.669_{\pm0.002}$ | $0.626_{\pm0.003}$ | $0.592_{\pm0.002}$ | - | - | $0.642_{\pm0.002}$ | $0.671_{\pm0.002}$ | $0.673_{\pm0.002}$ | $0.672_{\pm0.002}$ |

Table 19: Machine learning efficiency RMSE for seven benchmark models, the real training data and for CDTD with three different noise schedules. The standard deviation takes into account five different sampling seeds and uses the average results of the four machine learning efficiency models computed across ten model seeds.

| | Real Data | SMOTE | ARF | CTGAN | TVAE | TabDDPM | CoDi | TabSyn | CDTD (single) | CDTD (per type) | CDTD (per feature) |
|---|---|---|---|---|---|---|---|---|---|---|---|
| acsincome | $0.804_{\pm0.012}$ | - | $0.757_{\pm0.007}$ | $2.292_{\pm0.013}$ | $1.054_{\pm0.011}$ | - | $0.857_{\pm0.010}$ | $0.955_{\pm0.010}$ | $0.827_{\pm0.009}$ | $0.807_{\pm0.008}$ | $0.812_{\pm0.004}$ |
| beijing | $0.711_{\pm0.001}$ | $0.742_{\pm0.002}$ | $0.779_{\pm0.007}$ | $1.050_{\pm0.010}$ | $1.295_{\pm0.016}$ | $0.799_{\pm0.007}$ | $0.849_{\pm0.004}$ | $0.789_{\pm0.010}$ | $0.772_{\pm0.003}$ | $0.766_{\pm0.003}$ | $0.762_{\pm0.002}$ |
| lending | $0.030_{\pm0.000}$ | $0.042_{\pm0.001}$ | $0.274_{\pm0.007}$ | $0.137_{\pm0.007}$ | $0.404_{\pm0.007}$ | $0.789_{\pm0.033}$ | - | $0.305_{\pm0.006}$ | $0.071_{\pm0.001}$ | $0.066_{\pm0.002}$ | $0.062_{\pm0.002}$ |
| news | $1.001_{\pm0.002}$ | $1.180_{\pm0.107}$ | $0.923_{\pm0.052}$ | $1.906_{\pm0.019}$ | $3.999_{\pm0.175}$ | $0.171_{\pm0.006}$ | $1.302_{\pm0.074}$ | $0.397_{\pm0.037}$ | $0.848_{\pm0.081}$ | $0.752_{\pm0.067}$ | $0.717_{\pm0.045}$ |
| nmes | $1.001_{\pm0.003}$ | $1.112_{\pm0.044}$ | $0.972_{\pm0.024}$ | $1.331_{\pm0.052}$ | $1.127_{\pm0.047}$ | $1.200_{\pm0.054}$ | $1.137_{\pm0.052}$ | $0.563_{\pm0.005}$ | $1.154_{\pm0.052}$ | $1.109_{\pm0.055}$ | $1.136_{\pm0.074}$ |

# T    ABLATION STUDY DETAILS

Table 20: $L_2$ norm (incl. standard errors in subscripts) of the correlation matrix differences of real and synthetic train sets for five CDTD configurations with progressive addition of model components.

| Configuration | A | B | C | D | CDTD (per type) |
|---|---|---|---|---|---|
| acsincome | $0.138_{\pm 0.004}$ | $0.138_{\pm 0.003}$ | $0.135_{\pm 0.001}$ | $0.152_{\pm 0.001}$ | $0.135_{\pm 0.002}$ |
| adult | $0.135_{\pm 0.006}$ | $0.113_{\pm 0.004}$ | $0.159_{\pm 0.015}$ | $0.100_{\pm 0.010}$ | $0.125_{\pm 0.005}$ |
| bank | $0.478_{\pm 0.019}$ | $0.243_{\pm 0.015}$ | $0.269_{\pm 0.012}$ | $0.196_{\pm 0.012}$ | $0.231_{\pm 0.009}$ |
| beijing | $0.076_{\pm 0.009}$ | $0.070_{\pm 0.007}$ | $0.067_{\pm 0.004}$ | $0.068_{\pm 0.006}$ | $0.073_{\pm 0.007}$ |
| churn | $0.294_{\pm 0.053}$ | $0.280_{\pm 0.043}$ | $0.262_{\pm 0.030}$ | $0.239_{\pm 0.038}$ | $0.255_{\pm 0.021}$ |
| covertype | $1.178_{\pm 0.193}$ | $2.099_{\pm 0.010}$ | $1.974_{\pm 0.136}$ | $1.808_{\pm 0.010}$ | $1.357_{\pm 0.155}$ |
| default | $0.905_{\pm 0.110}$ | $0.799_{\pm 0.123}$ | $0.727_{\pm 0.132}$ | $0.521_{\pm 0.124}$ | $0.641_{\pm 0.130}$ |
| diabetes | $0.719_{\pm 0.049}$ | $1.435_{\pm 0.021}$ | $1.397_{\pm 0.005}$ | $1.230_{\pm 0.061}$ | $1.213_{\pm 0.029}$ |
| lending | $1.480_{\pm 0.046}$ | $1.127_{\pm 0.083}$ | $1.178_{\pm 0.059}$ | $1.295_{\pm 0.044}$ | $1.239_{\pm 0.090}$ |
| news | $2.484_{\pm 0.138}$ | $2.136_{\pm 0.417}$ | $1.973_{\pm 0.417}$ | $2.016_{\pm 0.483}$ | $1.811_{\pm 0.295}$ |
| nmes | $0.483_{\pm 0.048}$ | $0.421_{\pm 0.032}$ | $0.457_{\pm 0.041}$ | $0.450_{\pm 0.041}$ | $0.444_{\pm 0.075}$ |

Table 21: Jensen-Shannon divergence (incl. standard errors in subscripts) for five CDTD configurations with progressive addition of model components.

| Configuration | A | B | C | D | CDTD (per type) |
|---|---|---|---|---|---|
| acsincome | $0.016_{\pm 0.000}$ | $0.022_{\pm 0.000}$ | $0.026_{\pm 0.001}$ | $0.032_{\pm 0.001}$ | $0.022_{\pm 0.001}$ |
| adult | $0.010_{\pm 0.001}$ | $0.013_{\pm 0.001}$ | $0.013_{\pm 0.001}$ | $0.015_{\pm 0.001}$ | $0.015_{\pm 0.001}$ |
| bank | $0.008_{\pm 0.000}$ | $0.017_{\pm 0.000}$ | $0.011_{\pm 0.001}$ | $0.011_{\pm 0.001}$ | $0.011_{\pm 0.001}$ |
| beijing | $0.005_{\pm 0.001}$ | $0.005_{\pm 0.004}$ | $0.004_{\pm 0.002}$ | $0.004_{\pm 0.001}$ | $0.006_{\pm 0.001}$ |
| churn | $0.015_{\pm 0.003}$ | $0.011_{\pm 0.002}$ | $0.009_{\pm 0.003}$ | $0.010_{\pm 0.003}$ | $0.010_{\pm 0.002}$ |
| covertype | $0.002_{\pm 0.000}$ | $0.003_{\pm 0.000}$ | $0.004_{\pm 0.000}$ | $0.004_{\pm 0.000}$ | $0.003_{\pm 0.000}$ |
| default | $0.014_{\pm 0.001}$ | $0.015_{\pm 0.001}$ | $0.012_{\pm 0.002}$ | $0.017_{\pm 0.002}$ | $0.016_{\pm 0.001}$ |
| diabetes | $0.024_{\pm 0.000}$ | $0.030_{\pm 0.000}$ | $0.022_{\pm 0.000}$ | $0.024_{\pm 0.001}$ | $0.025_{\pm 0.000}$ |
| lending | $0.055_{\pm 0.001}$ | $0.055_{\pm 0.002}$ | $0.055_{\pm 0.001}$ | $0.057_{\pm 0.001}$ | $0.057_{\pm 0.001}$ |
| news | $0.003_{\pm 0.001}$ | $0.003_{\pm 0.001}$ | $0.003_{\pm 0.001}$ | $0.004_{\pm 0.001}$ | $0.003_{\pm 0.001}$ |
| nmes | $0.008_{\pm 0.002}$ | $0.008_{\pm 0.002}$ | $0.008_{\pm 0.002}$ | $0.009_{\pm 0.001}$ | $0.007_{\pm 0.002}$ |

Table 22: Wasserstein distance (incl. standard errors in subscripts) for five CDTD configurations with progressive addition of model components.

| Configuration | A | B | C | D | CDTD (per type) |
|---|---|---|---|---|---|
| acsincome | $0.005_{\pm 0.000}$ | $0.002_{\pm 0.000}$ | $0.002_{\pm 0.000}$ | $0.001_{\pm 0.000}$ | $0.001_{\pm 0.000}$ |
| adult | $0.008_{\pm 0.000}$ | $0.005_{\pm 0.000}$ | $0.006_{\pm 0.000}$ | $0.002_{\pm 0.000}$ | $0.003_{\pm 0.000}$ |
| bank | $0.007_{\pm 0.000}$ | $0.004_{\pm 0.000}$ | $0.005_{\pm 0.001}$ | $0.003_{\pm 0.000}$ | $0.003_{\pm 0.001}$ |
| beijing | $0.006_{\pm 0.000}$ | $0.004_{\pm 0.000}$ | $0.002_{\pm 0.000}$ | $0.002_{\pm 0.000}$ | $0.002_{\pm 0.000}$ |
| churn | $0.008_{\pm 0.001}$ | $0.007_{\pm 0.001}$ | $0.007_{\pm 0.001}$ | $0.007_{\pm 0.002}$ | $0.006_{\pm 0.001}$ |
| covertype | $0.003_{\pm 0.000}$ | $0.020_{\pm 0.000}$ | $0.019_{\pm 0.000}$ | $0.012_{\pm 0.000}$ | $0.009_{\pm 0.000}$ |
| default | $0.003_{\pm 0.000}$ | $0.004_{\pm 0.000}$ | $0.004_{\pm 0.000}$ | $0.003_{\pm 0.000}$ | $0.003_{\pm 0.000}$ |
| diabetes | $0.021_{\pm 0.000}$ | $0.042_{\pm 0.000}$ | $0.041_{\pm 0.000}$ | $0.033_{\pm 0.000}$ | $0.034_{\pm 0.000}$ |
| lending | $0.012_{\pm 0.000}$ | $0.011_{\pm 0.000}$ | $0.012_{\pm 0.000}$ | $0.013_{\pm 0.000}$ | $0.013_{\pm 0.000}$ |
| news | $0.009_{\pm 0.000}$ | $0.006_{\pm 0.000}$ | $0.007_{\pm 0.000}$ | $0.005_{\pm 0.000}$ | $0.006_{\pm 0.000}$ |
| nmes | $0.008_{\pm 0.001}$ | $0.007_{\pm 0.001}$ | $0.008_{\pm 0.000}$ | $0.008_{\pm 0.001}$ | $0.006_{\pm 0.000}$ |

Table 23: Detection score (incl. standard errors in subscripts) for five CDTD configurations with progressive addition of model components.

| Configuration | A | B | C | D | CDTD (per type) |
|---|---|---|---|---|---|
| acsincome | $0.547_{\pm 0.001}$ | $0.533_{\pm 0.003}$ | $0.540_{\pm 0.002}$ | $0.546_{\pm 0.003}$ | $0.527_{\pm 0.002}$ |
| adult | $0.640_{\pm 0.003}$ | $0.595_{\pm 0.002}$ | $0.621_{\pm 0.002}$ | $0.581_{\pm 0.002}$ | $0.590_{\pm 0.004}$ |
| bank | $0.880_{\pm 0.001}$ | $0.739_{\pm 0.006}$ | $0.798_{\pm 0.001}$ | $0.662_{\pm 0.005}$ | $0.701_{\pm 0.005}$ |
| beijing | $0.699_{\pm 0.003}$ | $0.658_{\pm 0.004}$ | $0.615_{\pm 0.002}$ | $0.626_{\pm 0.002}$ | $0.620_{\pm 0.005}$ |
| churn | $0.710_{\pm 0.008}$ | $0.610_{\pm 0.006}$ | $0.580_{\pm 0.011}$ | $0.560_{\pm 0.019}$ | $0.541_{\pm 0.008}$ |
| covertype | $0.887_{\pm 0.002}$ | $0.991_{\pm 0.001}$ | $0.989_{\pm 0.001}$ | $0.984_{\pm 0.001}$ | $0.981_{\pm 0.000}$ |
| default | $0.925_{\pm 0.002}$ | $0.816_{\pm 0.003}$ | $0.774_{\pm 0.003}$ | $0.759_{\pm 0.004}$ | $0.793_{\pm 0.005}$ |
| diabetes | $0.762_{\pm 0.001}$ | $0.895_{\pm 0.001}$ | $0.870_{\pm 0.002}$ | $0.851_{\pm 0.001}$ | $0.837_{\pm 0.001}$ |
| lending | $0.989_{\pm 0.002}$ | $0.938_{\pm 0.011}$ | $0.958_{\pm 0.003}$ | $0.944_{\pm 0.005}$ | $0.955_{\pm 0.003}$ |
| news | $0.993_{\pm 0.001}$ | $0.956_{\pm 0.002}$ | $0.965_{\pm 0.002}$ | $0.946_{\pm 0.002}$ | $0.950_{\pm 0.002}$ |
| nmes | $0.663_{\pm 0.014}$ | $0.651_{\pm 0.007}$ | $0.667_{\pm 0.009}$ | $0.648_{\pm 0.015}$ | $0.633_{\pm 0.010}$ |

Table 24: Distance to closest record of the generated data (incl. standard errors in subscripts) for five CDTD configurations with progressive addition of model components.

| | Real Test Set | A | B | C | D | CDTD (per type) |
|---|---|---|---|---|---|---|
| acsincome | $7.673_{\pm 0.017}$ | $8.550_{\pm 0.055}$ | $8.342_{\pm 0.037}$ | $8.284_{\pm 0.030}$ | $8.311_{\pm 0.041}$ | $8.397_{\pm 0.062}$ |
| adult | $1.870_{\pm 0.000}$ | $1.231_{\pm 0.016}$ | $1.318_{\pm 0.009}$ | $1.296_{\pm 0.012}$ | $1.520_{\pm 0.011}$ | $1.327_{\pm 0.015}$ |
| bank | $2.369_{\pm 0.000}$ | $1.583_{\pm 0.006}$ | $2.016_{\pm 0.007}$ | $2.069_{\pm 0.014}$ | $2.187_{\pm 0.009}$ | $1.913_{\pm 0.006}$ |
| beijing | $0.385_{\pm 0.000}$ | $0.592_{\pm 0.001}$ | $0.537_{\pm 0.002}$ | $0.505_{\pm 0.001}$ | $0.511_{\pm 0.001}$ | $0.473_{\pm 0.002}$ |
| churn | $0.347_{\pm 0.000}$ | $0.526_{\pm 0.014}$ | $0.383_{\pm 0.014}$ | $0.327_{\pm 0.022}$ | $0.333_{\pm 0.020}$ | $0.278_{\pm 0.012}$ |
| covertype | $0.529_{\pm 0.001}$ | $0.871_{\pm 0.009}$ | $1.919_{\pm 0.016}$ | $1.872_{\pm 0.022}$ | $1.698_{\pm 0.024}$ | $1.594_{\pm 0.009}$ |
| default | $1.812_{\pm 0.000}$ | $1.321_{\pm 0.015}$ | $1.333_{\pm 0.021}$ | $1.306_{\pm 0.020}$ | $1.484_{\pm 0.012}$ | $1.355_{\pm 0.017}$ |
| diabetes | $15.608_{\pm 0.055}$ | $13.299_{\pm 0.020}$ | $14.958_{\pm 0.029}$ | $15.007_{\pm 0.026}$ | $14.755_{\pm 0.045}$ | $14.468_{\pm 0.028}$ |
| lending | $11.184_{\pm 0.000}$ | $15.130_{\pm 0.320}$ | $15.112_{\pm 0.286}$ | $14.957_{\pm 0.149}$ | $14.916_{\pm 0.196}$ | $14.962_{\pm 0.090}$ |
| news | $3.615_{\pm 0.000}$ | $3.894_{\pm 0.016}$ | $3.737_{\pm 0.012}$ | $3.692_{\pm 0.014}$ | $3.746_{\pm 0.010}$ | $3.676_{\pm 0.008}$ |
| nmes | $1.931_{\pm 0.000}$ | $1.202_{\pm 0.019}$ | $1.014_{\pm 0.013}$ | $0.958_{\pm 0.008}$ | $0.952_{\pm 0.022}$ | $0.780_{\pm 0.031}$ |

Table 25: Machine learning efficiency F1 score for five CDTD configurations with progressive addition of model components. The standard deviation accounts for five different sampling seeds and uses the average results of the four machine learning efficiency models across ten model seeds.

| | Real Data | A | B | C | D | CDTD (per type) |
|---|---|---|---|---|---|---|
| adult | $0.797_{\pm 0.000}$ | $0.780_{\pm 0.002}$ | $0.788_{\pm 0.001}$ | $0.786_{\pm 0.001}$ | $0.790_{\pm 0.001}$ | $0.789_{\pm 0.002}$ |
| bank | $0.745_{\pm 0.002}$ | $0.759_{\pm 0.004}$ | $0.758_{\pm 0.006}$ | $0.760_{\pm 0.005}$ | $0.751_{\pm 0.005}$ | $0.766_{\pm 0.005}$ |
| churn | $0.873_{\pm 0.003}$ | $0.850_{\pm 0.006}$ | $0.861_{\pm 0.008}$ | $0.863_{\pm 0.004}$ | $0.860_{\pm 0.008}$ | $0.863_{\pm 0.007}$ |
| covertype | $0.817_{\pm 0.001}$ | $0.791_{\pm 0.001}$ | $0.745_{\pm 0.001}$ | $0.761_{\pm 0.001}$ | $0.768_{\pm 0.001}$ | $0.767_{\pm 0.001}$ |
| default | $0.674_{\pm 0.001}$ | $0.672_{\pm 0.002}$ | $0.674_{\pm 0.002}$ | $0.671_{\pm 0.002}$ | $0.673_{\pm 0.003}$ | $0.675_{\pm 0.002}$ |
| diabetes | $0.621_{\pm 0.002}$ | $0.616_{\pm 0.002}$ | $0.612_{\pm 0.003}$ | $0.612_{\pm 0.003}$ | $0.617_{\pm 0.002}$ | $0.619_{\pm 0.002}$ |

Table 27: Machine learning efficiency RMSE for five CDTD configurations with progressive addition of model components. The standard deviation accounts for five different sampling seeds and uses the average results of the four machine learning efficiency models across ten model seeds.

| | Real Data | A | B | C | D | CDTD (per type) |
|---|---|---|---|---|---|---|
| acsincome | $0.804_{\pm 0.012}$ | $0.868_{\pm 0.011}$ | $0.808_{\pm 0.014}$ | $0.820_{\pm 0.012}$ | $0.800_{\pm 0.011}$ | $0.807_{\pm 0.008}$ |
| beijing | $0.711_{\pm 0.001}$ | $0.801_{\pm 0.006}$ | $0.780_{\pm 0.005}$ | $0.771_{\pm 0.005}$ | $0.769_{\pm 0.006}$ | $0.766_{\pm 0.003}$ |
| lending | $0.030_{\pm 0.000}$ | $0.124_{\pm 0.006}$ | $0.059_{\pm 0.002}$ | $0.072_{\pm 0.001}$ | $0.067_{\pm 0.002}$ | $0.066_{\pm 0.002}$ |
| news | $1.001_{\pm 0.002}$ | $0.772_{\pm 0.019}$ | $0.835_{\pm 0.062}$ | $0.805_{\pm 0.079}$ | $0.763_{\pm 0.062}$ | $0.752_{\pm 0.067}$ |
| nmes | $1.001_{\pm 0.003}$ | $0.967_{\pm 0.064}$ | $1.128_{\pm 0.088}$ | $1.195_{\pm 0.087}$ | $1.225_{\pm 0.070}$ | $1.109_{\pm 0.055}$ |

Table 26: Machine learning efficiency AUC score for five CDTD configurations with progressive addition of model components. The standard deviation accounts for five different sampling seeds and uses the average results of the four machine learning efficiency models across ten model seeds.

| | Real Data | A | B | C | D | CDTD (per type) |
|---|---|---|---|---|---|---|
| adult | $0.915_{\pm 0.000}$ | $0.906_{\pm 0.001}$ | $0.909_{\pm 0.000}$ | $0.909_{\pm 0.000}$ | $0.909_{\pm 0.001}$ | $0.909_{\pm 0.001}$ |
| bank | $0.947_{\pm 0.000}$ | $0.945_{\pm 0.003}$ | $0.946_{\pm 0.002}$ | $0.945_{\pm 0.002}$ | $0.946_{\pm 0.000}$ | $0.946_{\pm 0.000}$ |
| churn | $0.964_{\pm 0.001}$ | $0.959_{\pm 0.003}$ | $0.961_{\pm 0.002}$ | $0.962_{\pm 0.002}$ | $0.960_{\pm 0.002}$ | $0.957_{\pm 0.008}$ |
| covertype | $0.892_{\pm 0.000}$ | $0.870_{\pm 0.000}$ | $0.826_{\pm 0.001}$ | $0.838_{\pm 0.001}$ | $0.844_{\pm 0.001}$ | $0.844_{\pm 0.000}$ |
| default | $0.768_{\pm 0.000}$ | $0.763_{\pm 0.002}$ | $0.764_{\pm 0.001}$ | $0.764_{\pm 0.002}$ | $0.764_{\pm 0.002}$ | $0.764_{\pm 0.002}$ |
| diabetes | $0.693_{\pm 0.001}$ | $0.675_{\pm 0.001}$ | $0.664_{\pm 0.001}$ | $0.671_{\pm 0.001}$ | $0.673_{\pm 0.001}$ | $0.673_{\pm 0.002}$ |

## U  TRAINING AND SAMPLING TIMES DETAILS

Table 28: Training times in minutes. TabDDPM produces NaNs during training on `acsincome` and `diabetes`, and is therefore excluded for these data. CoDi is considered prohibitively expensive to train on `diabetes` and `lending` and we report estimated (est.) training times.

| | SMOTE | ARF | CTGAN | TVAE | TabDDPM | CoDi | TabSyn | CDTD (per feature) |
|---|---|---|---|---|---|---|---|---|
| acsincome | - | 80.3 | 59.9 | 26.0 | - | 231.9 | 13.4 | 5.8 |
| adult | - | 7.4 | 36.2 | 23.7 | 38.3 | 48.3 | 32.7 | 6.9 |
| bank | - | 11.0 | 37.6 | 24.6 | 40.5 | 42.7 | 48.5 | 26.3 |
| beijing | - | 3.7 | 34.3 | 23.9 | 36.1 | 24.9 | 25.8 | 23.4 |
| churn | - | 0.3 | 27.1 | 13.7 | 18.2 | 25.7 | 21.5 | 6.1 |
| covertype | - | 130.2 | 58.0 | 36.5 | 44.9 | 69.2 | 30.7 | 28.2 |
| default | - | 12.0 | 38.3 | 24.8 | 38.9 | 45.9 | 40.1 | 26.4 |
| diabetes | - | 58.5 | 90.1 | 25.3 | - | 870 (est.) | 34.6 | 26.9 |
| lending | - | 5.2 | 157.9 | 36.6 | 48.7 | 3000 (est.) | 42.1 | 25.3 |
| news | - | 23.0 | 48.8 | 33.3 | 37.2 | 41.5 | 57.9 | 25.2 |
| nmes | - | 0.4 | 32.8 | 17.2 | 24.9 | 30.2 | 31.0 | 6.3 |

Table 29: Sample times in seconds per 1000 samples. TabDDPM produces NaNs during training on `acsincome` and `diabetes`, and is therefore excluded for these data. CoDi is considered prohibitively expensive to train on `diabetes` and `lending`.

| | SMOTE | ARF | CTGAN | TVAE | TabDDPM | CoDi | TabSyn | CDTD (per feature) |
|---|---|---|---|---|---|---|---|---|
| acsincome | 4674.45 | 4.20 | 0.23 | 0.07 | - | 10.26 | 3.53 | 0.59 |
| adult | 10.71 | 1.78 | 0.31 | 0.16 | 0.82 | 3.65 | 0.88 | 0.56 |
| bank | 16.19 | 2.24 | 0.44 | 0.44 | 0.87 | 3.38 | 0.80 | 0.64 |
| beijing | 3.98 | 0.34 | 0.41 | 0.32 | 2.09 | 2.45 | 0.99 | 0.26 |
| churn | 0.52 | 1.00 | 0.40 | 0.24 | 0.95 | 2.78 | 0.80 | 0.39 |
| covertype | 10913.34 | 9.74 | 0.28 | 0.25 | 2.45 | 4.35 | 0.85 | 1.97 |
| default | 10.00 | 2.07 | 0.27 | 0.25 | 0.86 | 3.48 | 0.82 | 0.60 |
| diabetes | 166.75 | 5.87 | 0.53 | 0.15 | - | - | 0.83 | 1.33 |
| lending | 4.06 | 2.49 | 0.45 | 0.54 | 4.33 | - | 0.85 | 0.69 |
| news | 66.49 | 3.89 | 0.43 | 0.30 | 5.13 | 2.93 | 0.86 | 0.85 |
| nmes | 0.69 | 1.54 | 0.31 | 0.17 | 4.17 | 2.91 | 0.82 | 0.55 |

Figures 24 and 25 show the benefit of deep generative models over SMOTE. Even though SMOTE is often praised as a simple, easy-to-use oversampling tool for tabular data, it relies on identifying nearest neighbors, making sampling very inefficient for larger datasets. As a consequence, we deem SMOTE to be infeasible to use for the `acsincome` and `covertype` datasets. The figures also illustrate the performance edge of CDTD, in particular compared to other diffusion-based models.

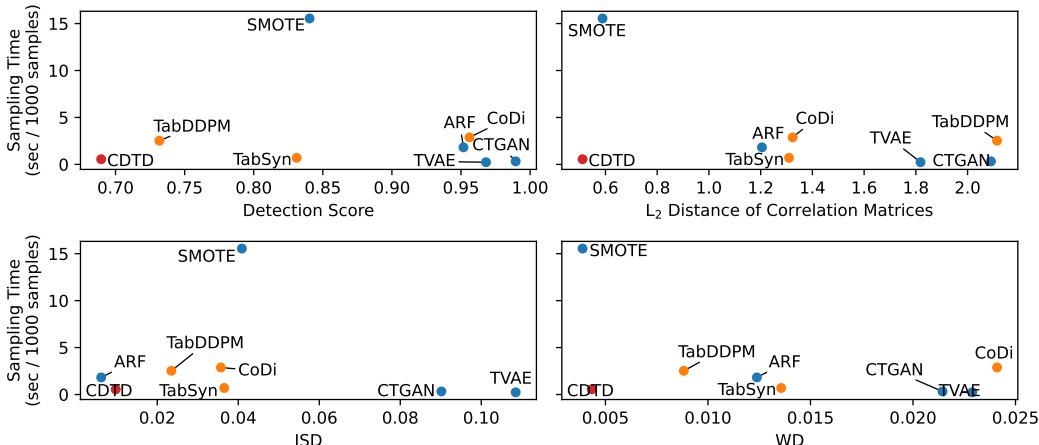

Figure 24: Average sample quality metrics as a function of sampling time. Diffusion-based models are indicated in orange. Only datasets for which we could retrieve results from all models are included, this excludes `acsincome`, `covertype`, `diabetes`, `lending`.

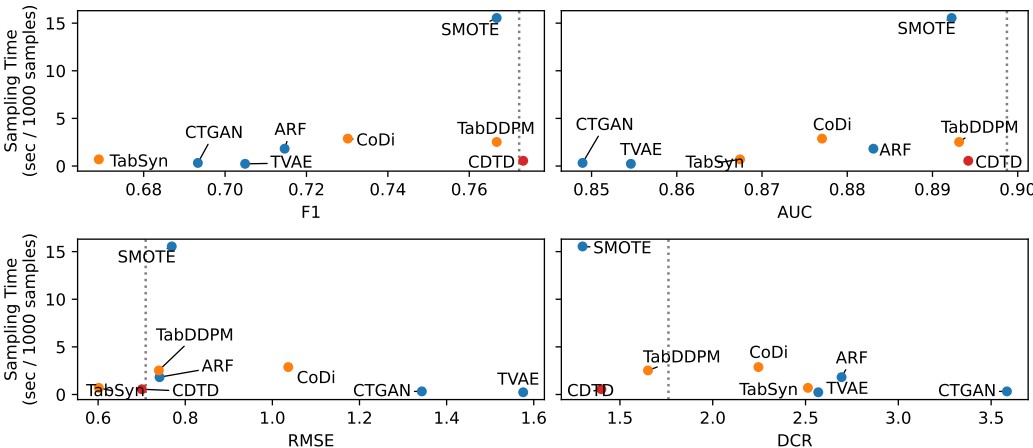

Figure 25: Average ML efficiency metrics and DCR as a function of sampling time. Diffusion-based models are indicated in orange. The dotted line indicates the test set performance of the real data. Only datasets for which we could retrieve results from all models are included, this excludes `acsincome`, `covertype`, `diabetes`, `lending`.

