# OpenReview forum: "Continuous Diffusion for Mixed-Type Tabular Data"
_ICLR.cc/2025/Conference — ICLR 2025 Poster_

### Official Review · Reviewer_697T · 2024-10-28

**Soundness:** 3
**Presentation:** 3
**Contribution:** 2
**Rating:** 6
**Confidence:** 3

**Summary:**

This paper trains a diffusion model for mixed-type (continuous and categorical) tabular data. The authors combine score matching with score interpolation to derive a diffusion model that add noise of categorical data in the embedding space, which makes it able to apply denoising process to mixed-type tabular data. To deal with the heterogenity of different types and features of data, the author further use noise calibration to balance the scale of continuous and categorical data. They also experimented on different types of noise schedules for better model training. The authors performed experiments and ablation studies on multiple datasets and used different metrics, the results showed that their model was able to achieve great performance on mixed-type tabular data.

**Strengths:**

- The authors proposed the first model to allow diffusion models to train on mixed-type tabular data, and allows the use of advanced techniques such as classifier-free guidance in training.
- To allow training on mixed-type tabular data, the authors have introduced several customizations to diffusion models: such as loss balancing, and noise schedules.
- The experiments and comparisons look quite complete for me: the authors compared with seven methods on eleven datasets, using different kinds of metrics. The analysis of the results are also quite good. Results have shown that their method can achieve good results and higher efficiency.
- The authors have provided details of all the implementation and experiments, they also provided their code in the supplementary. I believe this is quite useful for further researchers.

**Weaknesses:**

- The paper is more like a combination of existing models: diffusion models have already been applied to both categorical data and continuous data before. The authors combine them to deal with mixed-type data.
- The introduced changes: reweighting losses and changing noise schedules, although useful, but I think they are a bit minor to be counted as big technical contributions. The main reason is that they are a bit straightforward and also limited to current data setting. I am also curious that whether the noise schedules are also useful for data beyond mixed-type tabular data.

**Questions:**

Please see weaknesses part. I would like to discuss with the authors on those points.

---

> ### Author Response · Authors · 2024-11-14
>
> Thank you for your valuable comments and kind words about our work. Below we address specific questions.
>
> > The paper is more like a combination of existing models: diffusion models have already been applied to both categorical data and continuous data before. The authors combine them to deal with mixed-type data.
>
> Score matching and score interpolation have been *separately* applied before to generate continuous data and text (categorical data), respectively. The inherent difficulty in modeling *mixed-type* data, however, lies in **efficiently combining distinct diffusion processes**. In this regard, **we make several substantial contributions and improvements** over previous work on tabular data generation (see also Appendix N):
>
> - we ensure that the diffusion models for continuous and categorical data both use Gaussian noise. Previous diffusion models for tabular data relied on discrete multinomial diffusion for categorical features (e.g. [1], [2]).
> - we balance model capacity using adaptive and in particular type- or feature-specific noise schedules. This approach is unique to the mixed-type data setting and has not been explored so far.
> - we propose a novel loss calibration specific to the diffusion model context to ensure all features are treated equally. **No other diffusion model for mixed-type data addresses the balancing issue** arising from implicitly weighting the importance of some feature higher due to different loss scales.
>
>
> > reweighting losses and changing noise schedules, although useful, but I think they are a bit minor to be counted as big technical contributions. The main reason is that they are a bit straightforward and also limited to current data setting.
>
> First, we want to politely disagree with the notion that the fact that a model or design choice is straightforward automatically diminishes its contribution. The **models most impactful in practice are often simple in nature**. In fact, Reviewer cZJV mentions the simplicity of our method as a strength. We agree that each model component on its own is 'easy to understand'. However, we also show that **it is the *combination* of our design choices that lead to SOTA results** on a very diverse set of tabular data and benchmark models. Lastly, we agree that our results are only valid for the mixed-type tabular data setting. We make this scope of our work clear not only in the introduction of our work but also in the title. We also believe that the **domain of tabular data is extremely important to a large part of the general scientific community**, while receiving only limited attention in (generative) AI/ML.
>
> ---
>
> [1] Kotelnikov, Akim, et al. "Tabddpm: Modelling tabular data with diffusion models." International Conference on Machine Learning. PMLR, 2023.
>
> [2] Chaejeong Lee, et al. "CoDi: Co-evolving Contrastive Diffusion Models
> for Mixed-type Tabular Synthesis." In Proceedings of the 40th International Conference on Machine Learning. PMLR, 2023.

---

> ### Author Response · Authors · 2024-11-21
>
> Dear Reviewer,
>
> Thank you again for reviewing our paper. We have tried our best to address your questions (see our rebuttal in the top-level comment and above), and revised our paper by following suggestions from all reviewers.
>
> Please kindly let us know if you have any follow-up questions or areas needing further clarification. Your insights are valuable to us, and we stand ready to provide any additional information that could be helpful. Please also **consider raising your score** if you think our questions addressed your initial concerns sufficiently.

---

> > ### Comment · Reviewer_697T · 2024-11-21
> >
> > Thanks the authors for the rebuttal. While I do acknowledge the authors efforts in developing the first diffusion model that unifies continuous and discrete tabular data and achieve the SOTA results, I still think these contributions are not enough for a higher score as these efforts are mainly done for the tabular data. I do agree that tabular data itself are important for the scientific community, but the techniques here are not for a broader range of machine learning community. I am more inclined to maintain my positive score or slightly change it to 7 (not available currently).

---

### Official Review · Reviewer_cZJV · 2024-10-31

**Soundness:** 2
**Presentation:** 2
**Contribution:** 2
**Rating:** 5
**Confidence:** 3

**Summary:**

This paper combines score matching and score interpolation to ensure a common continuous noise distribution for both continuous and categorical features in tabular data generation. The proposed method focuses on adjusting noise schedules to enhance performance in generating synthetic tabular data.

**Strengths:**

1. Superior Performance: The results reported in the paper demonstrate strong performance on existing datasets, showing significant improvements compared to baseline methods.

2. Simplicity and Clarity: The methodology centers on adjusting noise schedules, which is straightforward and easy to understand, making the approach accessible and replicable.

**Weaknesses:**

1. Limited Novelty: The primary method introduced is Adaptive Noise Schedules, an area that has been extensively explored in prior research. Additionally, the paper addresses the homogenization of different data types—a problem that has been widely studied in previous tabular data generation works [1],[2]. Consequently, the novelty of this paper is significantly diminished, and the authors need to clearly articulate the innovative aspects of their approach.

2. Insufficient Experimental Reporting: The experimental results are not thoroughly reported. The authors compare their method with baseline approaches using only an average metric across various datasets, which leads to the loss of detailed information. To provide a more comprehensive evaluation, the authors should conduct detailed comparisons on each individual dataset, referencing methodologies from previous studies [1].

[1] Zhang, Hengrui, et al. "Mixed-type tabular data synthesis with score-based diffusion in latent space." arXiv preprint arXiv:2310.09656 (2023).

[2] Kotelnikov, Akim, et al. "Tabddpm: Modelling tabular data with diffusion models." International Conference on Machine Learning. PMLR, 2023.

**Questions:**

Please see the weakness part

---

> ### Author Response · Authors · 2024-11-14
>
> Thank you very much for the detailed review. Below we address specific questions.
>
> > The paper addresses the homogenization of different data types—a problem that has been widely studied in previous tabular data generation works [1],[2]. Consequently, the novelty of this paper is significantly diminished, and the authors need to clearly articulate the innovative aspects of their approach.
>
> We **politely disagree**. We do articulate the innovative aspects. In Appendix N, we compare our work to other diffusion-based generative models for tabular data, with an emphasis on the differences in the modeling approaches. **[2] does *not* address the problem of homogenization** of different data types at all. They simply combine two different diffusion processes without any thought about homogenization. For instance, they use a discrete diffusion process for categorical features which makes the noise schedules for continuous and categorical features irreconcilable. [1] address the homogenization somewhat as they encode all data into a latent space by training a separate VAE-based encoder. However, they **([1]) do *not* discuss potential implicit importance weights** for different features at all, which would require some kind of loss balancing for the VAE-based encoder (see [3] that show this issue for VAEs).
>
> **We are the first paper on tabular data that focuses on homogenizing data types effectively**. To this end, we propose a novel loss calibration, specific to tabular data and the diffusion generative framework. We are the first to ensure that the diffusion processes for both feature types rely *both* on Gaussian noise, such that we can balance their noise schedules effectively. Besides that, in the tabular data domain **the CDTD model is the first to show that adaptive and type- or feature-specific noise schedules improve sample quality and yield SOTA performance**.
>
>
> > Insufficient Experimental Reporting: The experimental results are not thoroughly reported. The authors compare their method with baseline approaches using only an average metric across various datasets, which leads to the loss of detailed information.
>
> **We disagree as we report *all* detailed results in Appendix R**. Our main results in Table 1 are only a means of summarizing the more detailed results. Since we benchmark on 11 datasets, against 7 benchmark models and evaluating on 8 different metrics, putting the detailed results in the main part of the paper is simply not feasible due to the page limit. The mentioned paper [1] only considers 6 datasets but still needs to resort to reporting some metrics only in the appendix, for instance, see Table 9-11 in the Appendix of [1]. We want to emphasize that in the tabular data domain the focus is much more on the average expected performance than the performance on a specific dataset. When applying our model in practice, knowing it does well on average across a diverse set of benchmark datasets is more valuable than knowing that it does well on a specific dataset which may or may not be similar to the dataset at hand.
>
> Note also that Table 1 does not show the average *metric* but rather the average *rank* of each model across all datasets based on the respective metrics (averaged over multiple seeds, similar to [1]). To be specific, for each dataset we retrieve the average (over multiple seeds) metric for each model. We then assign a rank to each model based on this average. We do so for all datasets and report the average rank across all datasets in Table 1. Hence, a lower rank in Table 1 for CDTD implies that considering its average performance over seeds it outperforms a model with a higher rank on average over all datasets.
>
> ---
>
> [1] Zhang, Hengrui, et al. "Mixed-type tabular data synthesis with score-based diffusion in latent space." arXiv preprint arXiv:2310.09656 (2023).
>
> [2] Kotelnikov, Akim, et al. "Tabddpm: Modelling tabular data with diffusion models." International Conference on Machine Learning. PMLR, 2023.
>
> [3] Chao Ma, et al. "VAEM: A Deep Generative Model for Heterogeneous Mixed Type Data." In Advances in Neural Information Processing Systems, volume 33, pp. 11237–11247. Curran Associates, Inc., 2020.

---

> ### Author Response · Authors · 2024-11-21
>
> Dear Reviewer,
>
> Thank you again for reviewing our paper. We have tried our best to address your questions (see our rebuttal in the top-level comment and above), and revised our paper by following suggestions from all reviewers.
>
> Please kindly let us know if you have any follow-up questions or areas needing further clarification. Your insights are valuable to us, and we stand ready to provide any additional information that could be helpful. Please also **consider raising your score** if you think our questions addressed your initial concerns sufficiently.

---

> ### Author Response · Authors · 2024-11-25
>
> Dear reviewer,
>
> as the **discussion period is coming to a close on Nov 27th**, we would greatly appreciate it if you could clarify any remaining concerns you may have. Please also **leave a comment in case you think that our replies have answered your concerns sufficiently**.

---

### Official Review · Reviewer_2dwE · 2024-11-02

**Soundness:** 2
**Presentation:** 2
**Contribution:** 2
**Rating:** 5
**Confidence:** 3

**Summary:**

The paper proposes a joint continuous diffusion model for mixed type tabular data. Due to the mixed type of data format, the paper designs a score matching method and a score interpolation method for both continuous and categorical data. It presents a loss calibration method to balance the losses between the continuous and categorical data, and proposes an adaptive noise schedule.

**Strengths:**

+ the paper is relatively easy to follow. The presentation of the paper is good
+ comparing with baseline methods, the proposed method seems to be working relatively well, although I have not directly worked on diffusion model for tabular data. There could be related work I am missing
+ contributions and limitations of the paper are properly discussed

**Weaknesses:**

- the scope of the method is relatively limited on just mixed type tabular data. Diffusion model for tabular data had already been studied (e.g. Kotelnikov, 2023), and the contribution of this paper seems to be rather incremental in the sense that it just makes the diffusion model work a bit better in mixed type tabular data
- the method, although makes sense, is not novel in the whole landscape of diffusion models. Modeling continuous and categorical features has been studied in the past and many had been cited by the authors. This seems to be mostly applying these to this particular type of data

**Questions:**

See the weakness section

---

> ### Author Response · Authors · 2024-11-14
>
> Thank you for your constructive comments. Below we address your specific questions.
>
> > the scope of the method is relatively limited on just mixed type tabular data.
>
> It is our **aim to focus on tabular data**! We state this clearly in the introduction as well as the title. As mentioned in our general response, we feel that the field of **tabular data is underresearched and undervalued**. There is a need for improvements in the generation of such data. Tabular data is the most important data in many scientific fields. In the context of diffusion models, tabular data needs special treatments and results and techniques from image diffusion cannot be directly applied. Despite tabular data being one of the predominant data types in industry, there is a **lack of specialized and improved diffusion models** for that data type. Just as we see improvements in diffusion models only focused on image data (see e.g. Stable Diffusion 3), we instead focus on tabular data, as we think this is an underrepresented data format in the current diffusion model literature.
>
> > Diffusion model for tabular data had already been studied (e.g. Kotelnikov, 2023), and the contribution of this paper seems to be rather incremental in the sense that it just makes the diffusion model work a bit better in mixed type tabular data
>
> We are glad that we could convince you that **our model produces SOTA results for mixed-type tabular data**. This is exactly the scope we defined and we do not claim these results would hold for other data types. One cannot expect a model designed for mixed-type data to perform well on e.g. image data or vice versa. All of our design choices have been made with tabular data in mind and **go far beyond the TabDDPM model by Kotelnikov (2023)**, as we highlight in Appendix N of the original submission. In fact, the *only* similarity between CDTD and TabDDPM is their focus on tabular data and the challenge of having to combine different diffusion processes to accommodate mixed-type data. In how this achieved, **our approaches differ substantially** as highlighted in Appendix N.  Our results show that **CDTD outperforms TabDDPM significantly**, not marginally.
>
>
> > the method, although makes sense, is not novel in the whole landscape of diffusion models. Modeling continuous and categorical features has been studied in the past and many had been cited by the authors.
>
> We are *not* making claims about innovating the whole landscape of diffusion models. Instead, **we deliberately focus *only* on mixed-type tabular data**. For this data type, the combination of distinct diffusion processes for continuous and categorical features is critical (as you also see in Kotelnikov (2023)). **Our model makes significant contributions** with the aim of homogenizing the two data types and therewith distinct diffusion processes. We compare our method to other tabular data diffusion models in Appendix N and explicitly highlight important differences. **No previous work addressed the construction of, for instance, homogeneous noise schedules or effective loss balancing.** In our main results, we show that **our CDTD model** that is designed to maximize the homogenization between continuous and categorical feature types **yields SOTA performance** on a diverse set of tabular datasets.

---

> ### Author Response · Authors · 2024-11-21
>
> Dear Reviewer,
>
> Thank you again for reviewing our paper. We have tried our best to address your questions (see our rebuttal in the top-level comment and above), and revised our paper by following suggestions from all reviewers.
>
> Please kindly let us know if you have any follow-up questions or areas needing further clarification. Your insights are valuable to us, and we stand ready to provide any additional information that could be helpful. Please also **consider raising your score** if you think our questions addressed your initial concerns sufficiently.

---

> ### Author Response · Authors · 2024-11-25
>
> Dear reviewer,
>
> as the **discussion period is coming to a close on Nov 27th**, we would greatly appreciate it if you could clarify any remaining concerns you may have. Please also **leave a comment in case you think that our replies have answered your concerns sufficiently**.

---

### Official Review · Reviewer_eakR · 2024-11-04

**Soundness:** 3
**Presentation:** 2
**Contribution:** 3
**Rating:** 6
**Confidence:** 3

**Summary:**

This paper proposes a continuous diffusion model for mixed-type tabular data, which
combines score matching and score interpolation techniques and imposes Gaussian diffusion
processes on both continuous and embedded categorical features.

It proposes several strategies to balance model capacity between continuous and categorical features, including
weighted loss calibration, adjusted model initialization, and the adaptive noise schedule
designs.

**Strengths:**

1. The problem is well motivated, and the intuition of this research is quite clear, which is to propose a unified pipeline to deal
with mixed type tabular data, which contains both continuous and categorical features.  In addition, the proposed strategies effectively deal with the imbalance and calibration between these two types of data.

2. Mathematical derivations are quite solid and appear to be valid, though having not checked completely.

3. Implement comprehensive experiments to demonstrate the effectiveness of the overall pipeline as well as each design component.

**Weaknesses:**

Weaknesses:


1. The overall pipeline has limited novelty, as the key idea to push categorical data into
embedding space and use a Gaussian diffusion process to deal with it is a common
practice.


2. The preliminary section 2.2 that introduces diffusion for categorical features is not as clear
as that in 2.1. I wish the author to elaborate this part a bit more since this should be the
basic foundation of CDTD.


3. The customization on tabular data in section 3.4 is quite heuristic, are they supported by
theoretical proof or are they simply determined from empirical results?

**Questions:**

The section that introduces learnable noise schedules is unclear. For example, its
motivation is confusing.   What does it mean that “given the same embedding dimension,
more noise is needed to remove the same amount of signal from embedding of features
with fewer classes”?

Can you explain the difference between Feature-specific Noise Schedules and Adaptive Noise Schedules in more detail ?

[post rebuttal comment]. Some of the questions above are addressed by the author. I am happy to upgrade my rate for the paper.

---

> ### Author Response · Authors · 2024-11-14
>
> Thank you for your valuable comments. Below we address specific questions.
>
> > The overall pipeline has limited novelty, as the key idea to push categorical data into embedding space and use a Gaussian diffusion process to deal with it is a common practice.
>
> We politely **disagree**. First, the overall pipeline consists of more than pushing categorical data into embedding space, since **we make several improvements** aimed at effectively combining diffusion for continuous features and categorical features. Second, in the domain of tabular data generation, **there exists no other model that models categorical data in embedding space**. Our key idea is not that we treat categories in embedding space. Instead, our **key idea is that improving homogenization of data types in a diffusion framework improves sample quality** and we clearly show in the model setup, our design choices as well as experimental results that this idea works well.
>
>
> > The preliminary section 2.2 that introduces diffusion for categorical features is not as clear as that in 2.1. I wish the author to elaborate this part a bit more since this should be the basic foundation of CDTD.
>
> Thank you for this remark. **We reworked section 2.2** (in blue in the updated pdf) to hopefully make it clearer. We highlighted that score interpolation relies on transforming the score matching problem into a discrete choice problem.
>
>
> > The customization on tabular data in section 3.4 is quite heuristic, are they supported by theoretical proof or are they simply determined from empirical results?
>
> Our choice is determined based on empirical results. Since tabular datasets vary considerably in their structure, features, etc., there is little ground for a theoretical proof. Our choice is based on the observation that in tabular data, unlike image data, the structure of the data matrix is fixed. As such, we only need to fill in the "pixel values", i.e. the details, instead of also generating the low level structure, as in images. **Details are generated at low noise levels, so we overweight the importance of lower noise levels at initialization**. Note that the **initialization only impacts training efficiency**, since a uniformly initialized noise schedule is able to convergence to the exact same final form but with more steps.
>
>
> > The section that introduces learnable noise schedules is unclear. For example, its motivation is confusing. What does it mean that “given the same embedding dimension, more noise is needed to remove the same amount of signal from embedding of features with fewer classes”?
>
> The motivation for learnable noise schedules is simple: Unlike the case of image data, we have no intuition of which noise levels are more or less important for generating tabular data. Since all features can have vastly different distributions, you may need to add only little noise to some features to reach the same signal to noise ratio in the data, but more noise to other features. **With heterogeneous tabular data, the amount of actual signal may vary a lot.** We can intuitively explain this using the example of categorical features you mentioned: Let us assume we have have two categorical features, feature A with 2 categories and feature B with 100 categories. Each of these categories gets assigned an embedding of the same size. The embeddings for feature A have to encode much less information in the same-dimensional space compared to feature B. After all, the embeddings for feature A only have to able to distinguish between two categories! There are a lot more ways to encode the difference between two separate categories in a say 16-dimensional embedding than ways to encode the difference between 100 categories. Hence, to arrive at the same signal-to-noise ratio for both features, we need to add more noise to feature A than B. Another way of thinking is that the embeddings of feature A are just less sensitive to noise because you need less information to begin with to distinguish between two categories than 100. If you need less information, then embeddings with relatively high added noise may still permit a good enough distinction between the two categories.
>
>
> > Can you explain the difference between Feature-specific Noise Schedules and Adaptive Noise Schedules in more detail?
>
> With *adaptive* noise schedule, we mean that the **noise schedules are learned and *adapt* to what is optimal** for the model during training. The term ***feature-specific* only refers to 'how many' different noise schedules** we are training. You could have a single adaptive noise schedule (our 'single' CDTD model) or one noise schedule for each feature, the 'feature-specific' CDTD. On the other hand, if you do not want to use adaptive noise schedules you may still come up with feature-specific noise schedules yourself.

---

> ### Author Response · Authors · 2024-11-21
>
> Dear Reviewer,
>
> Thank you again for reviewing our paper. We have tried our best to address your questions (see our rebuttal in the top-level comment and above), and revised our paper by following suggestions from all reviewers.
>
> Please kindly let us know if you have any follow-up questions or areas needing further clarification. Your insights are valuable to us, and we stand ready to provide any additional information that could be helpful. Please also **consider raising your score** if you think our questions addressed your initial concerns sufficiently.

---

> ### Author Response · Authors · 2024-11-25
>
> Dear reviewer,
>
> as the **discussion period is coming to a close on Nov 27th**, we would greatly appreciate it if you could clarify any remaining concerns you may have. Please also **leave a comment in case you think that our replies have answered your concerns sufficiently**.

---

### Author Response · Authors · 2024-11-14
**Overall Response**

Dear Reviewers,

We would like to thank all reviewers for providing constructive feedback. We are encouraged by the positive statements made. Specifically, that reviewers think our paper:

- proposes strategies that "**effectively deal** with the imbalance and calibration between" (Reviewer eakR) continuous and categorical features;
- implements "**comprehensive experiments** to demonstrate the effectiveness" (Reviewer eakR) and demonstrates "**strong performance** on existing datasets, showing significant improvements compared to baseline methods." (Reviewer cZJV);
- provides **experiments and comparisons** which "look **quite complete** for me: the authors compared with seven methods on eleven datasets, using different kinds of metrics. The analysis of the results are also quite good. Results have shown that their method can achieve **good results and higher efficiency**." (Reviewer 697T);
- and "centers on adjusting noise schedules, which is straightforward and easy to understand, making the **approach accessible and replicable**." (Reviewer cZJV)
- is "easy to follow. The presentation of the paper is good" (Reviewer 2dwE).

Lastly, we want to emphasize that our goal in this paper is, as our title conveys, to improve/develop diffusion models for mixed-type tabular data. We believe that this is an understudied (and perhaps undervalued) type of data. However this type of data is extremely important in many sciences (medicine, economics, social science in general). We are careful to not make claims about the generalization of our framework to any other data types.

Our approach is therefore focused on adapting the diffusion generative framework to tabular data and all of our design choices and model elements reflect this goal, which inherently comes with the challenge of homogenizing different feature types (continuous and categorical in our case).

We **updated the pdf with additional clarifications** (highlighted in blue) based on your comments. Please see our reviewer-specific feedback for more information.

---

### Meta-Review · Area_Chair_Jr6C · 2024-12-22

**Metareview:**

The paper proposes a continuous diffusion model for mixed-type tabular data that combines score matching and score interpolation to handle both continuous and categorical features. The key innovation is ensuring a common continuous noise distribution between feature types and using adaptive noise schedules per feature/data type to manage heterogeneity. The authors claim their method achieves state-of-the-art performance across benchmark datasets and captures feature correlations well. Strengths include its principled approach to balancing continuous and categorical features, comprehensive experiments, and clear problem statement. However, reviewers questioned the technical novelty since similar approaches exist for combining categorical and continuous data diffusion. Additional weaknesses include limited discussion of prior work and lingering questions about the necessity of some model components.

Nevertheless, reviewers failed to address or inadequately addressed many of the author's rebuttal comments in the discussion phase. In reviewing these responses, many of the initial reviewer complaints seem to be reasonably covered in the discussion, aside from claims of novelty. In a final review of the paper, I'm inclined to agree with the borderline positive reviews. The studies presented in the paper seem like they could benefit future research.

**Additional Comments On Reviewer Discussion:**

Reviewers raised concerns about limited technical novelty compared to existing methods for mixed-type data and questioned whether the authors were correctly attributing previous work. The authors responded by clarifying that in the tabular data domain, no other model addresses the homogenization of different data types in the same way, and highlighted their novel loss calibration and noise schedule approaches. They emphasized that tabular data is an undervalued but important domain and that their focus on this specific problem should not diminish the paper's contribution. Some reviewers did not respond, others remained unconvinced by these arguments but acknowledged the practical utility of the method. Authors also argued that their extensive empirical results demonstrate significant improvements over baselines despite concerns about incremental advances.

---

### Decision · Program_Chairs · 2025-01-22

Accept (Poster)